# Efficient Adaptive Federated Optimization

**Su Hyeong Lee**
Department of Statistics
University of Chicago

**Sidharth Sharma**
Department of Computer Science
Columbia University

**Manzil Zaheer**[*]
Google DeepMind

**Tian Li**
Department of Computer Science
University of Chicago

## Abstract

Adaptive optimization is critical in federated learning, where enabling adaptivity on both the server and client sides has proven essential for achieving optimal performance. However, the scalability of such jointly adaptive systems is often hindered by resource limitations in communication and memory. In this paper, we introduce a class of efficient adaptive algorithms, named $\texttt{FedAda}^2$ and its enhanced version $\texttt{FedAda}^2$++, designed specifically for large-scale, cross-device federated environments. $\texttt{FedAda}^2$ optimizes communication efficiency by avoiding the transfer of preconditioners between the server and clients. Additionally, $\texttt{FedAda}^2$++ extends this approach by incorporating memory-efficient adaptive optimizers on the client side, further reducing on-device memory usage. Theoretically, we demonstrate that $\texttt{FedAda}^2$ and $\texttt{FedAda}^2$++ achieve the same convergence rates for general, non-convex objectives as its more resource-intensive counterparts that directly integrate joint adaptivity. Extensive empirical evaluations on image and text datasets demonstrate both the advantages of joint adaptivity and the effectiveness and efficiency of $\texttt{FedAda}^2$/$\texttt{FedAda}^2$++.

## 1 Introduction

Federated learning is a distributed learning paradigm which aims to train statistical models across multiple clients while minimizing raw data exposure [1, 2, 3]. In vanilla federated learning, a central server orchestrates the training process by distributing the global model to a subsample of thousands or even millions of clients. These clients collaboratively perform local stochastic gradient descent while drawing from their private data streams. After several epochs have elapsed, each client communicates their aggregate updates to the server, which averages this information to make an informed adjustment to the global model. This algorithm, using non-adaptive weight updates, is called *FedAvg* [1]. A recent trend is to investigate utilizing adaptive optimizers to support federated learning [4]. Adaptivity can be employed in either the server-side or the client-side, where joint adaptivity (consisting of global *and* local adaptive updates) has been shown to play a pivotal role in accelerating convergence and enhancing accuracy [5, 6].

Nevertheless, efficiency challenges remain for the successful deployment of jointly adaptive algorithms in practice, especially in cross-device federated settings [7]. The server, which collects pseudogradients pushed by participating clients, consolidates a global approximation of the preconditioners for adaptive model updates. Typically, the server sends the preconditioners back to the clients to precondition local adaptive updates. However, this can lead to significant communication overhead that detracts from the advantages offered by adaptivity [8]. Furthermore, dynamically varying client

---

[*]Work done while at Google DeepMind.

39th Conference on Neural Information Processing Systems (NeurIPS 2025).

resource limitations restrict the reliability of client-side adaptive optimizers in practice, especially when additional memory is required for handling local preconditioners during each client model update.

In this work, we propose a class of efficient jointly adaptive distributed training algorithms, called $\texttt{FedAda}^2$ and $\texttt{FedAda}^2$++, to mitigate the aforementioned communication and memory restrictions while retaining the benefits of adaptivity. $\texttt{FedAda}^2$ maintains an identical communication complexity as the vanilla FedAvg algorithm. Instead of transmitting global server-side preconditioners from the server to the selected clients, we propose the simple strategy of allowing each client to initialize local preconditioners from constants such as zero, without any extra communication of preconditioners[2]. $\texttt{FedAda}^2$++ expands on this approach by adopting existing memory-efficient optimizers that factorize gradient statistics to reduced dimensions in order to save on-device memory when synthesizing local updates. We prove that for the general, non-convex setting, $\texttt{FedAda}^2$ and $\texttt{FedAda}^2$++ achieve the same convergence rate as prior adaptive federated optimizers (e.g., [4]). Conclusively, we aim to demonstrate that jointly adaptive federated learning, as well as adaptive client-side optimization, are practicable in real-world settings while sidestepping localized memory restrictions and communication bottlenecks with minimal performance degradation.

Our contributions may be summarized as follows.

- Motivated by the importance of joint server- and client-side adaptivity, we propose $\texttt{FedAda}^2$ and $\texttt{FedAda}^2$++ to avoid extra communication cost and reduce on-device memory while retaining the benefits of joint adaptive optimization (Section 3).

- We provide convergence analyses for a class of $\texttt{FedAda}^2$/$\texttt{FedAda}^2$++ algorithms instantiated with different server- and client-side adaptive methods and memory-efficient local optimizers (Section 4).

- Empirically, we show that $\texttt{FedAda}^2$/$\texttt{FedAda}^2$++, without transmitting preconditioners and employing on-device preconditioner compression, matches the performance of its more expensive counterparts, and outperforms baselines without joint adaptivity on both image and text datasets (Section 5).

## 2   Related Work

We now provide a brief overview of related work in adaptive federated learning and memory-efficient[3] preconditioning.

**Adaptive Federated Optimization.**   Adaptive optimization preconditions the gradients to enhance optimization efficacy, dynamically adjusting the learning rate for each model parameter [e.g., 9, 10, 11]. Recent developments in federated learning have leveraged adaptive methods for server and client model parameter updates. Frameworks such as FedAdam [4] and FederatedAGM [12] focus primarily on server-side adaptivity while using a constant learning rate for client updates. Additionally, FedCAMS [8] delves into communication-efficient adaptive optimization by implementing error feedback compression to manage client updates while maintaining adaptivity solely on the server side. Conversely, methodologies such as FedDA [13], FedLALR [14], Local AdaAlter [15], and Local AMSGrad [16] have adopted client-side adaptivity exclusively and demonstrate benefits. These approaches involve transmitting both client preconditioners and model parameters for global aggregation in the server. Moreover, some frameworks have embraced joint adaptivity. Local Adaptive FedOPT [5] implements joint adaptivity while incorporating an additional client correction term. These terms, along with transmitted client pseudogradients, are aggregated on the server to construct a global preconditioner used to synthesize the subsequent model update. Alternatively, frameworks such as MIME [17, 18] transmit additional optimizer state information aggregated

---

[2]We note that this pragmatic strategy has briefly appeared in a prior work [5] before, but lacked formal convergence guarantees and was not extensively studied via thorough empirical evaluations. We include a detailed discussion in Appendix B

[3]There are various notions of 'efficiency' of adaptive methods in the context of the federated learning, two of them being communication efficiency and client memory efficiency. Our contribution specifically targets reducing communication and memory costs incurred by *local preconditioners*, which is complementary with works that reduce communication by repeated local updates or model weight/pseudogradient compression (e.g., FedCAMS [8]) and may, in theory, even be combined. We note that such methods tend to study compression of local models for *server-only* adaptive optimizers, whereas our contribution lies in minimizing communication of preconditioners in *jointly adaptive* settings.

in the server to mimic adaptive updates in centralized settings, while maintaining frozen-state optimizers on the client-side. In contrast with all these approaches, FedAda$^2$ avoids the transmission of any local/global preconditioners and optimizer states entirely, maintaining precisely identical communication complexity as vanilla FedAvg despite leveraging joint adaptivity. We include further discussions in Appendix H.5.

**Memory-Efficient Adaptive Optimizers.** The implementation of local adaptive methods substantially increases client memory requirements, as it necessitates the maintenance of local preconditioners. For some models, it has been noted that the gradients combined with optimizer states consume significantly more memory than the actual model parameters themselves [19]. Memory-efficient adaptive optimizers have been extensively studied in prior literature. Algorithms such as Adafactor [20] address memory reduction by tracking moving averages of the reduction sums of squared gradients along a singular tensor axis, attaining a low-rank projection of the exponentially smoothed preconditioners. GaLore [21] targets the low-rank assumption of the gradient tensor, which reduces memory of both gradients and preconditioners. Shampoo [22] collapses gradient statistics into separate preconditioning matrices for each tensor dimension, which is extended via extreme tensoring [23]. In this paper, we focus on SM3 [24] in our implementation and experiments due to its empirical performance; however, our theoretical framework covers a broad class of memory-efficient optimizers applied on the client-side (Section 4 and Appendix E).

## 3  FedAda$^2$: Efficient Joint Server- and Client-Side Adaptivity

In federated learning, a typical server-side objective is formed by taking an average of all client objectives $F_i(x)$ for $i \in [N]$ and $x \in \mathbb{R}^d$:

$$f(x) = \frac{1}{N} \sum_{i=1}^{N} F_i(x). \tag{1}$$

In the case of unbalanced client data sizes or sampling probabilities, the objective becomes $\sum_{i=1}^{N} p_i F_i(x)$ on the right hand side where $p_i$ is proportional to the local data size of client $i$, or the sampling probability. With a slight abuse of notation, we denote $F_i(x) = \mathbb{E}_{z \sim \mathcal{D}_i}[F_i(x, z)]$ where $F_i(x, z)$ is the stochastically realized local objective and $\mathcal{D}_i$ is the data distribution of client $i$. The convergence analysis developed in Section 4 holds when $\mathcal{D}_i$ is taken to be the local population distribution, as well as when $\mathcal{D}_i$ is the local empirical distribution.

A key characteristic of cross-device federated settings is that clients cannot store or maintain 'states' across communication rounds [7]. To facilitate joint adaptivity in stateless federated systems, a natural baseline is to estimate pseudogradient statistics on the server (i.e., maintaining server-side or global preconditioners), which are then transmitted to to all participating clients at each communication round. Selected clients then perform local adaptive updates using preconditioners initialized from the global values. While this allows clients to leverage global preconditioner information for local model adjustments, transmitting global pseudogradient statistics, such as second moments, at every round substantially increases communication costs. On the other hand, warm-starting from previously stored preconditioners may not be beneficial, especially when the interval between consecutive rounds involving the same client is large so that the preconditioners can be very stale. Additionally, performing local adaptive updates induces client-side memory overhead. In the following, we discuss two key techniques for efficient federated adaptive optimization with convergence guarantees.

**Zero Local Preconditioner Initialization.** To enhance the feasibility of jointly adaptive federated learning in cross-device settings, we first address extra major communication bottlenecks brought by transmitting global preconditioners from the server to a subset of clients. We propose a simple strategy of uniformly initializing local preconditioners to zero (or some constant vector) at the beginning of each training round, thus eliminating the need for preconditioner transmission.

To describe the process in more detail, assume Adagrad (with momentum) as the server-side optimizer [4] for illustration purposes. We have the following server update rule (SU) for $-\Delta_i^t$ the accumulated pseudogradient from client $i$ at step $t$,

$$\Delta_t = \frac{1}{|\mathcal{S}^t|} \sum_{i \in \mathcal{S}^t} \Delta_i^t, \quad \widetilde{m}_t = \widetilde{\beta}_1 \widetilde{m}_{t-1} + (1 - \widetilde{\beta}_1)\Delta_t,$$
$$\widetilde{v}_t = \widetilde{v}_{t-1} + \Delta_t^2, \quad x_t = x_{t-1} + \eta \frac{\widetilde{m}_t}{\sqrt{\widetilde{v}_t} + \tau}. \tag{SU}$$

Here, $\widetilde{v}_t$ is the sum of squared server-side pseudogradient $-\Delta_t$, and $\widetilde{\beta}_1$ is the momentum coefficient controlling the moving average $\widetilde{m}_t$ of $-\Delta_t$. The set $\mathcal{S}_t \subset [N]$ gives the index of all participating clients at round $t$, and $\tau$ is a constant. An extension to the case when Adam is selected as the server optimizer is given in Appendix D.2. After obtaining an updated global preconditioner $\widetilde{v}_t$ at each communication round, in FedAda$^2$, the server does not communicate $\widetilde{v}_t$ to the participating clients. Instead, each client only receives $x_t$ and initializes the local preconditioners from zero. Empirically, we demonstrate this simple strategy does not degrade the performance relative to the alternative of transmitting global preconditioners, while being significantly more communication efficient for adaptive methods beyond AdaGrad (Section 5.1). In addition to communication reduction, this approach enables the use of different optimizers on the server and clients, as the server and client can maintain independent gradient statistic estimates. We further discuss the theoretical guarantees as well as implications of this general framework in Section 4 and Appendix E.

**Addressing Client-Side Resource Constraints.** To accommodate local memory restrictions, we further employ existing memory-efficient optimizers for all clients in FedAda$^2$++. Our framework allows any such optimizer to be used, including a heterogeneous mixture within each communication round. We provide a convergence guarantee for a very broad class of optimizer strategies in Theorem 4.1. We note that in order for convergence to be guaranteed, the memory-efficient optimizer must satisfy the conditions of Theorem E.1, which are non-restrictive[4]. The FedAda$^2$++ framework is summarized in Algorithm 1 below, presented in a simplified form. Local statistics or global statistics refer to those used to construct preconditioners (e.g., first or second moment), and selecting a memory-efficient optimizer strategy gives FedAda$^2$++.

---

**Algorithm 1** FedAda$^2$: Efficient Jointly Adaptive Optimization Framework (Simplified)

---

**Require:** Initial model $x_0$, total number of clients $N$
1: **for** $t = 1, \ldots, T$ **do**
2:      Sample client subset $\mathcal{S}^t \subset [N]$
3:      **for** each client $i \in S_l^t$ (in parallel) **do**
4:          $x_{i,0}^t \leftarrow x_{t-1}$
5:          local_statistics $\leftarrow 0$
6:          **for** $k = 1, \ldots, K$ **do**
7:              Draw gradient $g_{i,k}^t \sim \mathcal{D}_{i,\mathrm{grad}}(x_{i,k-1}^t)$
8:              $x_{i,k}^t \leftarrow \mathrm{Adap\_Opt.}(x_{i,k-1}^t, g_{i,k}^t, \text{local\_statistics})$
9:              (FedAda$^2$++: memory-efficient $\mathrm{Adap\_Opt.}$)
10:         **end for**
11:         $\Delta_i^t = x_{i,K}^t - x_{t-1}$
12:      **end for**
13:      $x^t \leftarrow \mathrm{Adap\_Opt.}(\{\Delta_i^t\}_{i \in S_l^t}, \text{global\_statistics})$ (for example, Eq. (SU))
14: **end for**

---

During implementation, we have chosen to instantiate FedAda$^2$++ with SM3 adaptations of Adam and Adagrad as the memory-efficient local optimizers (Appendix C) due to its strong empirical performance. Intuitively, SM3 exploits natural activation patterns observed in model gradients to efficiently synthesize a low-rank approximation of the preconditioner. It maintains the statistics in the granularity of parameter groups instead of individual coordinates. Our analyses in Section 4 hold for a class of memory-efficient local optimizers. Due to the highly technical nature of its implementation and the convergence analysis, specific algorithm design details and proofs have been relegated to Appendix C, D for interested readers only. Our convergence analysis further accounts for the use of delayed preconditioner updates, where clients update their second-moment gradient statistics after every $z$ minibatch backpropagations, introducing a $z$-step delay (see Algorithm 2, Appendix J.3).

## 4 Convergence Analyses

One of the challenges in proving the convergence bound for jointly adaptive systems lies in handling client-side adaptivity with multiple local updates. The individual gradients may not be linearly sepa-

---

[4]It can easily be shown that Adam, AdaGrad, SGD, as well as their memory-efficient counterparts [24] for the first two, all satisfy the optimizer conditions for guaranteed convergence.

rable due to dependencies between historical client gradients in the local model updates. Furthermore, the combination of server- and client-side adaptivity complicates analysis relative to prior works focusing on only server-side or only client-side adaptivity. To address these issues, we assume access to full batch client gradients, but allow for client partial participation.

**Assumption 1 ($L$-Smoothness).** The local objectives are $L$-smooth and satisfy $\|\nabla F_i(x) - \nabla F_i(y)\| \leq L\|x - y\|$ for all $x, y \in \mathcal{X}$ and $i \in [N]$.

**Assumption 2 (Bounded Gradients).** The full gradient of the local objective is bounded: $\left|[\nabla F_i(x)]_j\right| \leq G$ for $j \in [d], i \in [N]$.

We note that although we assume a uniform upper bound on the full batch gradient (as opposed to stochastic gradients) for the convergence results to hold. These assumptions are standard within the literature and have been used in previous works [15, 25, 26]. In particular, this delineates an $\widetilde{L}$-Lipschitz family of objectives given that the arguments are $\eta_\ell \varepsilon_s$-bounded away from each other,

$$\|\nabla F_i(x) - \nabla F_j(y)\| \leq \widetilde{L}\|x - y\| := \frac{2\sqrt{d}G}{\eta_\ell \varepsilon_s}\|x - y\|$$

for $i, j \in [N]$ and $\|x - y\| \geq \eta_\ell \varepsilon_s$. Here, $\varepsilon_s$ is an epsilon smoothing term that activates on the client side and $\eta_l$ is the local learning rate. This quantity is used in a gradient clipping step in `FedAda`$^2$ (full version Algorithm 5), where if the local gradient update is negligibly small in magnitude, then the gradient is autonomously clipped to 0. $\eta_\ell > 0$ is the local learning rate, and in particular, we note that $\widetilde{L} = \Theta(\eta_\ell^{-1})$. By taking $\varepsilon_s \to 0$, our algorithm recovers federated algorithms that do not utilize local gradient clipping. The definition the clipping threshold $\varepsilon_s$ is purely for analytical purposes; in our experiments, we take $\varepsilon_s$ to be a negligible value so that $m_k$ is not set to 0.

We now provide a convergence bound for the general, non-convex case under local gradient descent and partial client participation. The full theorem statement is provided in Appendix E as Theorem E.1. The SM3 instantiation of `FedAda`$^2$`++`, as well as the generalization to the case where we use Adam as the server/client optimizers are provided in Appendices D.1 and D.2. We note that the convergence bounds are derived for the deterministic setting. Extending the analysis to stochastic regimes and establishing analogous guarantees is a natural direction for future work.

**Theorem 4.1** (Simplified). *Under Assumptions 1 and 2 as well as some non-restrictive optimizer update conditions (Theorem E.1), for any choice of initialization $x_0$, Algorithm 1 satisfies*

$$\min_{t \in [T]} \|\nabla f(x_{t-1})\|^2 \leq \frac{\Psi_1 + \Psi_2 + \Psi_3 + \Psi_4 + \Psi_5}{\Psi_6}$$

*where asymptotically,*

$$\Psi_1 = \Theta(1), \ \Psi_2 = \eta^2 \eta_\ell^2 T, \ \Psi_3 = \eta \eta_\ell^2 T, \ \Psi_4 = \eta \eta_\ell \log(1 + T\eta_\ell^2),$$

*and*

$$\Psi_5 = \begin{cases} \eta^3 \eta_\ell^3 T & \text{if } \mathcal{O}(\eta_\ell) \leq \mathcal{O}(1) \\ \eta^3 \eta_\ell T & \text{if } \Theta(\eta_\ell) > \Omega(1) \end{cases}, \quad \Psi_6 = \begin{cases} \eta \eta_\ell T & \text{if } \mathcal{O}(T\eta_\ell^2) \leq \mathcal{O}(1) \\ \eta\sqrt{T} & \text{if } \Theta(T\eta_\ell^2) > \Omega(1) \end{cases}.$$

We defer the detailed proofs and complete statement of the bounds to Appendix D and E. A key theoretical insight in our work is that the benefits of joint adaptivity can be retained even without transmitting preconditioners. The condition $\mathcal{O}(\eta_\ell) \leq \mathcal{O}(1)$ refers to learning rates that are asymptotically constant or decaying with $T$, e.g., $\eta_\ell = 1/T, 1/T^2$, etc. Our convergence allows for delayed preconditioner updates. In Lemma D.1 in the appendix, we observe that a larger decay parameter $z$ can potentially slow down convergence. Empirically, the performance is robust to the choice of $z$ (Appendix J.3). We make no other assumptions on local or global learning rates ($\eta_l$ and $\eta$) to extract the most general use of Theorem 4.1. We have the following two corollaries.

**Corollary 4.2.** *Any of the following conditions are sufficient to ensure convergence of Algorithm 1:*

$$(A): \ \eta_\ell \leq \mathcal{O}(T^{-\frac{1}{2}}) \quad \text{for} \quad \Omega(T^{-1}) < \eta \eta_\ell < \mathcal{O}(1),$$
$$(B): \ \eta_\ell = \Theta(T^{-\frac{49}{100}}) \quad \text{for} \quad \Omega(T^{-\frac{1}{2}}) < \eta < \mathcal{O}(T^{\frac{12}{25}}).$$

**Corollary 4.3.** *Algorithm 1 converges at rate $\mathcal{O}(T^{-1/2})$.*

**Discussions of Convergence Bound.** There have been several recent works exploring adaptivity and communication efficiency in federated learning. The convergence rate in Corollary 4.3 matches the state-of-the-art for federated non-convex optimization methods [4, 8, 12, 14, 15, 16]. However, to the best of our knowledge, there are no known convergence results of jointly adaptive federated optimization that explicitly support several popular methods including Adam and AdaGrad.

**Importance of Adaptivity Parameters.** The $\varepsilon$-smoothing term (i.e., adaptivity parameter) is crucial for maintaining stability and ensuring convergence in adaptive optimizers. For example, PyTorch's implementation of Adam adopts $\varepsilon = 10^{-8}$ as its default value. In our convergence bounds (c.f., Theorem E.1, full version), the smoothing term explicitly appears in the denominator on the right-hand side of the convergence result via $\widetilde{L}$ and $\tau$. Setting $\varepsilon = 0$ or $\tau = 0$ causes the right-hand side of the convergence bounds to diverge, thereby undermining convergence. To address this, we ensure $\tau, \varepsilon > 0$ in our algorithm and impose the smoothing condition $m_l > 0$ in Theorem D.6 (the full version of Theorem 4.1).

**Extension of** `FedAda`$^2$**: Blended Optimization.** The gradient descent setting used in the analysis of Theorem 4.1 is conceptually equivalent to accessing oracle client workers capable of drawing their entire localized empirical data stream. While this constraint is a limitation of our theory, it enables us to derive stronger results and induce additional adaptive frameworks for which our analysis generalizes. For instance, our bound deterministically guarantees asymptotic stabilization of the minimum gradient, regardless of initialization or client subsampling procedure. In Appendix E, we extend the `FedAda`$^2$ framework to a even more general framework of federated blended optimization.

Blended optimization distributes local optimizer strategies during the subsampling process, which are formalized as functions that take as input the availability of client resources and outputs hyperparameters such as delay step size $z$ or choice of optimizer (Adam, AdaGrad, SGD, etc). These may be chosen to streamline model training based on a variety of factors, such as straggler mitigation or low availability of local resources. In particular, this framework permits the deployment of different adaptive optimizers per device for each round, enhancing the utility of communication-efficient frameworks that do not retain preconditioners between clients or between the server and client. This flexibility is especially beneficial in scenarios where there is non-uniformity between server and client adaptive optimizer choices, or between client-side optimizers.

## 5 Empirical Evaluation

In this section, we empirically demonstrate the performance of `FedAda`$^2$/`FedAda`$^2$++ compared with several baselines that are either non-adaptive or adaptive but inefficient. We first present our main results by comparing different instantiations of `FedAda`$^2$ with more expensive jointly adaptive baselines and non-jointly adaptive methods in Section 5.1. We then investigate the effects of hyperparameters in more detail in Section 5.2. We repeat every run for 20 times under different random seeds for statistical significance, and report $95\%$ confidence intervals as shaded error regions in all plots. Our experiments are designed to reflect memory- and communication-constrained scenarios, and we thoroughly demonstrate that our algorithms retain strong convergence behavior in such environments.

**Evaluation Setup.** We explore the impact of adaptivity on both text and image datasets, i.e., StackOverflow [27], CIFAR-100 [28], FEMNIST [29], and GLD-23K [30]. In StackOverflow, each client is a single user posting on the StackOverflow website. For images, we explore vision transformer models (ViT-S [31]) which are pretrained on ImageNet-21K [32], and finetune them the Google Landmarks dataset [30]. This represents a domain shift onto natural user-split pictorial data. We use the same model on the CIFAR-100 dataset [28], where we partition the data using LDA [33] with $\alpha = 0.001$, a non-IID statistical topic modeling algorithm. To assess the performance of all algorithms in an additional realistic heterogeneous federated learning scenario, we further utilize FEMNIST [29] where each client is an individual writer. This setup evaluates federated learning algorithms under non-IID conditions, highlighting challenges such as personalization and robustness to client heterogeneity. Details for federated dataset statistics, learning tasks, and hyperparameter tuning are provided in Appendix I.

**Description of Baselines.** Throughout this section, we compare with the following baselines. FedAvg is the vanilla FL algorithm introduced in McMahan et al. [1], without any additional

momentum for the server-side aggregation. FedAdaGrad or FedAdam are two examples of server-only adaptive federated optimization methods [4], where the server-side model updates are performed by an adaptive optimizer (e.g., AdaGrad/Adam) instead of vanilla averaging. 'Costly Joint Adaptivity' (named *Costly Joint Adap.* in the captions) indicates a jointly adaptive training regimen, where server-side preconditioners are transmitted to clients at every communication round. For instance, we may denote one such setup as 'AdaGrad-AdaGrad', where server-side AdaGrad preconditioners are distributed to the client-side AdaGrad optimizers for local preconditioner initialization. Removing server-side preconditioner transmission and using zero initialization of client-side preconditioners naturally results in a corresponding instantiation of $\texttt{FedAda}^2$, which is communication-efficient. Further compressing the local preconditioners using SM3 [24] to account for client memory resource limitations gives $\texttt{FedAda}^2$++. Therefore, the baselines and $\texttt{FedAda}^2$/$\texttt{FedAda}^2$++ may be viewed as natural and well-motivated variations via the addition of jointly adaptive updates and memory-efficient optimizers.

## 5.1 Empirical Performance of $\texttt{FedAda}^2$ and $\texttt{FedAda}^2$++

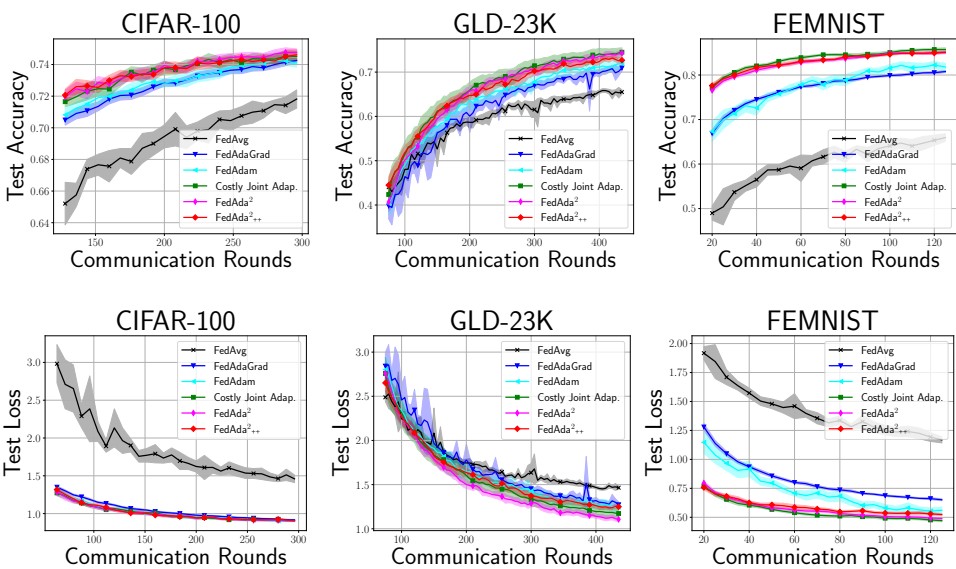

Figure 1: (Top) Test accuracies on CIFAR-100, GLD-23K, and FEMNIST datasets. If not otherwise specified, CIFAR-100 and GLD-23K use Adam for adaptivity. We see that jointly adaptive algorithms demonstrate improved performance over FedAvg and server-only adaptive systems. Furthermore, curtailing global preconditioner transmission in $\texttt{FedAda}^2$/$\texttt{FedAda}^2$++ does not degrade performance, and preserves the benefits of joint adaptivity while maintaining efficiency. (Bottom) Corresponding test losses resultant from model training on the three datasets.

$\texttt{FedAda}^2$ and $\texttt{FedAda}^2$++ **for Training Transformers.** We investigate the performance of finetuning vision transformer models (ViT-S [31]) on image data. For all runs on the CIFAR-100, FEMNIST, and GLD-23K datasets, we select Adam as the adaptive optimizer instantiation, except for the FedAdaGrad baseline. For CIFAR-100 (Figure 1, leftmost column), jointly adaptive and server-only adaptive methods (FedAdaGrad and FedAdam) converge faster and achieve higher accuracy than FedAvg. Methods utilizing joint adaptivity, including $\texttt{FedAda}^2$, show slightly faster convergence than FedAdam. While 'Costly Joint Adap.' attains similar performance to $\texttt{FedAda}^2$++, the latter is much more memory and communication efficient. Similar trends are observed on GLD-23K (middle column). The superior performance of jointly adaptive methods are especially pronounced for FEMNIST (rightmost column), where a significant gap can be observed between non-adaptive FedAvg, server-only adaptive FedAdam/FedAdaGrad, and the jointly adaptive $\texttt{FedAda}^2$/$\texttt{FedAda}^2$++, respectively. Additionally, in Appendix C, we incorporate the technique of delayed local preconditioner updates [22] to further mollify the computation burden on the clients, and verify that $\texttt{FedAda}^2$/$\texttt{FedAda}^2$++ are robust to the said delayed updates (Appendix J.3).

**Effectiveness under Additional Adaptive Optimizers.** Algorithm 1 provides a general framework, and in Figure 1, we focus on symmetric server-client optimizer configurations (e.g., Adam-Adam,

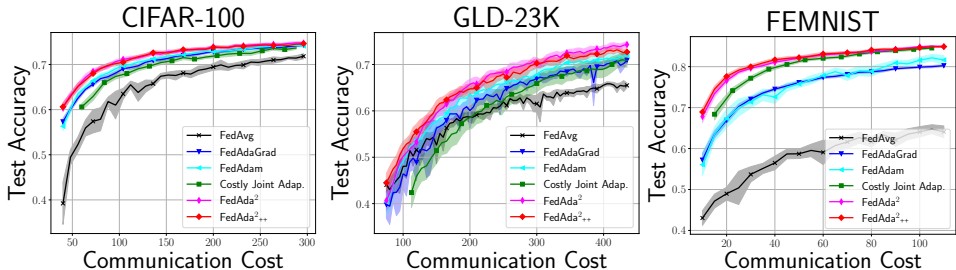

Figure 2: Test accuracies against actual communication cost (total transmitted bits normalized to that of FedAvg) of all algorithms under the same settings as in Figure 1. When controlling for communication complexity, FedAda$^2$/FedAda$^2$++ attain the fastest convergence over all baselines. That is, the improvement is even more significant when measured in actual communication cost, where FedAda$^2$/FedAda$^2$++ achieve faster convergence with fewer bits transmitted than even costly joint adaptivity.

AdaGrad-AdaGrad). Appendix J contains ablations on the delay parameter $z$, including what happens when there is a mismatch between the client and server adaptive optimizers (e.g., server-side Adam preconditioners communicated to client-side AdaGrad). That is, in Appendix J.2, Figure 10, we examine the performance of *asymmetric* server-client adaptivity setups under both jointly adaptive baselines and FedAda$^2$/FedAda$^2$++. Surprisingly, our results show that in the Costly Joint Adaptivity baseline, employing an unbalanced preconditioner (e.g., transmitting the server-side Adam preconditioner to client-side AdaGrad), does not significantly impact performance across a hyperparameter sweep. Additionally, FedAda$^2$ as well as FedAda$^2$++ demonstrates robust training dynamics across various adaptivity instantiations with non-existent performance degradation, highlighting its effectiveness in enabling efficient jointly adaptive optimization.

**Reduced Communication and Memory.** As discussed in Section 1, while joint adaptivity has been shown to substantially improve performance, its scalability and practical deployment are significantly hindered by communication and memory complexity in real-world settings. To mitigate this bottleneck, FedAda$^2$ forcibly matches the communication efficiency of FedAvg while retaining the advantages of joint adaptivity, while FedAda$^2$++ further compresses client memory with convergence guarantees (Theorem 4.1). In Figure 2, when evaluating convergence in terms of the actual communicated bits (communication rounds times number of bits per round), FedAda$^2$/FedAda$^2$++ significantly outperforms costly joint adaptivity, saving significant communication bandwidth. For instance, in the case of ViT, when FedAda$^2$++ is instantiated via SM3, storing second-moment estimates for preconditioning requires only 0.48% additional memory, compared to an $1\times$ increase otherwise. This corresponds to a 99% reduction in the extra client memory required for deploying joint adaptivity, making it far more practical for large-scale applications. In Table 1, we summarize the communication complexity and memory efficiency of FedAda$^2$/FedAda$^2$++ and baselines, compared to alternative adaptive frameworks such as MIME or MIMELite [17, 34]. Communication is the two-way (server-to-client and client-to-server) cost including both model parameters and preconditioners, and computation is the number of gradient calls per local iteration. Client-side memory cost includes the cost to maintain both gradients and potentially gradient statistics.

| Method | Joint Adaptivity? | Communication | Computation (#grad. calls) | Memory (client) |
|---|---|---|---|---|
| FedAvg | N | 2d | 1 | d |
| FedAdaGrad | N | 2d | 1 | d |
| FedAdam | N | 2d | 1 | d |
| MIME | N | 5d | 3 | 4d |
| MIMELite | N | 4d | 2 | 3d |
| Costly Joint Adap. | Y | 3d | 1 | 2d |
| FedAda$^2$ | Y | 2d | 1 | 2d |
| FedAda$^2$++ | Y | 2d | 1 | $\sim$ d |

Table 1: Comparison of various algorithms (assuming AdaGrad as the adaptive optimizer). $d$ denotes the model dimension.

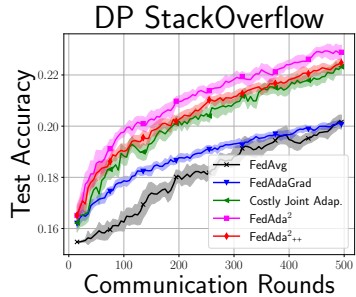

Figure 3: Performance of training Stack-Overflow in the private setting.

**Improvements in Private Settings.** Another critical constraint for federated learning is user privacy, and federated learning itself may not offer formal privacy guarantees [35]. Hence, additional privacy mechanisms are critical to provably protect user information. We focus on the statistical framework of differential privacy [36] where it guarantees that any third-party attacker cannot infer if any user participates in training or not [37]. We apply the popular subsampled Gaussian mechanism [38] to privatize all the methods, using noise multiplier $\sigma = 1$, which provides a privacy budget of $(\varepsilon, \delta) = (13.1, 0.0025)$ with optimal Rényi-Differential Privacy (RDP) [39] order 2.0. On the StackOverflow dataset with a logistic regression model (Figure 3), we observe that $\texttt{FedAda}^2$ and $\texttt{FedAda}^2\texttt{++}$ again outperform other baselines by a large margin with privacy constraints.

## 5.2 Effects of Varying Configurations

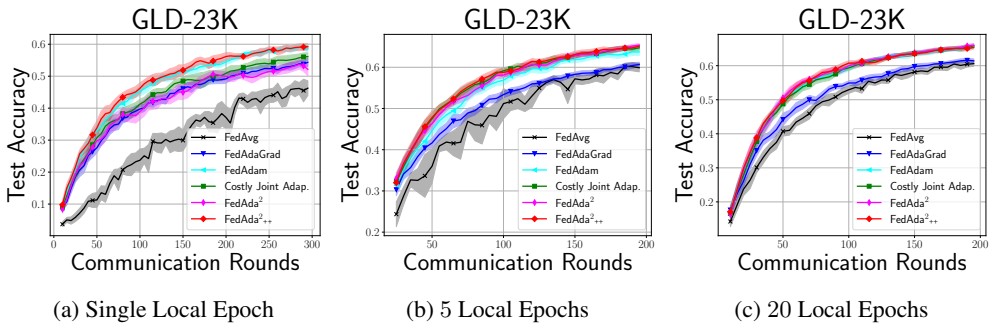

(a) Single Local Epoch     (b) 5 Local Epochs     (c) 20 Local Epochs

Figure 4: Algorithm testing performance comparison under varying client resource limitations (i.e., number of local epochs). When resources are constrained, $\texttt{FedAda}^2\texttt{++}$ converges the fastest, followed closely by FedAdam. Interestingly, the relative performance advantage of $\texttt{FedAda}^2\texttt{++}$ becomes less significant as the number of local epochs increases.

**Dynamics of $\texttt{FedAda}^2$ under a Varying Number of Local Epochs.** In Figure 4, we study the transfer learning setting of a vision model under a highly constrained, moderate, and sufficient client computation budget, corresponding to running 1, 5, and 20 local epochs on the clients. We see that when the number of epochs is low (Figure 4 (a)), $\texttt{FedAda}^2\texttt{++}$ achieves the best performance, closely followed by FedAdam. Interestingly, as the clients' computational budget increases, the relative performance advantage of $\texttt{FedAda}^2\texttt{++}$ diminishes. In such scenarios, jointly adaptive optimization outperforms FedAdam, although the margin is not substantial.

**Sensitivity to Hyperparameters.** In Figure 9 in the appendix, we plot test accuracies over the hyperparameter sweeps (all hyperparameters in Appendix I) for all algorithms, on the StackOverflow dataset. Server-only adaptivity stabilizes the performance of FedAvg, and costly joint adaptivity further enhances the stabilized accuracies. However, eliminating server preconditioner transmission in $\texttt{FedAda}^2$ destabilizes the accuracy, resulting in significantly poorer performance for the worst losses while retaining the best performing losses. We see that there is a wide range of hyperparameter under which $\texttt{FedAda}^2$ and $\texttt{FedAda}^2\texttt{++}$ have superior performance compared with the baselines. In particular, approximating the preconditioners in a memory-efficient manner using SM3 in $\texttt{FedAda}^2\texttt{++}$ is rather robust to hyperparameters, which we hypothesize is due to the denoising effect of projections during SM3 compression, previously unreported within the literature.

**Summary.** Across all datasets, we empirically demonstrate the benefits of joint server- and client-side adaptivity. However, vanilla implementation of joint adaptivity with transmitted global preconditioners can be expensive. We consistently find that initializing local preconditioners from zero ($\texttt{FedAda}^2$) does not underperform (yet sometimes outperforms) costly joint adaptivity with full server-side preconditioner transmission, while being much more efficient (Table 1). In general, we observe that $\texttt{FedAda}^2$ and $\texttt{FedAda}^2\texttt{++}$ algorithms retain the competitive advantage of joint adaptivity while being communication- and memory-efficient under a variety of optimizers, datasets, and model architectures. Across our experiments, we consistently find that adaptive methods (e.g., FedAdam, FedAdaGrad) outperform FedAvg, while jointly adaptive methods ($\texttt{FedAda}^2/\texttt{FedAda}^2\texttt{++}$) consistently outperform server-only adaptive baselines.

## 6 Conclusion and Future Work

In this work, we introduce FedAda$^2$ and FedAda$^2$++, a class of jointly adaptive algorithms designed to enhance scalability and performance in large-scale, cross-device federated environments. By mitigating the transfer of costly preconditioners in jointly adaptive methods, FedAda$^2$-class algorithms significantly reduce the communication overhead and extra on-device memory cost without degrading model performance. Our theoretical convergence guarantees and empirical results demonstrate the practical benefits of FedAda$^2$ in real-world federated learning scenarios. In Appendix A, we further extend our work by providing a novel regret-based theoretical analysis of how leveraging client-side adaptivity improves distributed learning for interested readers. In particular, we study the heavy-tailed regime, which sheds light on how client-side adaptivity mitigates the propagation of noisy updates–particularly relevant in transformer-based training. A promising future direction would be investigating how diverse client optimizer selections could combine their respective advantages in aggregate performance. Additionally, researchers could explore adapting various optimizer strategies across different timescales in blended optimization to better accommodate dynamically varying client resource constraints.

## Acknowledgments

We thank the anonymous reviewers for their insightful feedback, which has helped to improve the quality of this paper. The authors declare no conflicts of interest.

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

# Contents

# A   Importance of Client-Side Adaptivity

In this section, we motivate our work by providing a theoretical description of how leveraging client-side adaptivity improves distributed learning, which is validated in experiments (Section 5). Our analyses are motivated by prior works that uncover critical conditions under which centralized SGD can diverge, specifically in settings involving heavy-tailed gradient noise [40]. After analyzing the importance of client-side adaptivity, we propose efficient FL frameworks to mitigate the heightened resources induced by adaptive local optimizers in Section 3, which is $\texttt{FedAda}^2/\texttt{FedAda}^2$++. We begin by providing a definition of heavy-tailed noise following previous literature.

**Definition A.1.** A random variable $\xi \sim \mathcal{D}$ follows a **heavy-tailed** distribution if the $\alpha$-moment is infinite for $\alpha \geq 2$. In other words, we say that the stochastic gradient noise $g(x) - \nabla f(x)$ is heavy-tailed if $\mathbb{E}\left[\|g(x) - \nabla f(x)\|^\alpha\right]$ is bounded for $\alpha \in (0, 2)$ and unbounded for $\alpha \geq 2$, where $g(x)$ is the stochastic gradient under some model parameter $x$, and $\nabla f(x)$ the full gradient.

We may now present the following proposition.

**Proposition A.2.** *There exists a federated learning problem with heavy-tailed client-side gradient noise such that the following arguments hold:*

*(i) For vanilla FedAvg, given any client sampling strategy, if the probability $p_i^t$ of client $i$ with heavy-tailed gradient noise being sampled at communication round $t$ is non-zero, then $\mathbb{E}\|\nabla f(x_{t+1})\|^2 = \infty$ for any nontrivial learning rate schedule $\eta_\ell^t > 0$ and global parameter $x_{t+1}$.*

*(ii) Under an appropriate learning rate schedule, FedAvg with local adaptivity (i.e., via client-side AdaGrad) bounds the error in expectation as*

$$\lim_{t \to \infty} \mathbb{E}\|x_t - x^*\| \leq \frac{2\sqrt{3}}{1 - \hat{\varepsilon}} \quad \text{for some} \quad \hat{\varepsilon} \approx 0,$$

*where $x^*$ is the global optimum.*

A detailed proof is given by construction on a quadratic objective in Appendix A.1. We show that even a single client with heavy-tailed gradient noise is able to instantaneously propagate their volatility to the global model, which severely destabilizes distributed learning in expectation. Unfortunately, recent works have observed heavy-tailed gradient noise empirically, especially within model architectures utilizing attention mechanisms, including transformer-based models [40, 41, 42, 31, 43, 44, 45]. Proposition A.2 (ii) suggests that client-side adaptivity has the potential to stabilize local model updates pushed from diverse and large-scale distributed sources, if communication bottlenecks and memory efficiency can be addressed.

The construction of the federated problem in Proposition A.2 draws gradient noise from the Student $t$-distribution which is heavy-tailed depending on the parameter regime, whose moments are relatively controlled nevertheless. We may exacerbate the severity of gradient stochasticity by inserting a singular client with Cauchy-distributed noise, while enforcing all other clients to follow non-heavy-tailed Gaussian gradient noise. We further detail this setting in Proposition A.3, Appendix A.1.

## A.1   Constructions

**Overview of Student's *t*-distribution.**   For the convenience of the reader, we provide a brief summary of basic properties of the Student's $t$-distribution. Intuitively, the $t$-distribution can be understood as an approximation of the Gaussian with heavier tails. The density is given by

$$f_\nu(t) = \frac{\Gamma\left(\frac{\nu+1}{2}\right)}{\sqrt{\pi\nu}\Gamma\left(\frac{\nu}{2}\right)} \left(1 + \frac{t^2}{\nu}\right)^{-(\nu+1)/2}$$

where $\nu \in \mathbb{R}_{>0}$ is the degree of freedom (or normality parameter), and $\Gamma$ is the gamma function. We recover the normalized Gaussian as the degree of freedom tends to infinity. The first moment is $0$ for $\nu > 1$, and the second moment satisfies $\nu/(\nu-2)$ for $\nu > 2$ while being infinite for $1 < \nu \leq 2$, where the heavy-tails are most pronounced. Following the convention of [40], we refer to a distribution as being heavy-tailed if the second moment is infinite.

The following proposition showcases the utility of local adaptivity in federated learning.

**Proposition A.3.** *There exists a federated optimization problem with heavy-tailed client noise which satisfies the following under FedAvg (where appropriate learning rate schedules are chosen for (ii-iv)):*

*(i) Given any client sampling strategy, if the probability $p_i^t$ of client $i$ with heavy-tailed gradient noise being sampled at step $t$ is non-zero, then $\mathbb{E}\|\nabla f(x_{t+1})\|^2 = \infty$ for any nontrivial learning rate schedule $\eta_\ell^t > 0$.*

*(ii) Local adaptivity via client-side AdaGrad bounds the error in expectation as*

$$\lim_{t \to \infty} \mathbb{E}\|x_t - x^*\| \leq \frac{2\sqrt{3}}{1 - \hat{\varepsilon}} \quad \text{for some} \quad \hat{\varepsilon} \approx 0,$$

*where $x^*$ is the global optimum.*

*(iii) Furthermore, local adaptivity implicitly constructs a critical Lyapunov stable region which stabilizes the gradient variance via the following inequality which holds once any learned weight enters the region:*

$$\min_{t \in \{1, \dots, T\}} \mathbb{E}\|\nabla f(x_t)\|^2 \leq \mathcal{O}\left(\frac{1}{T}\right).$$

*(iv) The global gradient variance of the federated problem with heavy-tailed client noise is fully stabilized via*

$$\mathbb{E}[\|\nabla f(x_t)\|^2] \leq 2\|x_0\|^2 + 2\left(\int_1^\infty \frac{1}{x^2}\,\mathrm{d}x\right)^2 \quad \text{for} \quad \forall t \in \{1, \dots, T\}.$$

This proposition demonstrates that even a single client with heavy-tailed gradient noise is able to instantaneously propagate their volatility to the global model, which destabilizes federated training in expectation. However, recent work [40] has shown that heavy-tailed gradient distributions appear frequently in language model applications, and more generally within model architectures utilizing any kind of attention mechanism, including transformers. To our knowledge, this provable failure mode of distributed training resultant from the unbiased, yet heavy-tailed noise of a singular client has not previously been reported within the literature.

**Proof of (i).** Let the local stochastic objectives be given by $F_i(x, \xi_i) = x^2/2 + \xi_i x$ where gradient noise follows a $t$-distribution with $i + 1$ degrees of freedom, $\xi_i \sim t_{i+1}$ for $\forall i \in \{1, \dots, N\}$. This construction is chosen to materialize the setting in which only a singular client suffers from heavy-tailed noise ($i = 1$). Minibatches are sampled with replacement, which ensures that gradient noise in each client epoch are independent amongst and in between any two (possibly identical) clients, and further identically distributed conditional on the client ID $i$. Clearly, the global objective is

$$f(x) = \frac{1}{N} \sum_{i=1}^{N} \mathbb{E}_{\xi_i}\left[f_i(x, \xi_i)\right] = \frac{1}{N} \mathbb{E}\left[\frac{N}{2}x^2 + \sum_{i=1}^{N} \xi_i x\right] = \frac{1}{2}x^2.$$

For global step $t$, we subsample clients $\mathcal{S}^t$ following any sampling strategy, where $\mathcal{C}^t$ is the collection of all possible multisets $\mathcal{S}_r^t$ whose elements indicate (possibly repeated) client selection, with associated probabilities $p_C^t(r) > 0$ of realization for $r \in [|\mathcal{C}^t|]$. Assume that $1 \in \mathcal{S}_m^t$ for some $m$.

Then, FedAvg updates may be written

$$x_{t+1} = x_t - \frac{\eta_\ell}{|\mathcal{S}^t|} \sum_{i \in \mathcal{S}^t} \sum_{\ell=1}^{K} g_{i,\ell}^t$$

which gives the squared length of the global gradient under expectation as

$$\mathbb{E}_t\|\nabla f(x_{t+1})\|^2 = \mathbb{E}_t\left\|x_t - \frac{\eta_\ell}{|\mathcal{S}^t|}\sum_{i\in\mathcal{S}^t}\sum_{\ell=1}^{K}\left(\nabla f(x_{i,\ell-1}^t) + \xi_{i,\ell-1}^t\right)\right\|^2$$

$$= \mathbb{E}_{\xi|t}\mathbb{E}_{\mathcal{S}^t|\xi,t}\left\|x_t - \frac{\eta_\ell}{|\mathcal{S}^t|}\sum_{i\in\mathcal{S}^t}\sum_{\ell=1}^{K}\left(\nabla f(x_{i,\ell-1}^t) + \xi_{i,\ell-1}^t\right)\right\|^2$$

$$= \sum_{r=1}^{|\mathcal{C}^t|}\mathbb{E}_{\xi|t}p_C^t(r)\left\|x_t - \frac{\eta_\ell}{|\mathcal{S}_r^t|}\sum_{i\in\mathcal{S}_r^t}\sum_{\ell=1}^{K}\left(\nabla f(x_{i,\ell-1}^t) + \xi_{i,\ell-1}^t\right)\right\|^2$$

$$\geq p_C^t(m)\mathbb{E}_{\xi|t}\left\|x_t - \frac{\eta_\ell}{|\mathcal{S}_m^t|}\sum_{i\in\mathcal{S}_m^t}\sum_{\ell=1}^{K}\left(x_{i,\ell-1}^t + \xi_{i,\ell-1}^t\right)\right\|^2$$

where in the second equality we have conditioned on local gradient noise $\xi$ and stochastic realizations up to timestep $t$, using the law of iterated expectations. Recursively unravelling $x_{i,\ell-1}^t$ in terms of sampled noise and $x_{i,0}^t = x_t$ gives

$$x_{i,\ell-1}^t = x_{i,\ell-2}^t - \eta_\ell g_{i,\ell-2}^t = x_{i,0}^t - \eta_\ell\sum_{p=0}^{\ell-2}g_{i,p}^t$$

$$= x_{i,0}^t - \eta_\ell\left(\sum_{p=0}^{\ell-2}\nabla f(x_{i,p}^t) + \xi_{i,p}^t\right)$$

$$= x_{i,0}^t - \eta_\ell\left(\sum_{p=0}^{\ell-2}x_{i,p}^t + \xi_{i,p}^t\right)$$

$$= a_t x_t - \sum_{p=0}^{\ell-2}a_{i,p}^t\xi_{i,p}^t$$

where $a_t, a_{i,p}^t \in \mathbb{Q}[\eta_\ell]$ are polynomial functions of the learning rate with rational coefficients. Therefore, we have for $b_{i,p}^t \in \mathbb{Q}[\eta_\ell]$

$$p_C^t(m)\mathbb{E}_{\xi|t}\left\|x_t - \frac{\eta_\ell}{|\mathcal{S}_m^t|}\sum_{i\in\mathcal{S}_m^t}\sum_{\ell=1}^{K}\left(a_t x_t - \sum_{p=0}^{\ell-2}a_{i,p}^t\xi_{i,p}^t + \xi_{i,\ell-1}^t\right)\right\|^2$$

$$= p_C^t(m)\mathbb{E}_{\xi|t}\left\|\left(1 - \frac{\eta_\ell}{|\mathcal{S}_m^t|}\sum_{i\in\mathcal{S}_m^t}\sum_{\ell=1}^{K}a_t\right)x_t + \frac{\eta_\ell}{|\mathcal{S}_m^t|}\sum_{i\in\mathcal{S}_m^t}\sum_{\ell=1}^{K}\left(\sum_{p=0}^{\ell-2}a_{i,p}^t\xi_{i,p}^t + \xi_{i,\ell-1}^t\right)\right\|^2$$

$$= p_C^t(m)\mathbb{E}_{\xi|t}\left\|\left(1 - \frac{\eta_\ell}{|\mathcal{S}_m^t|}\sum_{i\in\mathcal{S}_m^t}\sum_{\ell=1}^{K}a_t\right)x_t\right\|^2 + \frac{\eta_\ell^2 p_C^t(m)}{|\mathcal{S}_m^t|^2}\mathbb{E}_{\xi|t}\left\|\sum_{i\in\mathcal{S}_m^t}\left(\sum_{p=0}^{K-2}b_{i,p}^t\xi_{i,p}^t + \xi_{i,K-1}^t\right)\right\|^2$$

$$\geq \frac{\eta_\ell^2 p_C^t(m)\mathbb{E}\left\|\xi_{1,K-1}^t\right\|^2}{|\mathcal{S}_m^t|^2} = \infty,$$

where we have used that $\xi_{i,\ell}^t \sim t_{i+1}$ independently with mean 0, for all permissible $i$, $\ell$, and $t$.

**Proof of (ii).** We show that under client-side AdaGrad with normalized gradients, the expected iterate norm converges to a noise-dependent constant at rate $\mathcal{O}(1/t)$, uniformly over client subsamples. The key idea is a decomposition of the update into a contraction term and a noise-dependent residual, showing the decay of the contraction under suitable $\varepsilon$ and learning rate schedules. We specialize to the setting with client-side AdaGrad with $K = 1$. Assume that clients $S^t$ have been selected to

participate in the round, which gives the update as

$$x_{t+1} = x_t - \frac{\eta_\ell}{|\mathcal{S}^t|} \sum_{i \in \mathcal{S}^t} \sum_{\ell=1}^{K} \frac{g_{i,\ell}^t}{\|g_{i,\ell}^t\| + \varepsilon} \tag{2}$$

$$= x_t - \frac{\eta_\ell}{|\mathcal{S}^t|} \sum_{i \in \mathcal{S}^t} \frac{\nabla f(x_{i,0}^t) + \xi_{i,1}^t}{\|\nabla f(x_{i,0}^t) + \xi_{i,1}^t\| + \varepsilon}$$

$$= x_t \left( 1 - \frac{\eta_\ell}{|\mathcal{S}^t|} \sum_{i \in \mathcal{S}^t} \frac{1}{\|x_t + \xi_i\| + \varepsilon} \right) - \frac{\eta_\ell}{|\mathcal{S}^t|} \sum_{i \in \mathcal{S}^t} \frac{\xi_i}{\|x_t + \xi_i\| + \varepsilon}$$

where we have gradually simplified notation. Noting that

$$\int \frac{1}{\|x_t + \xi_i\| + \varepsilon} \, p(\xi_i) \, \mathrm{d}\xi_i \leq \frac{1}{\varepsilon},$$

setting $\eta_\ell \leq \varepsilon$ gives

$$\|\nabla f(x_{t+1})\| = \|x_{t+1}\| \leq \|x_t\| \cdot \left( 1 - \frac{\eta_\ell}{|\mathcal{S}^t|} \sum_{i \in \mathcal{S}^t} \frac{1}{\|x_t + \xi_i\| + \varepsilon} \right) + \frac{\eta_\ell}{|\mathcal{S}^t|} \sum_{i \in \mathcal{S}^t} \frac{\|\xi_i\|}{\|x_t + \xi_i\| + \varepsilon}. \tag{3}$$

Using $\mathbb{E}_t$ to denote expectation conditional over realizations up to step $t$, we have

$$\mathbb{E}_t \|x_{t+1}\| \leq \|x_t\| \cdot \left( 1 - \frac{\eta_\ell}{|\mathcal{S}^t|} \mathbb{E}_t \left[ \sum_{i \in \mathcal{S}^t} \frac{1}{\|x_t + \xi_i\| + \varepsilon} \right] \right) + \frac{\eta_\ell}{|\mathcal{S}^t|} \sum_{i \in \mathcal{S}^t} \mathbb{E}_t \left[ \frac{\|\xi_i\|}{\|x_t + \xi_i\| + \varepsilon} \right].$$

To further bound the right hand side, consider the functional

$$I_i(\varepsilon) := \int \frac{1}{\|x_t + \xi_i\| + \varepsilon} \, p_{i+1}(\xi_i) \, \mathrm{d}\xi_i,$$

where clearly

$$I_i(0) \geq \int_{-x_t^-}^{-x_t^+} \frac{1}{\|x_t + \xi_i\|} \, p_{i+1}(\xi_i) \, \mathrm{d}\xi_i \approx \int_{0^-}^{0^+} \frac{p_{i+1}(-x_t)}{|x|} \, \mathrm{d}x = \infty$$

and $I_i(1) < 1$. By continuity and strict decay of $I_i(\varepsilon)$, there exists $1 \gg \hat{\varepsilon}_i > 0$ and $\varepsilon_i \in (0, 1]$ such that for all $i \in [N]$, we have $1 > I_i(\varepsilon) \geq 1 - \hat{\varepsilon}_i$ for $\varepsilon \in [\varepsilon_i, 1]$. Taking $\varepsilon \in [\max_{i \in [N]} \varepsilon_i, 1]$ and $\hat{\varepsilon} := \max_{i \in [N]} \hat{\varepsilon}_i$, we thus obtain

$$\mathbb{E}_t \|x_{t+1}\| \leq \|x_t\| \cdot (1 - \eta_\ell (1 - \hat{\varepsilon})) + \frac{\eta_\ell}{|\mathcal{S}^t|} \sum_{i \in \mathcal{S}^t} \mathbb{E}_t \left[ \frac{\|\xi_i\|}{\|x_t + \xi_i\| + \varepsilon} \right]. \tag{4}$$

To bound the remaining term, it is easy to show that $\|\xi_i\| p_{i+1}(\xi_i)$ is symmetric around the origin $O$, and strictly increases from 0 to $(3/2 + 2/(i+1))^{-1/2}$ while strictly decreasing afterwards. Defining the even extension of

$$h_{i+1}(\xi_i) = \begin{cases} -\frac{x}{(3/2 + 2/(i+1))^{-1/2}} + \sup_{\xi_i \in \mathbb{R}} \|\xi_i\| p_{i+1}(\xi_i) + \epsilon & \text{for } 0 \leq \xi_i \leq \left( \frac{3}{2} + \frac{2}{i+1} \right)^{-\frac{1}{2}}, \\ \|\xi_i\| p_{i+1}(\xi_i) & \text{for } \xi_i > \left( \frac{3}{2} + \frac{2}{i+1} \right)^{-\frac{1}{2}} \end{cases}$$

to be $h_{i+1}(\xi_i)$ for small $1 \gg \epsilon > 0$, we note that $1/(\|x_t + \xi_i\| + \varepsilon)$ analogously is symmetric around $\xi_i = -x_t$ while decaying with respect to the argument $\|x_t + \xi_i\|$. As $h_{i+1}(\xi_i)$ is symmetric around $O$ and decays moving to the left and right of $O$, by matching monotonicity and maxima with $1/(\|x_t + \xi_i\| + \varepsilon)$, we conclude that the left hand side of (5) is maximized for $x_t = 0$:

$$\mathbb{E}_t \left[ \frac{\|\xi_i\|}{\|x_t + \xi_i\| + \varepsilon} \right] \leq \int \frac{h_{i+1}(\xi_i)}{\|\xi_i\| + \varepsilon} \, \mathrm{d}\xi_i = B_i. \tag{5}$$

Asymptotically as $\xi_i \to \infty$, we have

$$\frac{h_{i+1}(\xi_i)}{\|\xi_i\| + \varepsilon} \lesssim p_{i+1}(\xi_i),$$

which gives that $B_i < \infty$. Letting $B := \max_{i \in [N]} B_i$ and scheduling the learning rate $\eta_\ell^t = 1/((t + t_0)(1 - \hat{\varepsilon}))$ where $t_0$ is the smallest positive integer satisfying $\eta_\ell^t < \varepsilon$ for all $t$, we thus conclude

$$\mathbb{E}\|x_{t+1}\| \leq \frac{t + t_0 - 1}{t + t_0}\mathbb{E}\|x_t\| + \frac{B}{(t + t_0)(1 - \hat{\varepsilon})}$$

$$\leq \frac{t + t_0 - 2}{t + t_0}\mathbb{E}\|x_{t-1}\| + \frac{2B}{(t + t_0)(1 - \hat{\varepsilon})}$$

$$\leq \cdots \leq \frac{t_0 - 1}{t + t_0}\mathbb{E}\|x_0\| + \frac{(t + 1)B}{(t + t_0)(1 - \hat{\varepsilon})}$$

$$\leq \mathcal{O}\left(\frac{1}{t}\right) + \frac{B}{1 - \hat{\varepsilon}}.$$

As this bound holds for any choice of client subsample $S^t$, we are done. It is easy to show by straightforward integration that $B < 2\sqrt{3}$.

**Proof of (iii).** Our strategy is to locate a 1-shot stabilization regime of the gradient norm that is formed via client adaptivity, which may be viewed as a Lyapunov stable region of the optimum $x^*$. From (3) and Jensen,

$$\|x_{t+1}\|^2 \leq 2\|x_t\|^2 \cdot \left(1 - \frac{\eta_\ell}{|S^t|}\sum_{i \in S^t}\frac{1}{\|x_t + \xi_i\| + \varepsilon}\right)^2 + \frac{2\eta_\ell^2}{|S^t|^2}\left(\sum_{i \in S^t}\frac{\|\xi_i\|}{\|x_t + \xi_i\| + \varepsilon}\right)^2$$

$$\leq 2\|x_t\|^2 \cdot \left(1 - \frac{\eta_\ell}{|S^t|}\sum_{i \in S^t}\frac{1}{\|x_t + \xi_i\| + \varepsilon}\right)^2 + \frac{2\eta_\ell^2}{|S^t|}\sum_{i \in S^t}\left(\frac{\|\xi_i\|}{\|x_t + \xi_i\| + \varepsilon}\right)^2.$$

We now impose $\eta_\ell \leq 2\varepsilon$, while letting $\|x_t\| < \delta$ for some $\delta \in \mathbb{R}_{>0}$. Taking expectations gives

$$\mathbb{E}_t\|x_{t+1}\|^2 \leq 2\|x_t\|^2 + \frac{2\eta_\ell^2}{|S^t|}\sum_{i \in S^t}\mathbb{E}_t\left(\frac{\|\xi_i\|}{\|x_t + \xi_i\| + \varepsilon}\right)^2,$$

and by similar arguments to the proof of **(ii)**, the summands of the second term are bounded uniformly by $\widetilde{B}$ which yields

$$\mathbb{E}\|x_{t+1}\|^2 \leq 2\delta^2 + 2\eta_\ell^2\widetilde{B}.$$

Setting $\delta, \eta_\ell^t \leq \mathcal{O}(1/\sqrt{T})$ immediately gives the desired inequality.

**Proof of (iv).** An advantage of client-side adaptive optimization is the autonomous normalization and clipping of the stochastic gradients. Let $\eta_\ell^t := 1/t^2$. Telescoping (2) gives

$$x_{T+1} = x_0 - \sum_{t=1}^{T}\frac{\eta_\ell^t}{|S^t|}\sum_{i \in S^t}\sum_{\ell=1}^{K}\frac{g_{i,\ell}^t}{\|g_{i,\ell}^t\| + \varepsilon},$$

which implies

$$\|x_{T+1} - x_0\| = \left\|\sum_{t=1}^{T}\frac{\eta_\ell^t}{|S^t|}\sum_{i \in S^t}\sum_{\ell=1}^{K}\frac{g_{i,\ell}^t}{\|g_{i,\ell}^t\| + \varepsilon}\right\|$$

$$\implies \|\|x_{T+1}\| - \|x_0\|\| \leq \left\|\sum_{t=1}^{T}\frac{\eta_\ell^t}{|S^t|}\sum_{i \in S^t}\sum_{\ell=1}^{K}\frac{g_{i,\ell}^t}{\|g_{i,\ell}^t\| + \varepsilon}\right\|$$

$$\implies \|x_{T+1}\| \leq \|x_0\| + \left\|\sum_{t=1}^{T}\frac{\eta_\ell^t}{|S^t|}\sum_{i \in S^t}\sum_{\ell=1}^{K}\frac{g_{i,\ell}^t}{\|g_{i,\ell}^t\| + \varepsilon}\right\|$$

$$\implies \mathbb{E}\|x_{T+1}\|^2 \leq 2\|x_0\|^2 + 2\mathbb{E}\left\|\sum_{t=1}^{T}\frac{\eta_\ell^t}{|S^t|}\sum_{i \in S^t}\sum_{\ell=1}^{K}\frac{g_{i,\ell}^t}{\|g_{i,\ell}^t\| + \varepsilon}\right\|^2.$$

Substituting the learning rate schedule $\eta_\ell^t := 1/t^2$ above gives

$$\mathbb{E}\left\|\sum_{t=1}^T \frac{\eta_\ell^t}{|\mathcal{S}^t|}\sum_{i\in\mathcal{S}^t}\sum_{\ell=1}^K \frac{g_{i,\ell}^t}{\|g_{i,\ell}^t\|+\varepsilon}\right\|^2 \leq \mathbb{E}\left\|\sum_{t=1}^T K\eta_\ell^t\right\|^2$$

$$\leq \mathbb{E}\left\|K\int_1^\infty \frac{1}{x^2}\,\mathrm{d}x\right\|^2.$$

Therefore, we conclude that for any $t$,

$$\mathbb{E}\|x_t\|^2 \leq 2\|x_0\|^2 + 2K^2\left(\int_1^\infty \frac{1}{x^2}\,\mathrm{d}x\right)^2.$$

## A.2  Deep Remorse of FedAvg and SGD

So far, we have examined toy problems in which heavy-tailed gradient noise is guaranteed to destabilize distributed training in expectation. We now prove that this is an instantiation of a more general phenomenon in federated learning where a family of online $\mu$-strongly convex global objectives collapses to the identical failure mode. To our knowledge, this provable limitation of distributed training resultant from the heavy-tailed noise of a singular client has not previously been established within the literature. The proofs of all results are given in the appendix.

**Definition A.4.** A learning algorithm $\mathcal{A}$ is **deeply remorseful** if it incurs infinite or undefined regret in expectation. If $\mathcal{A}$ is guaranteed to instantly incur such regret due to sampling even a single client with a heavy-tailed gradient noise distribution, then we say $\mathcal{A}$ is **resentful** of heavy-tailed noise.

**Theorem A.5.** *Let the global objectives $f_t(x)$ of a distributed training problem satisfy $\mu$-strong convexity for $t = 1,\ldots,T$. Assume that the participation probability of a client with a heavy-tailed stochastic gradient noise distribution is non-zero. Then, FedAvg becomes a deeply remorseful algorithm and is resentful of heavy-tailed noise. Furthermore, if the probability of the heavy-tailed client being sampled at step $t$ is nontrivial, then the variance of the global objective at $t + 1$ satisfies $\mathbb{E}\|f_{t+1}(x_{t+1})\|^2 = \infty$.*

In federated learning, we typically have $f_t(x) \equiv f(x)$ for all $t = 1,\ldots,T$ (i.e., the objective functions are the same across all rounds). Proposition A.2 intuits that inserting local adaptivity successfully breaks the generality of remorse and heavy-tailed resent for FedAvg. A high-level overview is that client-side AdaGrad clips the local updates of each iteration, which mollifies the impact of stochasticity in perturbing the weight updates. This gives Proposition A.6, which is formulated loosely without utilizing any advantages provided by local adaptivity except for clipping. Given that adaptive methods inherently include an implicit soft clipping mechanism due to the effects of preconditioning, we consider them to be preferable to clipped SGD for large-scale applications as they also offer the benefits of adaptivity. This preference holds, provided that the memory and computational constraints of the clients can be adequately managed.

**Proposition A.6.** *Introducing client-side adaptivity via AdaGrad for the setting in Theorem A.5 produces a non-remorseful and a non-resentful algorithm.*

The benefits of client-side adaptivity have also been shown in previous works (e.g., [46, 5]). We note that Proposition A.6 can be straightforwardly extended to jointly adaptive methods as well as for $f_t \in C(\mathbb{R}^d)$ not necessarily convex. An advantage of federated learning is that when done tactfully, the large supply of clients enable the trainer to draw from a virtually unlimited stream of computational power. The downside is that the global model may be strongly influenced by the various gradient distributions induced by the private client data shards. In this paper, we focus specifically on joint adaptive optimization as a countermeasure to stabilize learning. For this reason, we propose FedAda$^2$/FedAda$^2$++ in Section 3, which utilizes joint adaptivity in an efficient and scalable manner for distributed or federated training.

## B  Additional Related Works

Due to space restrictions, we include a brief additional discussion of some recent related works here. Mukherjee [47] propose FedSPS, a locally adaptive method using stochastic Polyak step sizes to

dynamically scale updates per client, achieving theoretical convergence under heterogeneous data. Li et al. [13] develop FedDA, a restarted dual averaging framework in which clients receive adaptive preconditioners and gradient estimates from the server and return dual states, enabling matching to standard non-adaptive gradient complexities for non-convex optimization. While these methods enhance local adaptivity, they often involve costly communication of dual states [13] or are not jointly adaptive [47]. By contrast, our approach maintains minimal communication and state, enabling efficient deployment while retaining joint adaptivity.

**Comparing with Wang et al. [5].** We note that Wang et al. [5] employs a similar strategy of initializing local preconditioners to zero. However, this work does not provide convergence guarantees for this efficient algorithm with vanishing errors, and it does not systematically evaluate it empirically. Instead, it proposes another adaptive optimization framework with extra communication overheads. Although the theoretical results establish upper bounds, crucially, in their Theorem 1, the terms involving $h_i$, $q_i$ cannot be computed in closed form for adaptive optimizers (only estimated numerically), limiting interpretability and practical applicability. To the best of our knowledge, our work is the first to provide rigorous convergence guarantees for FedAda$^2$/FedAda$^2$++ without leaving any residual terms that are not analytically characterized.

## C   Detailed FedAda$^2$ Algorithm Description

In the main text, we have opted to describe the intuitions behind SM3, due to its technical implementation. In this appendix section, we give a more through walk-through of our algorithm details for any interested readers wishing to reproduce our proof strategies or implementations.

---

**Algorithm 2** Adaptive server and client-side ADAGRAD with SM3 (FedAda$^2$++)

---

**Require:** A full cover $\{S_1, \ldots, S_q\} \subset \mathcal{P}([d])$ where $\bigcup_{b=1}^q S_b = \{1, \ldots, d\}$
    Update delay step size $z \in \mathbb{Z}_{\geq 1}$, initializations $x_0, \widetilde{v}_0 \geq \tau^2$ and $\widetilde{m}_0 \leftarrow 0$
    Local epsilon smoothing terms $\varepsilon_s, \varepsilon > 0$, global smoothing term $\tau > 0$
    Global decay parameter $\widetilde{\beta}_1 \in [0, 1)$
1: **for** $t = 1, \ldots, T$ **do**
2:     Sample subset $\mathcal{S}^t \subset [N]$ of clients using any sampling scheme
3:     **for** each client $i \in \mathcal{S}^t$ (in parallel) **do**
4:         Initialize $v_0 \geq 0$ (default value $v_0 \leftarrow 0$), $x_{i,0}^t \leftarrow x_{t-1}$
5:         **for** $k = 1, \ldots, K$ **do**
6:             Draw stochastic gradient $g_{i,k}^t \sim \mathcal{D}_{i,\mathrm{grad}}(x_{i,k-1}^t)$ with mean $\nabla F_i(x_{i,k-1}^t) \in \mathbb{R}^d$
7:             $m_k \leftarrow g_{i,k}^t$, $\mu_k(b) \leftarrow 0$ for $\forall b \in \{1, \ldots, q\}$
8:             **for** $j = 1, \ldots, d$ **do**
9:                 Approximate Preconditioner (SM3)
10:             **end for**
11:             **if** $0 < \|m_k/(\sqrt{v_k} + \varepsilon)\| < \varepsilon_s$, **do** $m_k \leftarrow 0$
12:             $x_{i,k}^t \leftarrow x_{i,k-1}^t - \eta_\ell \cdot m_k/(\sqrt{v_k} + \varepsilon)$
13:         **end for**
14:         $\Delta_i^t = x_{i,K}^t - x_{t-1}$
15:     **end for**
16:     Server Update (SU)
17: **end for**

---

**Addressing Client-Side Resource Constraints.** In this paper, we specifically focus on SM3 [24] adaptations of Adam and Adagrad. Intuitively, SM3 exploits natural activation patterns observed in model gradients to accumulate approximate parameter-wise statistics for preconditioning. More precisely, the gradient information in each coordinate element $\{1, \ldots, d\}$ is blanketed by a cover $\{S_1, \ldots, S_q\}$ satisfying $\bigcup_{b=1}^q S_b = \{1, \ldots, d\}$ for which an auxiliary $\mu_k(b)$ is assigned for each $b \in [q]$. The $\mu_k(b)$ then act to form $v_k$ as a coordinate ascent upper bound to the squared gradient sum $\sum_{\ell=1}^k (g_{i,\ell}^t)^2$ as SM3 iterates over each $j \in [d]$.

As an optional add-on, utilizing the staleness of gradients to construct preconditioners has previously been suggested as a strategy to accelerate adaptive optimization without hurting the performance [22,

48]. Therefore, we may optionally further mollify the burden of client-side adaptive optimizers by enforcing delayed preconditioner updates (Appendix J.3). This is given by the following SM3 update rule (SM3) which incorporates delay step $z$,

$$\text{SM3 Update:} \begin{cases} v_k(j) \leftarrow \min_{b:S_b \ni j} \mu_{k-1}(b) + \left(g_{i,k}^t(j)\right)^2 & \text{for } \frac{k-1}{z} \in \mathbb{Z} \\ \mu_k(b) \leftarrow \max\{\mu_k(b), v_k(j)\}, \text{for } \forall b: S_b \ni j \\ v_k(j) \leftarrow v_{k-1}(j) & \text{otherwise} \end{cases} \quad \text{(SM3)}$$

where $k$ is the index of local iteration (starting from 1). These methodologies are consolidated into $\texttt{FedAda}^2$, Algorithm 2. For simplicity, we describe the variant in which both the client and server employ AdaGrad as the adaptive optimizers. However, we present other instantiations of $\texttt{FedAda}^2$ with different adaptive methods in Appendix E and J.2.

We now present a description of SM3-I/II with delayed preconditioner updates as Algorithms 3 and 4. SM3-II capitalizes on a tighter approximation of the second moment, and empirically demonstrates better results. We have opted to implement a smoothing term $\varepsilon$ instead of treating any zero denominator as zero as done in the original work. In this paper, we provide the analysis for SM3-II which generalizes the analysis for SM3-I.

---

**Algorithm 3** Delayed preconditioner SM3-I

**Require:** Client learning rate $\eta_\ell$, step delay $z \in \mathbb{Z}_{\geq 1}$, and $\varepsilon$-smoothing term $\varepsilon > 0$
**Require:** A full cover $\{S_1, \ldots, S_k\} \subset \mathcal{P}([d])$ where $\bigcup_{\ell=1}^k S_\ell = \{1, \ldots, d\}$
1: **Initialize:** $x_1 = 0$ and $\mu_0(r) = 0$ for $\forall r \in \{1, \ldots, k\}$
2: **for** $t = 1, \ldots, K$ **do**
3:    $g_t \leftarrow \nabla \ell(x_t)$
4:    **if** $(t-1)/z \in \mathbb{Z}$ **then**
5:       **for** $r = 1, \ldots, k$ **do**
6:          $\mu_t(r) \leftarrow \mu_{t-1}(r) + \max_{j \in S_r} g_t^2(j)$
7:       **end for**
8:    **end if**
9:    **for** $j = 1, \ldots, d$ **do**
10:       $\nu_t(j) \leftarrow \min_{r:S_r \ni j} \mu_t(r)$ (minimum taken over all $r$ such that $j \in S_r$)
11:       $x_{t+1}(j) \leftarrow x_t(j) - \frac{\eta_\ell g_t(j)}{\sqrt{\nu_t(j)} + \varepsilon}$
12:    **end for**
13: **end for**

---

# D   Detailed Proofs

To enhance clarity, we present several lemmas before giving the proof of Theorem D.6. Note that Lemma D.1 is written in broadcasting notation, where the scalars in the right hand side have $\mathbf{1} \in \mathbb{R}^d$ implicitly multiplied and the inequality holds coordinatewise. For notational convenience, we will view $\Phi_1^K$, $\Phi_2^K$ as vectors.

**Lemma D.1.** *Under Algorithm 2, $|\Delta_i^t|$ is bounded by*

$$|\Delta_i^t| \leq \Phi_1^K := \eta_\ell \left( \sqrt{\left\lceil \frac{K}{z} \right\rceil} \cdot \log^{\frac{1}{2}} \left( 1 + \frac{\left\lceil \frac{K}{z} \right\rceil G^2}{\varepsilon^2} \right) + \frac{\eta_\ell (K - \left\lceil \frac{K}{z} \right\rceil) G}{\sqrt{v_0} + \varepsilon} \right).$$

*Proof.* Forming a bound for the pseudogradients is not trivial due to delayed preconditioner updates. We begin by noting that delayed gradient updates are initiated at local timesteps $k = nz + 1$ for $n \in \mathbb{Z}_{\geq 0}$. We now split cases $k/z \notin \mathbb{Z}$ and $k/z \in \mathbb{Z}$. In the first case, there exists $n \in \mathbb{Z}_{\geq 0}$ such that $nz + 1 \leq k < (n+1)z$, and the latest preconditioner update by client step $k$ is given at timestep $(\lceil k/z \rceil - 1)z + 1 = \lfloor k/z \rfloor z + 1$. In the second case, if $z \neq 1$, then step $k$ is just one step shy of a preconditioner update. The latest update is therefore held at step $(\lceil k/z \rceil - 1)z + 1$ which is no longer identical to $\lfloor k/z \rfloor z + 1$.

---

**Algorithm 4** Delayed preconditioner SM3-II

---

**Require:** Client learning rate $\eta_\ell$, step delay $z \in \mathbb{Z}_{\geq 1}$, and $\varepsilon$-smoothing term $\varepsilon > 0$
**Require:** A full cover $\{S_1, \ldots, S_k\} \subset \mathcal{P}([d])$ where $\bigcup_{\ell=1}^{k} S_\ell = \{1, \ldots, d\}$
1: **Initialize:** $x_1 = 0$ and $\mu_0'(r) = 0$ for $\forall r \in \{1, \ldots, k\}$
2: **for** $t = 1, \ldots, K$ **do**
3:   $g_t \leftarrow \nabla \ell(x_t)$
4:   $\mu_t'(r) \leftarrow 0$ for $\forall r \in [k]$
5:   **for** $j = 1, \ldots, d$ **do**
6:     **if** $(t-1)/z \in \mathbb{Z}$ **then**
7:       $\nu_t'(j) \leftarrow \min_{r:S_r \ni j} \mu_{t-1}'(r) + g_t^2(j)$
8:       **for all** $r : S_r \ni j$ **do**
9:         set $\mu_t'(r) \leftarrow \max\{\mu_t'(r), \nu_t'(j)\}$
10:      **end for**
11:    **else**
12:      $\nu_t'(j) \leftarrow \nu_{t-1}'(j)$
13:    **end if**
14:    $x_{t+1}(j) \leftarrow x_t(j) - \frac{\eta_\ell g_t(j)}{\sqrt{\nu_t'(j) + \varepsilon}}$
15:  **end for**
16: **end for**

---

With this observation, it is easy to show by induction that

$$v_k(j) \geq v_0(j) + \sum_{\ell=1}^{\lceil \frac{k}{z} \rceil} \left( g_{i,(\ell-1)z+1}^t(j) \right)^2 \quad \text{for} \quad j \in \{1, \ldots, d\} \quad \text{and} \quad k \in \{1, \ldots, K\}.$$

Recall that $\Delta_t = 1/|\mathcal{S}^t| \sum_{i \in \mathcal{S}^t} \Delta_i^t$ and $\Delta_i^t = x_{i,K}^t - x_{i,0}^t$. By telescoping for $K$ local steps and the definition of gradient updates in AdaSquare-SM3, we obtain

$$|\Delta_i^t| = \left| \sum_{p=1}^{K} \eta_\ell \frac{m_p}{\sqrt{v_p} + \varepsilon} \right| \leq \eta_\ell \sum_{p=1}^{K} \frac{|g_{i,p}^t|}{\sqrt{v_0 + \sum_{r=1}^{\lceil \frac{p}{z} \rceil}(g_{i,(r-1)z+1}^t)^2} + \varepsilon}$$

For $\mathcal{F} = \{0, 1, \ldots, \lceil K/z \rceil - 1\}z + 1$, we thus have that

$$|\Delta_i^t| \leq \eta_\ell \sum_{p \in \mathcal{F}} \frac{|g_{i,p}^t|}{\sqrt{v_0 + \sum_{r=1}^{\lceil \frac{p}{z} \rceil}(g_{i,(r-1)z+1}^t)^2} + \varepsilon}$$
$$+ \eta_\ell \sum_{p \in [K] \setminus \mathcal{F}} \frac{|g_{i,p}^t|}{\sqrt{v_0 + \sum_{r=1}^{\lceil \frac{p}{z} \rceil}(g_{i,(r-1)z+1}^t)^2} + \varepsilon}.$$

To obtain a deterministic bound, we cannot ignore the worst-case stochastic realization that $g_{i,(r-1)z+1}^t = 0$ for $\forall r \in [\lceil \frac{p}{z} \rceil]$, $p \in [K] \setminus \mathcal{F}$. Therefore, we form the upper bound (where $\sum_1^0 := 0$ by definition)

$$|\Delta_i^t| \leq \eta_\ell \underbrace{\sum_{p \in \mathcal{F}} \frac{|g_{i,p}^t|}{\sqrt{v_0 + |g_{i,p}^t|^2 + \sum_{r=1}^{\lceil \frac{p}{z} \rceil - 1}(g_{i,(r-1)z+1}^t)^2} + \varepsilon}}_{T_1} + \frac{\eta_\ell}{\sqrt{v_0} + \varepsilon} \left( \sum_{p \in [K] \setminus \mathcal{F}} |g_{i,p}^t| \right) \quad (6)$$

$$\leq \eta_\ell T_1 + \frac{\eta_\ell (K - \lceil \frac{K}{z} \rceil) G}{\sqrt{v_0} + \varepsilon}.$$

As 0 is trivially bounded by any non-negative upper bound, we may without loss of generality assume that $g_{i,(r-1)z+1}^t \neq 0$ for at least one $r \in [\lceil \frac{p}{z} \rceil]$. We further bound $T_1$ as follows:

$$T_1 \leq \sum_{p \in \mathcal{F}} \frac{|g_{i,p}^t|}{\sqrt{|g_{i,p}^t|^2 + \sum_{r=1}^{\lceil \frac{p}{z} \rceil - 1}(g_{i,(r-1)z+1}^t)^2 + \varepsilon}} \leq \sum_{p \in \mathcal{F}} \sqrt{\frac{|g_{i,p}^t|^2}{\varepsilon^2 + \sum_{r \in [p] \cap \mathcal{F}} |g_{i,r}^t|^2}}$$

$$\leq \sqrt{|\mathcal{F}|} \sqrt{\left( \sum_{p \in \mathcal{F}} \frac{|g_{i,p}^t|^2}{\varepsilon^2 + \sum_{r \in [p] \cap \mathcal{F}} |g_{i,r}^t|^2} \right)}$$

$$\leq \sqrt{\left\lceil \frac{K}{z} \right\rceil} \cdot \log^{\frac{1}{2}} \left( 1 + \sum_{p \in \mathcal{F}} \frac{|g_{i,p}^t|^2}{\varepsilon^2} \right)$$

Note the use of Cauchy Schwartz in the third inequality. A detailed proof of the log inequality used in the third line may be found as part of the proof of Theorem D.6, equation (11) which uses similar techniques. By Assumption 2, we are done. $\qquad \square$

The server-side pseudogradient updates may also be bounded as follows.

**Lemma D.2.** *Under Algorithm 2, each server step size is bounded in absolute value by*

$$\Phi_2^K := \min \left\{ \eta \sqrt{(1 - \widetilde{\beta}_1)(1 - \widetilde{\beta}_1^{2t})}, \frac{\eta}{\tau} \Phi_1^K \right\}.$$

*Proof.* Without loss of generality, we may let $\tau = 0$ when forming the first upper bound for expository purposes.

$$\eta \frac{|\widetilde{m}_t|}{\sqrt{\widetilde{v}_t} + \tau} \leq \frac{\eta(1 - \widetilde{\beta}_1) \sum_{\ell=1}^t \widetilde{\beta}_1^{t-\ell} |\Delta_\ell|}{\sqrt{\sum_{\ell=1}^t \Delta_\ell^2 + \tau^2} + \tau}$$

$$\leq \frac{\eta(1 - \widetilde{\beta}_1) \left( \sum_{\ell=1}^t \widetilde{\beta}_1^{t-\ell} |\Delta_\ell| \right) \sqrt{\sum_{\ell=1}^t \widetilde{\beta}_1^{2t-2\ell}}}{\sqrt{\sum_{\ell=1}^t \Delta_\ell^2} \sqrt{\sum_{\ell=1}^t \widetilde{\beta}_1^{2t-2\ell}}}$$

$$\leq \eta \sqrt{1 - \widetilde{\beta}_1} \sqrt{1 - \widetilde{\beta}_1^2} \sqrt{\sum_{\ell=1}^t \widetilde{\beta}_1^{2t-2\ell}}$$

$$= \eta \sqrt{1 - \widetilde{\beta}_1} \sqrt{1 - \widetilde{\beta}_1^{2t}}.$$

Note that the final inequality is obtained using Cauchy-Schwartz, while the second bound in the lemma statement follows from the first inequality and Lemma D.1. $\qquad \square$

Finally, we form a loose upper bound for the gradient variance.

**Lemma D.3.** *For $k \in \{1, \ldots, K\}$, the uncentered variance estimate $v_k$ as well as $\mu_k$ in Algorithm 2 are bounded by*

$$(B1): \quad 0 \leq \mu_k(b) \leq dkG^2 \quad for \quad and \quad b \in \{1, \ldots, q\},$$
$$(B2): \quad 0 \leq v_k(j) \leq dkG^2 \quad for \quad j \in \{1, \ldots, d\}.$$

*Proof.* Non-negativity of the variance estimates $v_k$ is trivial and implies the non-negativity of $\mu_k$, thus we focus on the upper bound for which we use dual induction. The case $k = 1$ is satisfied by zero initialization. Assuming the inequality holds for $k \leftarrow k - 1$, we have for each $j$

$$v_k(j) = \min_{b: S_b \ni j} \mu_{k-1}(b) + \left( g_{i,k}^t(j) \right)^2 \leq d(k-1)G^2 + G^2 \leq dkG^2.$$

Now, $\mu_k$ is initialized to zero at the start of each step $k$ and its entries are increased while broadcasting over each coordinate $j \in \{1, \ldots, d\}$ by

$$\mu_k(b) \leftarrow \max\{\mu_k(b), v_k(j)\} \quad for \quad \forall b : j \in S_b.$$

For $j = 1$, it is clear that

$$\mu_k(b) \leftarrow v_k(j) \leq dkG^2 \quad \text{for} \quad \forall b \in \{1, \ldots, q\}.$$

For $j \geq 2$, inductively, we have

$$\mu_k(b) \leftarrow \max\{\mu_k(b), v_k(j)\} \leq dkG^2$$

as both arguments of the maximum function are upper bounded by $dkG^2$. This completes the proof. $\qquad \square$

## D.1 Precompact Convergence Analysis

We aim to analyze the convergence of learning algorithms under the general, non-convex setting. However, extremely popular and well known adaptive optimizers such as Adam whose efficacy is strongly supported by empirical evidence have been shown to fail to converge even for convex settings [11]. Therefore, recent works have investigated the asymptotic stabilization of gradients, instead of requiring strict convergence to local or global optima of the objective [4, 8, 12, 14, 15, 16, 40]. Such convergence bounds are of the form $\min_t \|\nabla f(x_t)\| \leq \mathcal{O}(T^{-\alpha})$, and are interpreted via the following lemma:

**Lemma D.4.** *For $x_t$ the $t$-step parameters of any objective $f(x)$ learned by an algorithm, let $\min_{1 \leq t \leq T} \|\nabla f(x_t)\| \leq \mathcal{O}(T^{-\alpha})$ for $\alpha > 0$. Then, there exists a learning algorithm which outputs parameters $\{\widetilde{x}_1, \widetilde{x}_2, \ldots\}$ such that $\|\nabla f(\tilde{x}_t)\| \to 0$ as $t \to \infty$.*

*Proof.* Assuming otherwise gives that $\|\nabla f(x_t)\|$ is $\varepsilon$-bounded away from 0 for some $\varepsilon > 0$, for any parameter $x_t$ realized by the algorithm. Clearly, $\min_{1 \leq t \leq T} \|\nabla F(x_t)\| \to 0$ as $T \to \infty$ gives a contradiction. More constructively, note that $\forall \varepsilon > 0$, $\exists \widetilde{T}(\varepsilon) \in \mathbb{N}$ such that $T \geq \widetilde{T}(\varepsilon) \implies \min_{1 \leq t \leq T} \|\nabla f(x_t)\| < \varepsilon$. Letting $\varepsilon = 1/n$ for $n \in \mathbb{N}$ and $T_n := \widetilde{T}(1/n)$, we have that there exists $t_n \in [T_n]$ such that $\|\nabla f(x_{t_n})\| < 1/n$. Letting $\widetilde{x}_i := x_{t_i}$ extracts the desired parameter sequence. $\qquad \square$

This notion of convergence can be formalized as *precompact convergence* which is consistent with sequence properties of precompact normed sets. In this paper, we explicitly formalize the conventions used in prior works, and take the term convergence to mean precompact convergence unless stated otherwise.

**Definition D.5** (Precompact convergence). A sequence $\{y_n\}_{n \in \mathbb{N}}$ in a normed space $\mathcal{Y}$ is said to converge precompactly to $y \in \mathcal{Y}$ if there exists $\varphi : \mathbb{N} \to \mathbb{N}$ such that $y_{\varphi(n)} \to y$.

Our goal is to develop principled federated algorithms whose global gradients are guaranteed to converge precompactly to 0 regardless of parameter initialization, in the general, non-convex setting. Note that precompact convergence must allow for convergence to each element $y_n$ of the sequence. Now, we are ready to present Theorem D.6.

We note that prior work, such as FedNAR [49] and FedOPT [4], primarily analyzes either server-side or client-side adaptivity in isolation. In contrast, our algorithms incorporate joint adaptivity–combining client- and server-level adaptive updates–even without explicit preconditioner transmission, necessitating a distinct convergence analysis. Intuitively, the proof relies on the $L$-smoothness property and a careful decomposition of the inner product between the gradient and the adaptive update. A key challenge is handling the time-varying preconditioner $(\sqrt{\widetilde{v}_t})$, which is addressed by splitting the analysis into a term capturing the change in the preconditioner and a main descent term. The descent term is analyzed using auxiliary variables $\gamma_r, \alpha_r$ and Young's inequality, which isolates the sufficient descent (negative definite term) from the client drift error (discrepancy between global and local gradients). After telescoping the inequalities over $T$ iterations, the proof employs specialized techniques to bound the resulting summations. For instance, an inductive argument (Lemma D.8) yields a logarithmic bound for errors related to preconditioner updates, while the accumulated client drift is rigorously controlled by demonstrating that the exponential decay of the momentum parameter dominates the polynomial growth of the drift (Lemma D.7). Rearranging these bounds provides the final convergence rate.

**Theorem D.6.** *In Algorithm 2, we have that*

$$\min_{t\in[T]}\|\nabla f(x_{t-1})\|^2 \le \frac{\Psi_1 + \Psi_2 + \Psi_3 + \Psi_4 + \Psi_5}{\Psi_6},$$

*where*

$$\Psi_1 = f(x_0) - f(x^*),$$

$$\Psi_2 = \frac{\eta^2 LTd\|\Phi_1^K\|^2}{\tau^2},$$

$$\Psi_3 = \frac{(1 - \widetilde{\beta}_1^T)\eta\eta_\ell K\widetilde{L}T\|\Phi_1^K\|^2}{\widetilde{\alpha}_1\tau(\sqrt{v_0} + \varepsilon)^2},$$

$$\Psi_4 = \frac{(1 - \widetilde{\beta}_1)\eta\eta_\ell KLTc(\widetilde{\beta}_1)\|\Phi_2^K\|^2}{\widetilde{\alpha}_1\tau(\sqrt{v_0} + \varepsilon)^2},$$

$$\Psi_5 = \frac{\eta d\|\Phi_1^K\|G\left(1 - \widetilde{\beta}_1 + \log\left(1 + \frac{T\|\Phi_1^K\|^2}{\tau^2}\right)\right)}{\tau},$$

$$\Psi_6 = \frac{3(1 - \widetilde{\beta}_1)\eta\widetilde{\gamma}_1 T}{4\left(\sqrt{T\|\Phi_1^K\|^2 + \widetilde{v}_0} + \tau\right)}.$$

*Here, the constant $c$ is defined with respect to $\widetilde{\beta}_1$ as*

$$c(\widetilde{\beta}_1) := \sum_{u=0}^{\widetilde{u}_0(\widetilde{\beta}_1)} \widetilde{\beta}_1^u u^2 + \int_{\widetilde{u}_0(\widetilde{\beta}_1)}^{\infty} \frac{1}{x^2}\mathrm{d}x \quad \text{for} \quad \widetilde{u}_0(\widetilde{\beta}_1) = \inf\{u \in \mathbb{N} : \widetilde{\beta}_1^v v^2 < \frac{1}{v^2} \text{ for } \forall v \ge u\}$$

*and the intermediary $\widetilde{\gamma}_1, \widetilde{\alpha}_1$ values are defined as*

$$\widetilde{\gamma}_1 := \eta_\ell \frac{K}{\sqrt{v_0 + dKG^2} + \varepsilon}, \quad \widetilde{\alpha}_1 := \frac{1}{2\sqrt{v_0 + dKG^2} + 2\varepsilon}.$$

*Proof.* To enhance readability, we use both coordinatewise and broadcasting notation, where a $[\cdot]_j$ subscript is attached for the $j$-th coordinate. In particular, the arguments are detailed mostly in the latter notation as it significantly clarifies the intuitions behind the proof. By $L$-smoothness, we have

$$f(x_t) \le f(x_{t-1}) + \langle \nabla f(x_{t-1}), x_t - x_{t-1}\rangle + \frac{L}{2}\|x_t - x_{t-1}\|^2$$

$$= f(x_{t-1}) + \eta\left\langle\nabla f(x_{t-1}), \frac{\widetilde{\beta}_1^t\widetilde{m}_0 + (1 - \widetilde{\beta}_1)\sum_{r=1}^t \widetilde{\beta}_1^{t-r}\Delta_r}{\sqrt{\widetilde{v}_t} + \tau}\right\rangle + \frac{\eta^2 L}{2}\left\|\frac{\widetilde{\beta}_1^t\widetilde{m}_0 + (1 - \widetilde{\beta}_1)\sum_{r=1}^t \widetilde{\beta}_1^{t-r}\Delta_r}{\sqrt{\widetilde{v}_t} + \tau}\right\|^2$$

$$= f(x_{t-1}) + \eta T_{0,0} + (1 - \widetilde{\beta}_1)\eta\sum_{r=1}^t T_{0,r} + \frac{\eta^2 L}{2}\left\|\frac{\widetilde{\beta}_1^t\widetilde{m}_0 + (1 - \widetilde{\beta}_1)\sum_{r=1}^t \widetilde{\beta}_1^{t-r}\Delta_r}{\sqrt{\widetilde{v}_t} + \tau}\right\|^2 \quad (7)$$

where for $r \in [t]$,

$$T_{0,r} = \widetilde{\beta}_1^{t-r}\left\langle\nabla f(x_{t-1}), \frac{\Delta_r}{\sqrt{\widetilde{v}_t} + \tau}\right\rangle \quad \text{and} \quad T_{0,0} = \left\langle\nabla f(x_{t-1}), \frac{\widetilde{\beta}_1^t\widetilde{m}_0}{\sqrt{\widetilde{v}_t} + \tau}\right\rangle. \quad (8)$$

Note that $T_{0,0}$ can only decay exponentially as training progresses, as $\sqrt{\widetilde{v}_t}$ is monotonically increasing with respect to $t$ and $\nabla f(x_{t-1})$ is coordinatewise bounded by $G$. We decompose $T_{0,r}$ further by

$$T_{0,r} = \underbrace{\widetilde{\beta}_1^{t-r}\left\langle\nabla f(x_{t-1}), \frac{\Delta_r}{\sqrt{\widetilde{v}_t} + \tau} - \frac{\Delta_r}{\sqrt{\widetilde{v}_{t-1}} + \tau}\right\rangle}_{T_{1,r}} + \underbrace{\widetilde{\beta}_1^{t-r}\left\langle\nabla f(x_{t-1}), \frac{\Delta_r}{\sqrt{\widetilde{v}_{t-1}} + \tau}\right\rangle}_{T_{2,r}}.$$

A bound for $T_{1,r}$ can be obtained as:

$$T_{1,r} = \widetilde{\beta}_1^{t-r} \left\langle \nabla f(x_{t-1}), \frac{\Delta_r(\sqrt{\widetilde{v}_{t-1}} - \sqrt{\widetilde{v}_t})}{(\sqrt{\widetilde{v}_t} + \tau)(\sqrt{\widetilde{v}_{t-1}} + \tau)} \right\rangle$$

$$= \widetilde{\beta}_1^{t-r} \left\langle \nabla f(x_{t-1}), \frac{-\Delta_r \Delta_t^2}{(\sqrt{\widetilde{v}_t} + \tau)(\sqrt{\widetilde{v}_{t-1}} + \tau)(\sqrt{\widetilde{v}_{t-1}} + \sqrt{\widetilde{v}_t})} \right\rangle$$

$$\leq \widetilde{\beta}_1^{t-r} \left\langle |\nabla f(x_{t-1})|, \frac{|\Delta_r|\Delta_t^2}{(\widetilde{v}_t + \tau^2)(\sqrt{\widetilde{v}_{t-1}} + \tau)} \right\rangle$$

$$\leq \widetilde{\beta}_1^{t-r} \sum_{j=1}^{d} G \left[ \frac{|\Delta_r|\Delta_t^2}{(\widetilde{v}_t + \tau^2)(\sqrt{\widetilde{v}_{t-1}} + \tau)} \right]_j$$

$$\leq \frac{\|\Phi_1^K\| G \widetilde{\beta}_1^{t-r}}{\tau} \sum_{j=1}^{d} \left[ \frac{\Delta_t^2}{\widetilde{v}_t} \right]_j.$$

Lemma G.2 is used to obtain the final inequality. For $T_{2,r}$, we apply a further decomposition for $\gamma_r > 0$ allowed to be arbitrary within a compact interval $\epsilon \eta_\ell$-bounded away from 0,

$$T_{2,r} = \underbrace{\widetilde{\beta}_1^{t-r} \left\langle \frac{\nabla f(x_{t-1})}{\sqrt{\widetilde{v}_{t-1}} + \tau}, \Delta_r + \gamma_r \nabla f(x_{t-1}) \right\rangle}_{T_{2,r}^1} - \gamma_r \widetilde{\beta}_1^{t-r} \left\| \frac{\nabla f(x_{t-1})}{\sqrt{\sqrt{\widetilde{v}_{t-1}} + \tau}} \right\|^2.$$

For expository purposes, we present the case in which local gradient clipping is not triggered. The analysis directly generalizes to the setting where clipping activates, where we assume that not all gradients are clipped to 0 for the algorithm to proceed. Unraveling the definition of $\Delta_r$ gives

$$\Delta_r = \frac{-\eta_\ell}{|\mathcal{S}^r|} \sum_{i \in \mathcal{S}^r} \sum_{p=1}^{K} \frac{g_{i,p}^r}{\sqrt{v_{i,p}^r} + \varepsilon},$$

which intuits the following value

$$\gamma_r := \frac{\eta_\ell}{|\mathcal{S}^r|} \sum_{i \in \mathcal{S}^r} \sum_{p=1}^{K} \frac{1}{\sqrt{v_{i,p}^r} + \varepsilon}.$$

We have by Assumption 2 and Lemma D.3 that

$$\gamma_r \in [\widetilde{\gamma}_1, \widetilde{\gamma}_2] := \left[ \eta_\ell \sum_{p=1}^{K} \frac{1}{\sqrt{v_0 + dKG^2} + \varepsilon}, \frac{\eta_\ell K}{\sqrt{v_0} + \varepsilon} \right].$$

Expanding $T_{2,r}^1$ for $\alpha_r > 0$ to be fixed,

$$
\widetilde{\beta}_1^{t-r} \left\langle \frac{\nabla f(x_{t-1})}{\sqrt{\widetilde{v}_{t-1}} + \tau}, \Delta_r + \gamma_r \nabla f(x_{t-1}) \right\rangle
$$

$$
= \frac{\widetilde{\beta}_1^{t-r}}{|\mathcal{S}^r|} \sum_{i \in \mathcal{S}^r} \sum_{p=1}^K \left\langle \frac{\nabla f(x_{t-1})}{\sqrt{\widetilde{v}_{t-1}} + \tau}, \frac{\eta_\ell \left( \nabla f(x_{t-1}) - g_{i,p}^r \right)}{\sqrt{v_p} + \varepsilon} \right\rangle
$$

$$
\leq \frac{\eta_\ell \widetilde{\beta}_1^{t-r} \alpha_r K}{2|\mathcal{S}^r|} \sum_{i \in \mathcal{S}^r} \left\| \frac{\nabla f(x_{t-1})}{\sqrt{\sqrt{\widetilde{v}_{t-1}} + \tau}} \right\|^2
$$

$$
+ \frac{\eta_\ell \widetilde{\beta}_1^{t-r}}{2|\mathcal{S}^r| \alpha_r} \sum_{i \in \mathcal{S}^r} \sum_{p=1}^K \left\| \frac{\left( \nabla f(x_{t-1}) - \nabla F_i(x_{i,p-1}^r) \right)}{\sqrt{\sqrt{\widetilde{v}_{t-1}} + \tau} \left( \sqrt{v_p} + \varepsilon \right)} \right\|^2
$$

$$
\leq \frac{\eta_\ell \widetilde{\beta}_1^{t-r} \alpha_r K}{2} \left\| \frac{\nabla f(x_{t-1})}{\sqrt{\sqrt{\widetilde{v}_{t-1}} + \tau}} \right\|^2
$$

$$
+ \frac{\eta_\ell \widetilde{\beta}_1^{t-r}}{2|\mathcal{S}^r| \alpha_r \tau (\sqrt{v_0} + \varepsilon)^2} \sum_{i \in \mathcal{S}^r} \sum_{p=1}^K \left\| \nabla f(x_{t-1}) - \nabla F_i(x_{i,p-1}^r) \right\|^2.
$$

where in the first inequality we drew the deterministic gradient instead of accessing the stochastic sample via full gradient descent. The first term is controlled by setting

$$
\alpha_r = \frac{\gamma_r}{2\eta_\ell K} \in [\widetilde{\alpha}_1, \widetilde{\alpha}_2] := \left[ \frac{1}{2\sqrt{v_0 + dKG^2} + 2\varepsilon}, \frac{1}{2\sqrt{v_0} + 2\varepsilon} \right].
$$

We aim to bound the second term via majorization and telescoping arguments. We have by $L$-smoothness, Lemmas D.1, D.2, and Assumption 2 that

$$
\left\| \nabla f(x_{t-1}) - \nabla F_i(x_{i,p-1}^r) \right\|^2 \leq \frac{1}{N} \sum_{i' \in [N]} \left\| \left( \nabla F_{i'}(x_{t-1}) - \nabla F_i(x_{i,p-1}^r) \right) \right\|^2
$$

$$
= \frac{1}{N} \sum_{i' \in [N]} \left\| \left( \nabla F_{i'}(x_{t-1}) - \nabla F_{i'}(x_{r-1}) + \nabla F_{i'}(x_{r-1}) - \nabla F_i(x_{i,p-1}^r) \right) \right\|^2
$$

$$
\leq \frac{2}{N} \sum_{i' \in [N]} \left( \left\| \nabla F_{i'}(x_{t-1}) - \nabla F_{i'}(x_{r-1}) \right\|^2 + \left\| \nabla F_{i'}(x_{r-1}) - \nabla F_i(x_{i,p-1}^r) \right\|^2 \right)
$$

$$
\leq \frac{2L}{N} \sum_{i' \in [N]} \left\| x_{t-1} - x_{r-1} \right\|^2 + \frac{2\widetilde{L}}{N} \sum_{i' \in [N]} \left\| x_{i,p-1}^r - x_{i,0}^r \right\|^2
$$

$$
= 2L \left\| x_{t-1} - x_{r-1} \right\|^2 + 2\widetilde{L} \left\| x_{i,p-1}^r - x_{i,0}^r \right\|^2
$$

$$
\leq 2L(t-r) \sum_{o=r}^{t-1} \left\| x_o - x_{o-1} \right\|^2 + 2\widetilde{L} \| \Phi_1^p \|^2
$$

$$
\leq 2L(t-r)^2 \| \Phi_2^K \|^2 + 2\widetilde{L} \| \Phi_1^K \|^2.
$$

Note that the first inequality was obtained by Jensen, while the third inequality uses that the client weights $x_{i,0}^r$ are synchronized to the global weights $x_{r-1}$ for $\forall i \in [N]$ at the start of training. Now, we have

$$
\frac{\eta_\ell \widetilde{\beta}_1^{t-r}}{2|\mathcal{S}^r| \alpha_r \tau (\sqrt{v_0} + \varepsilon)^2} \sum_{i \in \mathcal{S}^r} \sum_{p=1}^K \left( 2L(t-r)^2 \| \Phi_2^K \|^2 + 2\widetilde{L} \| \Phi_1^K \|^2 \right)
$$

$$
\leq \frac{\eta_\ell \widetilde{\beta}_1^{t-r} K L (t-r)^2 \| \Phi_2^K \|^2}{\alpha_r \tau (\sqrt{v_0} + \varepsilon)^2} + \frac{\eta_\ell \widetilde{\beta}_1^{t-r} \widetilde{L} K \| \Phi_1^K \|^2}{\alpha_r \tau (\sqrt{v_0} + \varepsilon)^2}.
$$

Collecting terms gathered thus far gives

$$(1 - \widetilde{\beta}_1)\eta \sum_{r=1}^{t} T_{0,r} \leq (1 - \widetilde{\beta}_1)\eta \sum_{r=1}^{t} \left( \frac{\|\Phi_1^K\|G\widetilde{\beta}_1^{t-r}}{\tau} \sum_{j=1}^{d} \left[ \frac{\Delta_t^2}{\widetilde{v}_t} \right]_j - \frac{3\gamma_r \widetilde{\beta}_1^{t-r}}{4} \left\| \frac{\nabla f(x_{t-1})}{\sqrt{\sqrt{\widetilde{v}_{t-1}} + \tau}} \right\|^2 \right)$$

$$+ (1 - \widetilde{\beta}_1)\eta \sum_{r=1}^{t} \left( \frac{\eta_\ell \widetilde{\beta}_1^{t-r} KL(t-r)^2 \|\Phi_2^K\|^2}{\alpha_r \tau (\sqrt{v_0} + \varepsilon)^2} + \frac{\eta_\ell \widetilde{\beta}_1^{t-r} \widetilde{L} K \|\Phi_1^K\|^2}{\alpha_r \tau (\sqrt{v_0} + \varepsilon)^2} \right).$$

Now, let us bound the final term in equation (7),

$$\left\| \frac{\widetilde{\beta}_1^t \widetilde{m}_0 + (1 - \widetilde{\beta}_1) \sum_{r=1}^{t} \widetilde{\beta}_1^{t-r} \Delta_r}{\sqrt{\widetilde{v}_t} + \tau} \right\|^2 \leq 2 \left\| \frac{\widetilde{\beta}_1^t \widetilde{m}_0}{\sqrt{\widetilde{v}_t} + \tau} \right\|^2 + 2 \left\| \frac{(1 - \widetilde{\beta}_1) \sum_{r=1}^{t} \widetilde{\beta}_1^{t-r} \Delta_r}{\sqrt{\widetilde{v}_t} + \tau} \right\|^2$$

$$\leq 2 \left\| \frac{\widetilde{\beta}_1^t \widetilde{m}_0}{\sqrt{\widetilde{v}_t} + \tau} \right\|^2 + 2 \left\| \frac{(1 - \widetilde{\beta}_1) \sum_{r=1}^{t} \widetilde{\beta}_1^{t-r} \max_{r \in [t]} |\Delta_r|}{\sqrt{\widetilde{v}_t} + \tau} \right\|^2$$

$$\leq 2 \left\| \frac{\widetilde{\beta}_1^t \widetilde{m}_0}{\sqrt{\widetilde{v}_t} + \tau} \right\|^2 + 2 \left\| \frac{(1 - \widetilde{\beta}_1^t)}{\sqrt{\widetilde{v}_t} + \tau} \right\|^2 \|\Phi_1^K\|^2$$

$$\leq 2 \left\| \frac{\widetilde{\beta}_1^t \widetilde{m}_0}{\sqrt{\widetilde{v}_t} + \tau} \right\|^2 + 2d \frac{\|\Phi_1^K\|^2}{\tau^2}.$$

Substituting into equation (7) gives that

$$f(x_t) \leq f(x_{t-1}) + \eta T_{0,0} + \eta^2 L \left\| \frac{\widetilde{\beta}_1^t \widetilde{m}_0}{\sqrt{\widetilde{v}_t} + \tau} \right\|^2 + \frac{\eta^2 L d \|\Phi_1^K\|^2}{\tau^2} + (1 - \widetilde{\beta}_1)\eta \sum_{r=1}^{t} \left( \frac{\|\Phi_1^K\|G\widetilde{\beta}_1^{t-r}}{\tau} \sum_{j=1}^{d} \left[ \frac{\Delta_t^2}{\widetilde{v}_t} \right]_j \right)$$

$$+ (1 - \widetilde{\beta}_1)\eta \sum_{r=1}^{t} \left( \frac{\eta_\ell \widetilde{\beta}_1^{t-r} KL(t-r)^2 \|\Phi_2^K\|^2}{\alpha_r \tau (\sqrt{v_0} + \varepsilon)^2} + \frac{\eta_\ell \widetilde{\beta}_1^{t-r} \widetilde{L} K \|\Phi_1^K\|^2}{\alpha_r \tau (\sqrt{v_0} + \varepsilon)^2} \right)$$

$$+ (1 - \widetilde{\beta}_1)\eta \sum_{r=1}^{t} \left( -\frac{3\gamma_r \widetilde{\beta}_1^{t-r}}{4} \left\| \frac{\nabla f(x_{t-1})}{\sqrt{\sqrt{\widetilde{v}_{t-1}} + \tau}} \right\|^2 \right). \tag{9}$$

Note that the exponential decay caused by $\widetilde{\beta}_1$ in the third term will expectedly dominate the effect of first order moment initialization $\widetilde{m}_0$ as training progresses, and summation over $t \in [T]$ gives $\mathcal{O}(1)$. We initialize $\widetilde{m}_0 \leftarrow 0$ to further simplify the equations. We also further exacerbate the upper bound by substituting $\widetilde{\gamma}_1, \widetilde{\alpha}_1$ into $\gamma_r, \alpha_r$ respectively, which achieves independence from $r$. Telescoping equation (9) then gives

$$\frac{3(1 - \widetilde{\beta}_1)\eta\widetilde{\gamma}_1}{4} \sum_{t=1}^{T} \sum_{r=1}^{t} \widetilde{\beta}_1^{t-r} \left\| \frac{\nabla f(x_{t-1})}{\sqrt{\sqrt{\widetilde{v}_{t-1}} + \tau}} \right\|^2 \leq f(x_0) - f(x^*) + \frac{(1 - \widetilde{\beta}_1)\eta \|\Phi_1^K\|G}{\tau} \sum_{t=1}^{T} \sum_{r=1}^{t} \sum_{j=1}^{d} \widetilde{\beta}_1^{t-r} \left[ \frac{\Delta_t^2}{\widetilde{v}_t} \right]_j$$

$$+ \frac{\eta^2 LTd \|\Phi_1^K\|^2}{\tau^2} + \frac{(1 - \widetilde{\beta}_1)\eta\eta_\ell K}{\widetilde{\alpha}_1 \tau (\sqrt{v_0} + \varepsilon)^2} \sum_{t=1}^{T} \sum_{r=1}^{t} \left( L\widetilde{\beta}_1^{t-r}(t-r)^2 \|\Phi_2^K\|^2 + \widetilde{L}\widetilde{\beta}_1^{t-r} \|\Phi_1^K\|^2 \right). \tag{10}$$

To complete the proof, we aim to ease a logarithm out from the third term on the right hand side. For this purpose, we induce a recursion with a $\log$ bound

$$(1 - \widetilde{\beta}_1) \sum_{t=1}^{T} \sum_{r=1}^{t} \widetilde{\beta}_1^{t-r} \frac{\Delta_{t,j}^2}{\sum_{\ell=1}^{t} \Delta_{\ell,j}^2 + \tau^2} \leq \sum_{t=1}^{T} (1 - \widetilde{\beta}_1^t) \frac{\Delta_{t,j}^2}{\sum_{\ell=1}^{t} \Delta_{\ell,j}^2 + \tau^2}$$

$$\leq a_T + c_T \log(1 + b_T). \tag{11}$$

Setting $T = 1$ gives

$$(1 - \widetilde{\beta}_1) \frac{\Delta_{1,j}^2}{\Delta_{1,j}^2 + \tau^2} \leq a_1 + c_1 \log(1 + b_1),$$

and setting $a_T = 1 - \widetilde{\beta}_1$ satisfies this inequality (among other choices). Assuming formula (11) holds for $T$, let us explore the induction condition for $T + 1$, which is

$$\sum_{t=1}^{T}(1 - \widetilde{\beta}_1^t)\frac{\Delta_{t,j}^2}{\sum_{\ell=1}^{t}\Delta_{\ell,j}^2 + \tau^2} + (1 - \widetilde{\beta}_1^{T+1})\frac{\Delta_{T+1,j}^2}{\sum_{\ell=1}^{T+1}\Delta_{\ell,j}^2 + \tau^2} \leq a_{T+1} + c_{T+1}\log\left(1 + b_{T+1}\right).$$

For simplicity, we impose that $c_t$ is a monotonically increasing non-negative sequence of $t$. We intend to contain the increase in the left hand side as $T$ grows in the $\log$ argument only, in the right hand side. Therefore, we select $a_{T+1} = a_T$. For a suitable choice of $b_{T+1}$ satisfying strong induction, it is enough to resolve

$$(1 - \widetilde{\beta}_1^{T+1})\frac{\Delta_{T+1,j}^2}{\sum_{\ell=1}^{T+1}\Delta_{\ell,j}^2 + \tau^2} \leq c_{T+1}\log\left(\frac{1 + b_{T+1}}{1 + b_T}\right) = c_{T+1}\log\left(1 + \frac{b_{T+1} - b_T}{1 + b_T}\right).$$

Here, we used monotonicity of $c_t$. Noting that $\log(1 + x) \geq x/(1 + x)$, it is again enough to resolve

$$\frac{\Delta_{T+1,j}^2}{\sum_{\ell=1}^{T+1}\Delta_{\ell,j}^2 + \tau^2} \leq \frac{c_{T+1}(b_{T+1} - b_T)}{b_{T+1} + 1}$$

$$\iff \frac{\Delta_{T+1,j}^2}{\sum_{\ell=1}^{T+1}\Delta_{\ell,j}^2 + \tau^2} + c_{T+1}b_T \leq \left(c_{T+1} - \frac{\Delta_{T+1,j}^2}{\sum_{\ell=1}^{T+1}\Delta_{\ell,j}^2 + \tau^2}\right)b_{T+1}.$$

By positivity of $b_t$ for $t > 1$, a necessary condition is therefore that

$$c_{T+1} \geq \frac{\Delta_{T+1,j}^2}{\sum_{\ell=1}^{T+1}\Delta_{\ell,j}^2 + \tau^2}$$

In order to enhance the tightness of our bound, we choose the minimal permissible value $c_t = 1$ uniformly, which is attained as a suprema. In this setting, we are left with a recursion

$$\frac{\Delta_{T+1,j}^2}{\sum_{\ell=1}^{T+1}\Delta_{\ell,j}^2 + \tau^2} = \frac{b_{T+1} - b_T}{b_{T+1} + 1},$$

and collecting the terms in the form $b_{T+1} = b_T\omega_1(\Delta) + \omega_2(\Delta)$ would provide an optimal recursive bound given our simplifying assumptions, starting with $b_1 = 0$. A less optimal but simpler bound can be formed by selecting $b_{T+1} = b_T + \Delta_{T+1,j}^2/\tau^2$ for $b_1 = \Delta_{1,j}^2/\tau^2$. Therefore, we arrive at

$$(1 - \widetilde{\beta}_1)\sum_{t=1}^{T}\sum_{r=1}^{t}\widetilde{\beta}_1^{t-r}\frac{\Delta_{t,j}^2}{\sum_{\ell=1}^{t}\Delta_{\ell,j}^2 + \tau^2} \leq 1 - \widetilde{\beta}_1 + \log\left(1 + \sum_{\ell=1}^{T}\left(\frac{\Delta_{\ell,j}}{\tau}\right)^2\right)$$

$$\leq 1 - \widetilde{\beta}_1 + \log\left(1 + \frac{T\|\Phi_1^K\|^2}{\tau^2}\right). \qquad (12)$$

The remaining term to be bounded in equation (10) is given

$$\frac{(1 - \widetilde{\beta}_1)\eta\eta_\ell KL}{\widetilde{\alpha}_1\tau(\sqrt{v_0} + \varepsilon)^2}\sum_{t=1}^{T}\sum_{r=1}^{t}\left(\widetilde{\beta}_1^{t-r}(t - r)^2\|\Phi_2^K\|^2\right).$$

The trick is to notice that the explosion of the series caused by double summation is culled selectively in reverse chronological order by the exponential, rendering the tail end asymptotically vacuous. Note that $(1 - \widetilde{\beta}_1)$ stabilizes the divergence as $\widetilde{\beta}_1 \to 1^-$ in the limit. By a change of variable $u = t - r$,

$$(1 - \widetilde{\beta}_1)\sum_{t=1}^{T}\sum_{r=1}^{t}\widetilde{\beta}_1^{t-r}(t - r)^2 = (1 - \widetilde{\beta}_1)\sum_{u=0}^{T-1}\widetilde{\beta}_1^u u^2(T - u).$$

Defining

$$\widetilde{u}_0(\widetilde{\beta}_1) = \inf\{u \in \mathbb{N} : \widetilde{\beta}_1^v v^2 < \frac{1}{v^2} \text{ for } \forall v \geq u\},$$

let

$$c(\widetilde{\beta}_1) := \sum_{u=0}^{\widetilde{u}_0(\widetilde{\beta}_1)} \widetilde{\beta}_1^u u^2 + \int_{\widetilde{u}_0(\widetilde{\beta}_1)}^{\infty} \frac{1}{x^2} \mathrm{d}x.$$

Then, we claim that

$$(1 - \widetilde{\beta}_1) \sum_{t=1}^{T} \sum_{r=1}^{t} \widetilde{\beta}_1^{t-r}(t-r)^2 \leq (1 - \widetilde{\beta}_1)c(\widetilde{\beta}_1)T.$$

We prove this by induction. The case $T = 1$ is trivial. Now, assume the desired inequality holds until $T$. For $T + 1$, we want to show

$$(1 - \widetilde{\beta}_1) \sum_{u=0}^{T} \widetilde{\beta}_1^u u^2 (T - u + 1) \leq (1 - \widetilde{\beta}_1)c(\widetilde{\beta}_1)(T + 1)$$

$$\iff (1 - \widetilde{\beta}_1) \sum_{u=0}^{T-1} \widetilde{\beta}_1^u u^2 (T - u) + (1 - \widetilde{\beta}_1) \sum_{u=0}^{T} \widetilde{\beta}_1^u u^2 \leq (1 - \widetilde{\beta}_1)c(\widetilde{\beta}_1)(T + 1)$$

and thus by the inductive hypothesis it is enough to show

$$\sum_{u=0}^{T} \widetilde{\beta}_1^u u^2 \leq c(\widetilde{\beta}_1).$$

However, this is trivial by the definition of $c(\widetilde{\beta}_1)$. Upon substitution into equation (10) and noting that

$$\frac{3(1 - \widetilde{\beta}_1)\eta\widetilde{\gamma}_1}{4} \sum_{t=1}^{T} \sum_{r=1}^{t} \widetilde{\beta}_1^{t-r} \left\| \frac{\nabla f(x_{t-1})}{\sqrt{\sqrt{\widetilde{v}_{t-1}} + \tau}} \right\|^2 \geq \frac{3(1 - \widetilde{\beta}_1)\eta\widetilde{\gamma}_1 T}{4\left(\sqrt{T\|\Phi_1^K\|^2 + \widetilde{v}_0} + \tau\right)} \min_{t \in [T]} \|\nabla f(x_{t-1})\|^2$$

we simplify as

$$\frac{3(1 - \widetilde{\beta}_1)\eta\widetilde{\gamma}_1 T}{4\left(\sqrt{T\|\Phi_1^K\|^2 + \widetilde{v}_0} + \tau\right)} \min_{t \in [T]} \|\nabla f(x_{t-1})\|^2 \leq f(x_0) - f(x^*) + \frac{\eta^2 LTd\|\Phi_1^K\|^2}{\tau^2}$$

$$+ \frac{(1 - \widetilde{\beta}_1^T)\eta\eta_\ell KT\widetilde{L}\|\Phi_1^K\|^2}{\widetilde{\alpha}_1\tau(v_0 + \varepsilon)^2} + \frac{(1 - \widetilde{\beta}_1)\eta\eta_\ell KTLc(\widetilde{\beta}_1)\|\Phi_2^K\|^2}{\widetilde{\alpha}_1\tau(v_0 + \varepsilon)^2} \tag{13}$$

$$+ \frac{\eta d\|\Phi_1^K\|G\left(1 - \widetilde{\beta}_1 + \log\left(1 + \frac{T\|\Phi_1^K\|^2}{\tau^2}\right)\right)}{\tau}$$

Therefore, we immediately conclude that

$$\min_{t \in [T]} \|\nabla f(x_{t-1})\|^2 \leq \frac{\Psi_1 + \Psi_2 + \Psi_3 + \Psi_4 + \Psi_5}{\Psi_6},$$

where

$$\Psi_1 = f(x_0) - f(x^*),$$

$$\Psi_2 = \frac{\eta^2 LTd\|\Phi_1^K\|^2}{\tau^2},$$

$$\Psi_3 = \frac{(1 - \widetilde{\beta}_1^T)\eta\eta_\ell K\widetilde{L}T\|\Phi_1^K\|^2}{\widetilde{\alpha}_1\tau(\sqrt{v_0} + \varepsilon)^2},$$

$$\Psi_4 = \frac{(1 - \widetilde{\beta}_1)\eta\eta_\ell KLTc(\widetilde{\beta}_1)\|\Phi_2^K\|^2}{\widetilde{\alpha}_1\tau(\sqrt{v_0} + \varepsilon)^2},$$

$$\Psi_5 = \frac{\eta d\|\Phi_1^K\|G\left(1 - \widetilde{\beta}_1 + \log\left(1 + \frac{T\|\Phi_1^K\|^2}{\tau^2}\right)\right)}{\tau},$$

$$\Psi_6 = \frac{3(1 - \widetilde{\beta}_1)\eta\widetilde{\gamma}_1 T}{4\left(\sqrt{T\|\Phi_1^K\|^2 + \widetilde{v}_0} + \tau\right)}.$$

Here, the constant $c$ is defined with respect to $\widetilde{\beta}_1$ as

$$c(\widetilde{\beta}_1) := \sum_{u=0}^{\widetilde{u}_0(\widetilde{\beta}_1)} \widetilde{\beta}_1^u u^2 + \int_{\widetilde{u}_0(\widetilde{\beta}_1)}^{\infty} \frac{1}{x^2} \mathrm{d}x \quad \text{for} \quad \widetilde{u}_0(\widetilde{\beta}_1) = \inf\{u \in \mathbb{N} : \widetilde{\beta}_1^v v^2 < \frac{1}{v^2} \text{ for } \forall v \geq u\}$$

and the intermediary $\widetilde{\gamma}_1, \widetilde{\alpha}_1$ values are defined as

$$\widetilde{\gamma}_1 := \eta_\ell \frac{K}{\sqrt{v_0 + dKG^2} + \varepsilon}, \quad \widetilde{\alpha}_1 := \frac{1}{2\sqrt{v_0 + dKG^2} + 2\varepsilon}.$$

This concludes the proof. $\qquad\square$

Note that we have also shown the following two useful lemmas:

**Lemma D.7.** *For $\widetilde{\beta}_1 \in [0, 1)$ and $T \in \mathbb{Z}_{\geq 0}$, let*

$$\widetilde{u}_0(\widetilde{\beta}_1) = \inf\{u \in \mathbb{N} : \widetilde{\beta}_1^v v^2 < \frac{1}{v^2} \text{ for } \forall v \geq u\},$$

*and*

$$c(\widetilde{\beta}_1) := \sum_{u=0}^{\widetilde{u}_0(\widetilde{\beta}_1)} \widetilde{\beta}_1^u u^2 + \int_{\widetilde{u}_0(\widetilde{\beta}_1)}^{\infty} \frac{1}{x^2} \mathrm{d}x.$$

*Then, we have that*

$$\sum_{t=1}^{T} \sum_{r=1}^{t} \widetilde{\beta}_1^{t-r}(t-r)^2 \leq c(\widetilde{\beta}_1) T.$$

**Lemma D.8.** *Let $\Delta_{\ell,j} \in \mathbb{R}$, $\widetilde{\beta}_1 \in [0, 1)$, and $T \in \mathbb{Z}_{\geq 0}$. Then,*

$$(1 - \widetilde{\beta}_1) \sum_{t=1}^{T} \sum_{r=1}^{t} \widetilde{\beta}_1^{t-r} \frac{\Delta_{t,j}^2}{\sum_{\ell=1}^{t} \Delta_{\ell,j}^2 + \tau^2} \leq 1 - \widetilde{\beta}_1 + \log\left(1 + \frac{T\|\Phi_1^K\|^2}{\tau^2}\right).$$

We present the following corollary.

**Corollary D.9.** *Any of the following conditions are sufficient to ensure convergence of Algorithm 2:*

$$(A): \quad \eta_\ell \leq \mathcal{O}(T^{-1/2}) \quad \text{for} \quad \Omega(T^{-1}) < \eta\eta_\ell < \mathcal{O}(1),$$
$$(B): \quad \eta_\ell = \Theta(T^{-\frac{49}{100}}) \quad \text{for} \quad \Omega(T^{-\frac{1}{2}}) < \eta < \mathcal{O}(T^{\frac{12}{25}}).$$

*Proof.* The proof is formed by comparing orders of $T$. Recall that $\widetilde{\gamma}_1 = \Theta(\eta_\ell)$ and $\widetilde{L} = \Theta(\eta_\ell^{-1})$. As $\Phi_1^K = \Theta(\eta_\ell)$ and $\Phi_2^K = \Theta\left(\min\{\eta, \eta\eta_\ell\}\right)$, we have for $\eta = \Theta(T^{p_1})$ and $\eta_\ell = \Theta(T^{p_2})$,

$$\psi_1 = \Theta(1)$$
$$\psi_2 = \eta^2 \eta_\ell^2 T$$
$$\psi_3 = \eta\eta_\ell^2 T$$
$$\psi_4 = \begin{cases} \eta^3 \eta_\ell^3 T & \text{if } \mathcal{O}(\eta_\ell) \leq \mathcal{O}(1) \\ \eta^3 \eta_\ell T & \text{if } \Theta(\eta_\ell) > \Omega(1) \end{cases}$$
$$\psi_5 = \eta\eta_\ell \log(1 + T\eta_\ell^2)$$
$$\psi_6 = \begin{cases} \eta\eta_\ell T & \text{if } \mathcal{O}(T\eta_\ell^2) \leq \mathcal{O}(1) \\ \eta\sqrt{T} & \text{if } \Theta(T\eta_\ell^2) > \Omega(1) \end{cases}.$$

If $\mathcal{O}(T\eta_\ell^2) \leq \mathcal{O}(1)$, then $\mathcal{O}(\eta_\ell) \leq \mathcal{O}(1)$ which implies

$$\frac{\psi_1}{\psi_6} : (\eta\eta_\ell T)^{-1} = \Theta\left(T^{-(p_1+p_2+1)}\right)$$

$$\frac{\psi_2}{\psi_6} : \eta\eta_\ell = \Theta\left(T^{p_1+p_2}\right)$$

$$\frac{\psi_3}{\psi_6} : \eta_\ell = \Theta\left(T^{p_2}\right)$$

$$\frac{\psi_4}{\psi_6} : \eta^2\eta_\ell^2 = \Theta\left(T^{2p_1+2p_2}\right)$$

$$\frac{\psi_5}{\psi_6} : \frac{\log(1+T\eta_\ell^2)}{T} = \mathcal{O}(T^{-1})$$

This implies that we must have that $p_2 \leq -1/2$ and $-1 < p_1 + p_2 < 0$ for guaranteed convergence. Thus, $\eta_\ell \leq \mathcal{O}(T^{-1/2})$ such that $\Omega(T^{-1}) < \eta\eta_\ell < \mathcal{O}(1)$ is a sufficient condition. For instance, let $\eta_\ell = \Theta(T^{-1/2})$ and $\Omega(T^{-1/2}) < \eta < \mathcal{O}(T^{1/2})$.

Now, assume $\Theta(T\eta_\ell^2) > \Omega(1)$. If $\Theta(\eta_\ell) > \Omega(1)$, $\Psi_3/\Psi_6$ diverges. Therefore, let $\eta_\ell \leq \mathcal{O}(1)$. We have

$$\frac{\psi_1}{\psi_6} : (\eta\sqrt{T})^{-1} = \Theta(T^{-p_1-\frac{1}{2}})$$

$$\frac{\psi_2}{\psi_6} : \eta\eta_\ell^2\sqrt{T} = \Theta(T^{p_1+2p_2+\frac{1}{2}})$$

$$\frac{\psi_3}{\psi_6} : \eta_\ell^2\sqrt{T} = \Theta(T^{2p_2+\frac{1}{2}})$$

$$\frac{\psi_4}{\psi_6} : \eta^2\eta_\ell^3\sqrt{T} = \Theta(T^{2p_1+3p_2+\frac{1}{2}})$$

$$\frac{\psi_5}{\psi_6} : \frac{\eta_\ell \log(1+T\eta_\ell^2)}{\sqrt{T}} < \mathcal{O}(T^{-\frac{1}{2}+p_2})$$

Therefore, it suffices to satisfy

$$-\frac{1}{2} < p_2 \leq -\frac{1}{4}, \quad -\frac{1}{2} < p_1, \quad p_1 + 2p_2 < -\frac{1}{2}, \quad 2p_1 + 3p_2 < -\frac{1}{2}.$$

An example satisfying these conditions are

$$\eta_\ell = \Theta(T^{-\frac{49}{100}}), \quad \Omega(T^{-\frac{1}{2}}) < \eta < \mathcal{O}(T^{\frac{12}{25}}).$$

$\square$

Note that for all cases, $\eta_\ell$ must decay to establish convergence. However, striking a balance between local and global learning rates provably allows for greater than $\Omega(T^{1/3})$ divergence in the server learning rate without nullifying desirable convergence properties. This theoretically demonstrates the enhanced robustness properties of adaptive client-side federated learning algorithms to mitigate suboptimal choices of server learning rates.

**Corollary D.10.** *Algorithm 2 converges at rate $\mathcal{O}(T^{-1/2})$.*

*Proof.* If $\mathcal{O}(T\eta_\ell^2) \leq \mathcal{O}(1)$, then we juxtapose $\psi_1/\psi_6$ and $\psi_2/\psi_6$. It is clear that the minimax value of the respective powers are attained at $p_1 + p_2 = -1/2$, realized by $p_2 = -1/2$ and $p_1 = 0$. In this case, clearly $\Theta(\psi_i/\psi_6) \leq \mathcal{O}(T^{-1/2})$ for $1 \leq i \leq 5$. If $\Theta(T\eta_\ell^2) > \Omega(1)$, then our strategy should be to minimize $p_2$ due to positive coefficients in the powers $\psi_i/\psi_6$. Thus, let $p_2 = -1/2 + \varepsilon$ for $1 \gg \varepsilon > 0$. Then, the order of decay in $\psi_2/\psi_6$ is $p_1 - 1/2 + 2\varepsilon$, which is once again matched against $-p_1 - 1/2$, the power of $\psi_1/\psi_6$. Taking the limit $\varepsilon \to 0^+$, minimax$\{p_1 - 1/2, -p_1 - 1/2\}$ for the range $-1/2 < p_1$ is attained at $p_1 = 0$. This sets the maximal decay rate to $\mathcal{O}(T^{-1/2})$ for the second case. $\square$

## D.2  Extension to Adam

The extension to the case where Adam is selected as the optimizer for the server, or for both the server and client is straightforward. We present the latter as it generalizes the former analysis. As in Lemma D.1, we have the following bound for the compressed SM3 estimates of the second moment,

$$v_k(j) \geq v_0(j) + \sum_{\ell=1}^{\lceil \frac{k}{z} \rceil} \left( g_{i,(\ell-1)z+1}^t(j) \right)^2 \quad \text{for} \quad j \in \{1, \dots, d\} \quad \text{and} \quad k \in \{1, \dots, K\},$$

which allows bounds to be established for the local and global pseudogradients following analogous logic as Lemmas D.2, F.2. As before, we arrive at equation (8) where due to exponential moving averaging on the server side, we have

$$\widetilde{v}_t = \widetilde{\beta}_2^t \widetilde{v}_0 + (1 - \widetilde{\beta}_2) \sum_{\ell=1}^t \widetilde{\beta}_2^{t-r} \Delta_\ell.$$

Now, decompose $T_{0,r}$ as

$$T_{0,r} = \underbrace{\widetilde{\beta}_1^{t-r} \left\langle \nabla f(x_{t-1}), \frac{\Delta_r}{\sqrt{\widetilde{v}_t} + \tau} - \frac{\Delta_r}{\sqrt{\widetilde{\beta}_2 \widetilde{v}_{t-1}} + \tau} \right\rangle}_{T_{1,r}} + \underbrace{\widetilde{\beta}_1^{t-r} \left\langle \nabla f(x_{t-1}), \frac{\Delta_r}{\sqrt{\widetilde{\beta}_2 \widetilde{v}_{t-1}} + \tau} \right\rangle}_{T_{2,r}},$$

where $T_{1,r}$ may be bounded via

$$T_{1,r} = \widetilde{\beta}_1^{t-r} \left\langle \nabla f(x_{t-1}), \frac{\Delta_r(\sqrt{\widetilde{\beta}_2 \widetilde{v}_{t-1}} - \sqrt{\widetilde{v}_t})}{(\sqrt{\widetilde{v}_t} + \tau)(\sqrt{\widetilde{\beta}_2 \widetilde{v}_{t-1}} + \tau)} \right\rangle$$

$$= \widetilde{\beta}_1^{t-r} \left\langle \nabla f(x_{t-1}), \frac{-\Delta_r \Delta_t^2 (1 - \widetilde{\beta}_2)}{(\sqrt{\widetilde{v}_t} + \tau)(\sqrt{\widetilde{\beta}_2 \widetilde{v}_{t-1}} + \tau)(\sqrt{\widetilde{\beta}_2 \widetilde{v}_{t-1}} + \sqrt{\widetilde{v}_t})} \right\rangle$$

$$\leq \frac{\|\Phi_1^K\| G \widetilde{\beta}_1^{t-r} (1 - \widetilde{\beta}_2)}{\tau} \sum_{j=1}^d \left[ \frac{\Delta_t^2}{\widetilde{v}_t} \right]_j.$$

Due to the exponential decay parameter in the first pseudogradient moment, we have

$$\eta \sum_{t=1}^T \sum_{r=1}^t \frac{\|\Phi_1^K\| G \widetilde{\beta}_1^{t-r} (1 - \widetilde{\beta}_2)}{\tau} \sum_{j=1}^d \left[ \frac{\Delta_t^2}{\widetilde{v}_t} \right]_j \leq \eta \sum_{t=1}^T \sum_{r=1}^t \frac{\|\Phi_1^K\|^3 G \widetilde{\beta}_1^{t-r} (1 - \widetilde{\beta}_2)}{\tau^2}$$

$$\leq \frac{\eta \|\Phi_1^K\|^3 G T (1 - \widetilde{\beta}_2)}{\tau^2}.$$

An analogue of the arguments made in the proof of Theorem 4.1 with appropriate modifications, e.g.,

$$\gamma_r := \frac{\eta_\ell}{|\mathcal{S}^r|} \sum_{i \in \mathcal{S}^r} \sum_{p=1}^K \frac{(1 - \beta_1) \sum_{\ell=1}^p \beta_1^{p-\ell}}{\sqrt{(1 - \beta_2) \sum_{\ell=1}^{\lceil \frac{p}{z} \rceil} \beta_2^{\lceil \frac{p}{z} \rceil - \ell} (g_{i,(\ell-1)z+1}^r)^2} + \varepsilon},$$

gives the main change as the asymptotic behavior of $\Psi_5$, which now satisfies

$$\Psi_5 = \Theta \left( \eta \eta_\ell^3 T \right).$$

The convergence rate is still dominated by $\Psi_1$, $\Psi_2$ as in Corollary D.10, which gives $\mathcal{O}(T^{-1/2})$.

## E  Federated Blended Optimization

In federated blended optimization, we distribute local optimizer strategies during the subsampling process which may be formalized as functions that take as input the availability of client resources,

---

**Algorithm 5** Server-side ADAGRAD and client-side optimizer mixture (`FedAda`$^2$)

---

**Require:** Local optimizer strategies $O_1, \ldots, O_{Op}$ (e.g. Adam, AdaGrad, SGD...)
**Require:** Initializations $x_0, \widetilde{v}_0 \geq \tau^2$ and $\widetilde{m}_0 \leftarrow 0$
**Require:** Global decay parameter $\widetilde{\beta}_1 \in [0, 1)$
 1: **for** $t = 1, \ldots, T$ **do**
 2:     Sample participating client multiset $S_l^t$ for each optimizer strategy $l \in [Op]$
 3:     **for** each sampled client collection $l \in [Op]$ (in parallel) **do**
 4:         **for** each client $i \in S_l^t$ (in parallel) **do**
 5:             $x_{i,0}^{t,l} \leftarrow x_{t-1}$
 6:             $x_{i,K(O_l^i)}^{t,l} \leftarrow \text{Optimize}(O_l, i, x_{i,0}^{t,l}, Clip = \text{True})$
 7:             $\Delta_i^{t,l} = w(O_l) \left( x_{i,K(O_l^i)}^{t,l} - x_{t-1} \right)$
 8:         **end for**
 9:     **end for**
10:     $S \leftarrow \sum_{l \in [Op]} |S_l^t|$
11:     $\Delta_t = \frac{1}{S} \sum_{l \in [Op]} \sum_{i \in S_l^t} \Delta_i^{t,l}$
12:     $\widetilde{m}_t = \widetilde{\beta}_1 \widetilde{m}_{t-1} + (1 - \widetilde{\beta}_1) \Delta_t$
13:     $\widetilde{v}_t = \widetilde{v}_{t-1} + \Delta_t^2$
14:     $x_t = x_{t-1} + \eta \frac{\widetilde{m}_t}{\sqrt{\widetilde{v}_t} + \tau}$
15: **end for**

---

and outputs the number of local epochs, $K(O_l^i)$, as well as additional hyperparameters such as delay step size $z$ or preconditioner initialization. These may be chosen to streamline model training based on a variety of factors, such as straggler mitigation or dynamically restricted availability of local resources.

In the general formulation of `FedAda`$^2$, blended optimization allows the trainer to utilize the unique strengths of each individual optimizer, balancing resource constraints and client noise. Each client has the option to run different optimizer strategies as the training rounds progress, depending on varying individual resource constraints or distribution shift in the local data stream. This faithfully corresponds to real-world settings where the availability of local resources are actively dynamic. Future work will provide empirical results on the performance of blended optimization, including identifying the settings in which mixing optimizer strategies are advantageous for distributed learning. The following theorem shows that under certain non-restrictive conditions, blended optimization still allows for convergence of the global gradient objective. We first present a paragraph on the key intuitions and insights behind the proof to improve accessibility.

**Intuitions.** The convergence guarantee for Federated Blended Optimization is obtained by treating each client's local optimizer (e.g., SGD, AdaGrad, Adam, or SM3) as a black-box whose per-step preconditioner and scaling factors are uniformly bounded. First, one shows that no single client update can blow up, i.e., its total local drift $\|\Delta_i\|$ is bounded by $\Phi_1^K$, and similarly the server-side pseudo-gradient remains bounded by $\Phi_2^K$. Smoothness of the global objective then lets us decompose the inner product between the true gradient and the aggregated update into two parts: a small "denominator-drift" term that is absorbed into the variance, and a main descent term that, after introducing an auxiliary pivot $\gamma_r$ and choosing appropriate weights $\alpha_r$, yields a guaranteed reduction in gradient norm. Telescoping the resulting inequality with geometric-series weights recovers the same $O(1/\sqrt{T})$ nonconvex rate as in SM3-`FedAda`$^2$++. In this way, federating over multiple heterogeneous optimizers requires only uniform bounds on their preconditioners and step-sizes, and no change in the high-level convergence rate, provided each strategy respects these minimal stability conditions.

**Theorem E.1.** *Let Assumptions 1, 2 hold. Given client $i \in [N]$, strategy $l \in [Op]$, global timestep $r$, and local timestep $p$, assume that the optimizer strategies satisfy the parameter update rule*

$$ x_{i,p}^{r,l} = x_{i,p-1}^{r,l} - \eta_\ell \sum_{\ell=1}^{p} \frac{a_{i,\ell}^{r,l} g_{i,\ell}^{r,l}}{\vartheta_{i,\ell}^{r,l}(g_{i,1}^{r,l}, \ldots, g_{i,\ell}^{r,l})} $$

*where*

$$0 < m_l \leq \vartheta_{i,\ell}^{r,l}(g_{i,1}^{r,l}, \ldots, g_{i,\ell}^{r,l}) \leq M_l \quad \text{and} \quad 0 < a_l \leq a_{i,\ell}^{r,l} \leq A_l$$

*for all possible values of $i, \ell, r, l$. If $1 \leq K(O_l^i) \leq K$ and $0 < \Xi^- < w(O_l^i) < \Xi^+$, then Algorithm 5 admits an identical convergence bound as Theorem D.6, with $\Psi_3, \Psi_4$ replaced by*

$$\Psi_3 = (1 - \widetilde{\beta}_1^T)\eta\eta_\ell C T \widetilde{L} \|\Phi_1^K\|^2,$$
$$\Psi_4 = (1 - \widetilde{\beta}_1)\eta\eta_\ell C T L c(\widetilde{\beta}_1)\|\Phi_2^K\|^2,$$
$$C = \frac{(\Xi^+)^2 K(K+1)(\max_{l \in [Op]} A_l^2)}{2\widetilde{\alpha}_1 \tau \min_{l \in [Op]} m_l^2}.$$

*The intermediary $\widetilde{\gamma}_1, \widetilde{\alpha}_1$ values are defined as*

$$\widetilde{\gamma}_1 := \eta_\ell \frac{\Xi^- \min_{l \in [Op]} a_l}{\max_{l \in [Op]} M_l}, \quad \widetilde{\alpha}_1 := \frac{\Xi^- \min_{l \in [Op]} a_l}{K(K+1)\max_{l \in [Op]} M_l}.$$

We have opted to provide a looser bound for expository purposes, and the proof straightforwardly generalizes to finer bounds that depend on the individual characteristics of the optimizer strategy (e.g. $m_l, M_l, A_l$, etc). The extension to server-side Adam updates follows analogous steps to Section D.2.

It is easy to show that under the bounded gradient assumption (Assumption 4), Adam, AdaGrad, and SGD (including under SM3 for the former two) all satisfy the optimizer condition depicted in Theorem E.1. In Appendix F and G, we materialize two realizations of this framework as additional examples, using client-side Adam and AdaGrad with delayed preconditioner updates. Note that delayed updates require the debiasing term in Adam to be adjusted accordingly. To prove Theorem E.1, we begin with the following lemma.

**Lemma E.2.** *Under Algorithm 5, $|\Delta_i^{t,l}|$ is bounded by*

$$\Phi_1^K := \eta_\ell \Xi^+ \frac{K(K+1)\max_{l \in [Op]} A_l G}{2\min_{l \in [Op]} m_l},$$

*and the server-side pseudogradient is bounded in absolute value by*

$$\Phi_2^K := \min\left\{\eta\sqrt{(1 - \widetilde{\beta}_1)(1 - \widetilde{\beta}_1^{2t})}, \frac{\eta}{\tau}\Phi_1^K\right\}.$$

*Proof.* Unraveling the definition of $\Delta_i^{t,l}$, we have

$$\Delta_i^{t,l} := -\eta_\ell w(O_l)\left(\sum_{p=1}^{K(O_l^i)} \sum_{\ell=1}^{p} \frac{a_{i,\ell}^{r,l} g_{i,\ell}^{r,l}}{\vartheta_{i,\ell}^{r,l}(g_{i,1}^{r,l}, \ldots, g_{i,\ell}^{r,l})}\right),$$

which immediately gives

$$|\Delta_i^{t,l}| \leq \eta_\ell \Xi^+ \left(\sum_{p=1}^{K}\sum_{\ell=1}^{p} \frac{A_l G}{m_l}\right) = \eta_\ell \Xi^+ \frac{K(K+1)A_l G}{2m_l}.$$

For the server bound, the proof is identical to Lemma D.2. □

We are now ready to prove Theorem E.1.

*Proof.* As the proof follows a similar structure to Theorem 4.1, we provide only an outline for repetitive steps while focusing on differing aspects. As before, $L$-smoothness gives that

$$f(x_t) \leq f(x_{t-1}) + \eta T_{0,0} + (1 - \widetilde{\beta}_1)\eta \sum_{r=1}^{t} T_{0,r} + \frac{\eta^2 L}{2}\left\|\frac{\widetilde{\beta}_1^t \widetilde{m}_0 + (1 - \widetilde{\beta}_1)\sum_{r=1}^{t} \widetilde{\beta}_1^{t-r}\Delta_r}{\sqrt{\widetilde{v}_t} + \tau}\right\|^2 \tag{14}$$

where for $r \in [t]$,

$$T_{0,r} = \widetilde{\beta}_1^{t-r} \left\langle \nabla f(x_{t-1}), \frac{\Delta_r}{\sqrt{\widetilde{v}_t} + \tau} \right\rangle \quad \text{and} \quad T_{0,0} = \left\langle \nabla f(x_{t-1}), \frac{\widetilde{\beta}_1^t \widetilde{m}_0}{\sqrt{\widetilde{v}_t} + \tau} \right\rangle.$$

Decomposing $T_{0,r}$ as

$$T_{0,r} = \underbrace{\widetilde{\beta}_1^{t-r} \left\langle \nabla f(x_{t-1}), \frac{\Delta_r}{\sqrt{\widetilde{v}_t} + \tau} - \frac{\Delta_r}{\sqrt{\widetilde{v}_{t-1}} + \tau} \right\rangle}_{T_{1,r}} + \underbrace{\widetilde{\beta}_1^{t-r} \left\langle \nabla f(x_{t-1}), \frac{\Delta_r}{\sqrt{\widetilde{v}_{t-1}} + \tau} \right\rangle}_{T_{2,r}},$$

$T_{1,r}$ is bounded by

$$T_{1,r} \leq \frac{\|\Phi_1^K\| \|G \widetilde{\beta}_1^{t-r}}{\tau} \sum_{j=1}^{d} \left[ \frac{\Delta_t^2}{\widetilde{v}_t} \right]_j.$$

For $T_{2,r}$, we aim to apply a further decomposition for $\gamma_r > 0$,

$$T_{2,r} = \underbrace{\widetilde{\beta}_1^{t-r} \left\langle \frac{\nabla f(x_{t-1})}{\sqrt{\widetilde{v}_{t-1}} + \tau}, \Delta_r + \gamma_r \nabla f(x_{t-1}) \right\rangle}_{T_{2,r}^1} - \gamma_r \widetilde{\beta}_1^{t-r} \left\| \frac{\nabla f(x_{t-1})}{\sqrt{\sqrt{\widetilde{v}_{t-1}} + \tau}} \right\|^2.$$

Unraveling the definition of $\Delta_r$ gives

$$\Delta_r = \frac{1}{\sum_{l \in [Op]} |S_l^r|} \sum_{l \in [Op]} \sum_{i \in S_l^r} \Delta_i^{r,l} = \frac{-\eta_\ell}{\sum_{l \in [Op]} |S_l^r|} \sum_{l \in [Op]} \sum_{i \in S_l^r} \sum_{p=1}^{K(O_i^i)} \sum_{\ell=1}^{p} \frac{w(O_l) a_{i,\ell}^{r,l} g_{i,\ell}^{r,l}}{\vartheta_{i,\ell}^{r,l}(g_{i,1}^{r,l}, \ldots, g_{i,\ell}^{r,l})},$$

which induces the following value

$$\gamma_r := \frac{\eta_\ell}{\sum_{l \in [Op]} |S_l^t|} \sum_{l \in [Op]} \sum_{i \in S_l^t} \sum_{p=1}^{K(O_i^i)} \sum_{\ell=1}^{p} \frac{w(O_l) a_{i,\ell}^{r,l}}{\vartheta_{i,\ell}^{r,l}(g_{i,1}^{r,l}, \ldots, g_{i,\ell}^{r,l})} = \sum_{l \in [Op]} \gamma_r^l.$$

For the purposes of the proof, we shall consider a local device to have been dropped and unsampled if any runs less than 1 epoch. Then, we have

$$\gamma_r \in [\widetilde{\gamma}_1, \widetilde{\gamma}_2] := \left[ \eta_\ell \frac{\Xi^- \min_{l \in [Op]} a_l}{\max_{l \in [Op]} M_l}, \eta_\ell \frac{\Xi^+ K(K+1) \max_{l \in [Op]} a_l}{2 \min_{l \in [Op]} M_l} \right].$$

Expanding $T_{2,r}^1$ for $\alpha_r^l > 0$ to be fixed,

$$
\widetilde{\beta}_1^{t-r} \left\langle \frac{\nabla f(x_{t-1})}{\sqrt{\widetilde{v}_{t-1}} + \tau}, \Delta_r + \gamma_r \nabla f(x_{t-1}) \right\rangle
$$

$$
= \frac{\widetilde{\beta}_1^{t-r}}{\sum_{l\in[Op]}|S_l^r|} \sum_{l\in[Op]} \sum_{i\in S_l^r} \sum_{p=1}^{K(O_l^i)} \sum_{\ell=1}^{p} \left\langle \frac{\nabla f(x_{t-1})}{\sqrt{\widetilde{v}_{t-1}} + \tau}, \frac{\eta_\ell w(O_l) a_{i,\ell}^{r,l}(\nabla f(x_{t-1}) - g_{i,\ell}^{r,l})}{\vartheta_{i,\ell}^{r,l}(g_{i,1}^{r,l}, \ldots, g_{i,\ell}^{r,l})} \right\rangle
$$

$$
\leq \frac{\eta_\ell \widetilde{\beta}_1^{t-r}}{4\sum_{l\in[Op]}|S_l^r|} \sum_{l\in[Op]} \alpha_r^l \sum_{i\in\mathcal{S}_l^r} K(O_l^i)(K(O_l^i)+1) \left\| \frac{\nabla f(x_{t-1})}{\sqrt{\sqrt{\widetilde{v}_{t-1}} + \tau}} \right\|^2
$$

$$
+ \frac{\eta_\ell \widetilde{\beta}_1^{t-r}}{2\sum_{l\in[Op]}|S_l^r|} \sum_{l\in[Op]} \frac{1}{\alpha_r^l} \sum_{i\in S_l^r} \sum_{p=1}^{K(O_l^i)} \sum_{\ell=1}^{p} \left\| \frac{w(O_l) a_{i,\ell}^{r,l}\left(\nabla f(x_{t-1}) - \nabla F_i(x_{i,\ell-1}^{r,l})\right)}{\vartheta_{i,\ell}^{r,l}(g_{i,1}^{r,l}, \ldots, g_{i,\ell}^{r,l})\sqrt{\sqrt{\widetilde{v}_{t-1}} + \tau}} \right\|^2
$$

$$
\leq \frac{\eta_\ell \widetilde{\beta}_1^{t-r} \max_{l\in[Op]} \alpha_r^l K(K+1)}{4} \left\| \frac{\nabla f(x_{t-1})}{\sqrt{\sqrt{\widetilde{v}_{t-1}} + \tau}} \right\|^2
$$

$$
+ \frac{\eta_\ell \widetilde{\beta}_1^{t-r}(\Xi^+)^2}{2\tau \sum_{l\in[Op]}|S_l^r|} \sum_{l\in[Op]} \frac{A_l^2}{\alpha_r^l m_l^2} \sum_{i\in S_l^r} \sum_{p=1}^{K(O_l^i)} \sum_{\ell=1}^{p} \left\| \nabla f(x_{t-1}) - \nabla F_i(x_{i,\ell-1}^{r,l}) \right\|^2
$$

We aim to control the first term by setting for all $l \in [Op]$

$$
\alpha_r^l = \frac{\gamma_r}{\eta_\ell K(K+1)} \in [\widetilde{\alpha}_1, \widetilde{\alpha}_2] := \left[ \frac{\Xi^- \min_{l\in[Op]} a_l}{K(K+1)\max_{l\in[Op]} M_l}, \frac{\Xi^+ K(K+1)\max_{l\in[Op]} a_l}{2K(K+1)\min_{l\in[Op]} M_l} \right].
$$

Via gradient clipping as before, we have

$$
\left\| \nabla f(x_{t-1}) - \nabla F_i(x_{i,\ell-1}^{r,l}) \right\|^2 \leq 2L(t-r)^2 \|\Phi_2^K\|^2 + 2\widetilde{L}\|\Phi_1^K\|^2.
$$

Noting that

$$
\frac{\eta_\ell \widetilde{\beta}_1^{t-r}(\Xi^+)^2}{2\tau \sum_{l\in[Op]}|S_l^r|} \sum_{l\in[Op]} \frac{A_l^2}{\alpha_r^l m_l^2} \sum_{i\in S_l^r} \sum_{p=1}^{K(O_l^i)} \sum_{\ell=1}^{p} \left\| \nabla f(x_{t-1}) - \nabla F_i(x_{i,\ell-1}^{r,l}) \right\|^2
$$

$$
\leq \frac{\eta_\ell(\Xi^+)^2 K(K+1)(\max_{l\in[Op]} A_l^2)}{2\widetilde{\alpha}_1 \tau \min_{l\in[Op]} m_l^2} \left( L\widetilde{\beta}_1^{t-r}(t-r)^2 \|\Phi_2^K\|^2 + \widetilde{L}\widetilde{\beta}_1^{t-r}\|\Phi_1^K\|^2 \right),
$$

collecting terms into equation (14) gives that

$$
f(x_t) \leq f(x_{t-1}) + \eta T_{0,0} + \eta^2 L \left\| \frac{\widetilde{\beta}_1^t \widetilde{m}_0}{\sqrt{\widetilde{v}_t} + \tau} \right\|^2 + \frac{\eta^2 Ld\|\Phi_1^K\|^2}{\tau^2} + (1-\widetilde{\beta}_1)\eta \sum_{r=1}^{t} \left( \frac{\|\Phi_1^K\|\|G\widetilde{\beta}_1^{t-r}}{\tau} \sum_{j=1}^{d} \left[ \frac{\Delta_t^2}{\widetilde{v}_t} \right]_j \right)
$$

$$
+ (1-\widetilde{\beta}_1)\eta\eta_\ell \sum_{r=1}^{t} \underbrace{\frac{(\Xi^+)^2 K(K+1)(\max_{l\in[Op]} A_l^2)}{2\widetilde{\alpha}_1 \tau \min_{l\in[Op]} m_l^2}}_{C} \left( L\widetilde{\beta}_1^{t-r}(t-r)^2 \|\Phi_2^K\|^2 + \widetilde{L}\widetilde{\beta}_1^{t-r}\|\Phi_1^K\|^2 \right)
$$

$$
+ (1-\widetilde{\beta}_1)\eta \sum_{r=1}^{t} \left( -\frac{3\gamma_r \widetilde{\beta}_1^{t-r}}{4} \left\| \frac{\nabla f(x_{t-1})}{\sqrt{\widetilde{v}_{t-1}} + \tau} \right\|^2 \right). \tag{15}
$$

By initializing $\widetilde{m}_0 \leftarrow 0$ and enhancing the upper bound by substituting $\widetilde{\gamma}_1$ into $\gamma_r$, telescoping gives

$$\frac{3(1-\widetilde{\beta}_1)\eta\widetilde{\gamma}_1}{4} \sum_{t=1}^{T} \sum_{r=1}^{t} \widetilde{\beta}_1^{t-r} \left\| \frac{\nabla f(x_{t-1})}{\sqrt{\sqrt{\widetilde{v}_{t-1}}+\tau}} \right\|^2 \leq f(x_0) - f(x^*) + \frac{(1-\widetilde{\beta}_1)\eta\|\Phi_1^K\|G}{\tau} \sum_{t=1}^{T}\sum_{r=1}^{t}\sum_{j=1}^{d} \widetilde{\beta}_1^{t-r} \left[ \frac{\Delta_t^2}{\widetilde{v}_t} \right]_j$$

$$+ \frac{\eta^2 LTd\|\Phi_1^K\|^2}{\tau^2} + (1-\widetilde{\beta}_1)\eta\eta_\ell C \sum_{t=1}^{T}\sum_{r=1}^{t}\left( L\widetilde{\beta}_1^{t-r}(t-r)^2\|\Phi_2^K\|^2 + \widetilde{L}\widetilde{\beta}_1^{t-r}\|\Phi_1^K\|^2 \right). \qquad (16)$$

Again by noting that

$$\frac{3(1-\widetilde{\beta}_1)\eta\widetilde{\gamma}_1}{4} \sum_{t=1}^{T} \sum_{r=1}^{t} \widetilde{\beta}_1^{t-r} \left\| \frac{\nabla f(x_{t-1})}{\sqrt{\sqrt{\widetilde{v}_{t-1}}+\tau}} \right\|^2 \geq \frac{3(1-\widetilde{\beta}_1)\eta\widetilde{\gamma}_1 T}{4\left(\sqrt{T\|\Phi_1^K\|^2+\widetilde{v}_0}+\tau\right)} \min_{t\in[T]} \|\nabla f(x_{t-1})\|^2,$$

Lemmas D.7 and D.8 give that

$$\frac{3(1-\widetilde{\beta}_1)\eta\widetilde{\gamma}_1 T}{4\left(\sqrt{T\|\Phi_1^K\|^2+\widetilde{v}_0}+\tau\right)} \min_{t\in[T]} \|\nabla f(x_{t-1})\|^2 \leq f(x_0) - f(x^*) + \frac{\eta^2 LTd\|\Phi_1^K\|^2}{\tau^2}$$

$$+ (1-\widetilde{\beta}_1^T)\eta\eta_\ell CT\widetilde{L}\|\Phi_1^K\|^2 + (1-\widetilde{\beta}_1)\eta\eta_\ell CTLc(\widetilde{\beta}_1)\|\Phi_2^K\|^2$$

$$+ \frac{\eta d\|\Phi_1^K\|G\left(1-\widetilde{\beta}_1 + \log\left(1+\frac{T\|\Phi_1^K\|^2}{\tau^2}\right)\right)}{\tau}.$$

This implies that

$$\min_{t\in[T]} \|\nabla f(x_{t-1})\|^2 \leq \frac{\Psi_1 + \Psi_2 + \Psi_3 + \Psi_4 + \Psi_5}{\Psi_6},$$

where

$$\Psi_1 = f(x_0) - f(x^*),$$
$$\Psi_2 = \frac{\eta^2 LTd\|\Phi_1^K\|^2}{\tau^2},$$
$$\Psi_3 = (1-\widetilde{\beta}_1^T)\eta\eta_\ell CT\widetilde{L}\|\Phi_1^K\|^2,$$
$$\Psi_4 = (1-\widetilde{\beta}_1)\eta\eta_\ell CTLc(\widetilde{\beta}_1)\|\Phi_2^K\|^2,$$
$$\Psi_5 = \frac{\eta d\|\Phi_1^K\|G\left(1-\widetilde{\beta}_1 + \log\left(1+\frac{T\|\Phi_1^K\|^2}{\tau^2}\right)\right)}{\tau},$$
$$\Psi_6 = \frac{3(1-\widetilde{\beta}_1)\eta\widetilde{\gamma}_1 T}{4\left(\sqrt{T\|\Phi_1^K\|^2+\widetilde{v}_0}+\tau\right)},$$
$$C = \frac{(\Xi^+)^2 K(K+1)(\max_{l\in[Op]} A_l^2)}{2\widetilde{\alpha}_1 \tau \min_{l\in[Op]} m_l^2}.$$

The intermediary $\widetilde{\gamma}_1, \widetilde{\alpha}_1$ values are defined as

$$\widetilde{\gamma}_1 := \eta_\ell \frac{\Xi^- \min_{l\in[Op]} a_l}{\max_{l\in[Op]} M_l}, \quad \widetilde{\alpha}_1 := \frac{\Xi^- \min_{l\in[Op]} a_l}{K(K+1)\max_{l\in[Op]} M_l}.$$

$\square$

# F   Adam with Delayed Updates (ADMU)

Considering client-side resource constraints in the federated setting, we propose an adapted version of Adam with delayed precondtioner updates aimed at relieving the cost of moment estimate computation in Algorithm 6 which we call ADMU.

---

**Algorithm 6** Adam with Delayed Moment Updates (ADMU)

---

**Require:** $\eta_\ell$: Step size
**Require:** $z \in \mathbb{Z}_{\geq 1}$: Step delay for second moment estimate updates (where $z = 1$ gives no delay)
**Require:** $\beta_1, \beta_2 \in [0, 1)$: Exponential decay rates for the moment estimates
**Require:** $f(x)$: Stochastic objective function with parameters $x$
**Require:** $x_0$: Initial parameter vector
**Require:** $\varepsilon > 0$: Smoothing term
1: Initialize $m_0 \leftarrow 0$ (1st moment vector)
2: Initialize $v_0 \leftarrow 0$ (2nd moment vector)
3: Initialize $t \leftarrow 0$ (Timestep)
4: **while** not converged **do**
5:     $t \leftarrow t + 1$
6:     $g_t \leftarrow \nabla_x f_t(x_{t-1})$
7:     $m_t \leftarrow \beta_1 \cdot m_{t-1} + (1 - \beta_1) \cdot g_t$
8:     $\hat{m}_t \leftarrow m_t / (1 - \beta_1^t)$
9:     **if** $(t - 1)/z \in \mathbb{Z}$ **then**
10:       $v_t \leftarrow \beta_2 \cdot v_{t-1} + (1 - \beta_2) \cdot g_t^2$
11:       $\hat{v}_t \leftarrow v_t / (1 - \beta_2^{\lfloor \frac{t-1}{z} \rfloor + 1})$
12:     **else**
13:       $\hat{v}_t \leftarrow \hat{v}_{t-1}$
14:     **end if**
15:     $x_t \leftarrow x_{t-1} - \eta_\ell \cdot \hat{m}_t / (\sqrt{\hat{v}_t} + \varepsilon)$
16: **end while**
17: **return** $x_t$

---

Following [10], we provide an intuitive justification for the initialization bias correction employed in ADMU. Recall that the motivation for adaptive step-size in ADAM is updating the parameters via empirical estimates of the pseudo-gradient $\mathbb{E}[g]/\sqrt{\mathbb{E}[g^2]}$, which allows for both momentum and autonomous annealing near steady states. The square root is taken in the denominator to homogenize the degree of the gradient. Bias correction for ADMU adheres to the same principle, while requiring an additional assumption of gradient stabilization during the $z$-step preconditioner update delay. An equivalent formulation of the moment estimates in Algorithm 6 for general $t$ is given

$$m_t = m_0 \beta_1^t + (1 - \beta_1) \sum_{r=1}^{t} \beta_1^{t-r} \cdot g_r,$$

$$v_t = v_0 \beta_2^{\lfloor \frac{t-1}{z} \rfloor + 1} + (1 - \beta_2) \sum_{r=1}^{t} \beta_2^{\lfloor \frac{t-1}{z} \rfloor + 1 - \lceil \frac{r}{z} \rceil} \cdot g_{\lceil \frac{r}{z} \rceil z - z + 1} \odot g_{\lceil \frac{r}{z} \rceil z - z + 1} \cdot \chi_{\{\frac{r-1}{z} \in \mathbb{Z}_{\geq 0}\}}$$

$$= v_0 \beta_2^{\lfloor \frac{t-1}{z} \rfloor + 1} + (1 - \beta_2) \sum_{r=1}^{\lceil \frac{t}{z} \rceil} \beta_2^{\lceil \frac{t}{z} \rceil - r} g_{(r-1)z+1} \odot g_{(r-1)z+1}. \tag{17}$$

We work with $v_t$ as the proof for $m_t$ is analogous with $z = 1$. Assume that the gradients $g_1, \ldots, g_t$ are drawn from a latent gradient distribution $g_i \sim \widetilde{\mathcal{D}}(g_i)$. We aim to extract a relation between the expected delayed exponential moving average of the second moment $\mathbb{E}[v_t]$ and the true gradient expectation $\mathbb{E}[g_t^2]$. Taking expectation of both sides in equation (17),

$$\mathbb{E}[v_t] = v_0 \beta_1^{\lfloor \frac{t-1}{z} \rfloor + 1} + (1 - \beta_2) \sum_{r=1}^{\lceil \frac{t}{z} \rceil} \beta_2^{\lceil \frac{t}{z} \rceil - r} \mathbb{E}\left[g_{(r-1)z+1}^2\right]$$

$$\approx \zeta + (1 - \beta_2)\mathbb{E}\left[g_t^2\right] \sum_{r=1}^{\lceil \frac{t}{z} \rceil} \beta_2^{\lceil \frac{t}{z} \rceil - r}$$

$$\approx \mathbb{E}[g_t^2] \left(1 - \beta_1^{\lfloor \frac{t-1}{z} \rfloor + 1}\right).$$

Here, we have used zero initialization for the first moment estimate, while accumulating any error terms in $\zeta$. Several assumptions can lead to small $\zeta$. As in [10], we assume that $\beta_1$ is chosen small enough that the exponential moving average decay undermines the influence of non-recent gradients $g_i$ for $i < \left\lceil \frac{t}{z} \right\rceil z - z + 1$. A second assumption is that the latent gradient distribution remains stable during the $z$-step delay as training progresses, allowing the approximation $\mathbb{E}[g_t] \approx \mathbb{E}[g_{\left\lceil \frac{t}{z} \right\rceil z - z + 1}]$. This leaves the residual scaling of the true gradient second moment of the form $1 - \beta^\varphi$, which is caused by (zero) initialization as setting $v_0 = \mathbb{E}[g_t^2]$ eliminates $\beta^\varphi$. Therefore, bias correction is enforced by scaling the empirical $v_t$ estimate by the inverse. We note that $v_0$ need not be initialized to 0, in which case we should additionally translate $v_t$ by $-v_0 \beta_1^{\left\lfloor \frac{t-1}{z} \right\rfloor + 1}$ prior to the inverse scaling.

## F.1 Non-convex convergence analysis

---

**Algorithm 7** Adaptive server-side ADAGRAD and client-side ADAM (FedAdaAdam)

---

**Require:** Update delay step size $z \in \mathbb{Z}_{\geq 1}$, initializations $x_0, \widetilde{v}_0 \geq \tau^2$ and $\widetilde{m}_0 \leftarrow 0$
**Require:** Global and local decay parameters $\widetilde{\beta}_1, \widetilde{\beta}_2, \beta_1, \beta_2 \in [0, 1)$
**Require:** Pseudogradient weighting schedule $\Xi^1 \times \cdots \times \Xi^T \in \mathbb{R}^{|\mathcal{S}^1|} \times \cdots \times \mathbb{R}^{|\mathcal{S}^T|}$ for $\|\Xi^t\|_\infty \leq B$
**Require:** Client epoch schedule $\overline{K}^1 \times \cdots \times \overline{K}^T \in \mathbb{Z}_{\geq 1}^{|\mathcal{S}^1|} \times \cdots \times \mathbb{Z}_{\geq 1}^{|\mathcal{S}^T|}$ for $\|\overline{K}^t\|_\infty \leq K, \forall t \in [T]$
**Require:** Local epsilon smoothing term $\varepsilon_s > 0$
1: **for** $t = 1, \ldots, T$ **do**
2:     Sample subset $\mathcal{S}^t \subset [N]$ of clients
3:     **for** each client $i \in \mathcal{S}^t$ (in parallel) **do**
4:         $x_{i,0}^t \leftarrow x_{t-1}$
5:         Initialize $m_0, v_0 \geq 0$ with default values $m_0, v_0 \leftarrow 0$
6:         **for** $k = 1, \ldots, \overline{K}_i^t$ **do**
7:             Draw stochastic gradient $g_{i,k}^t \sim \mathcal{D}(x_{i,k-1}^t)$ with mean $\nabla F_i(x_{i,k-1}^t) \in \mathbb{R}^d$
8:             $m_k \leftarrow \beta_1 \cdot m_{k-1} + (1 - \beta_1) \cdot g_{i,k}^t$
9:             $\hat{m}_k \leftarrow m_k/(1 - \beta_1^k)$
10:           **if** $(k-1)/z \in \mathbb{Z}$ **then**
11:               $v_k \leftarrow \beta_2 \cdot v_{k-1} + (1 - \beta_2) \cdot g_{i,k}^t \odot g_{i,k}^t$
12:               $\hat{v}_k \leftarrow v_k/(1 - \beta_2^{\left\lfloor \frac{k-1}{z} \right\rfloor + 1})$
13:           **else**
14:               $v_k \leftarrow v_{k-1}$
15:           **end if**
16:           **if** $0 < \|\hat{m}_k/(\sqrt{\hat{v}_k} + \epsilon)\| < \varepsilon_s$ **then**
17:               $m_k \leftarrow 0$
18:           **end if**
19:           $x_{i,k}^t \leftarrow x_{i,k-1}^t - \eta_\ell \cdot \hat{m}_k/(\sqrt{\hat{v}_k} + \epsilon)$
20:         **end for**
21:         $\Delta_i^t = \Xi_i^t \left( x_{i, \overline{K}_i^t}^t - x_{t-1} \right)$
22:     **end for**
23:     $\Delta_t = \frac{1}{|\mathcal{S}^t|} \sum_{i \in \mathcal{S}^t} \Delta_i^t$
24:     $\widetilde{m}_t = \widetilde{\beta}_1 \widetilde{m}_{t-1} + (1 - \widetilde{\beta}_1)\Delta_t$
25:     $\widetilde{v}_t = \widetilde{v}_{t-1} + \Delta_t^2$
26:     $x_t = x_{t-1} + \eta \frac{\widetilde{m}_t}{\sqrt{\widetilde{v}_t} + \tau}$
27: **end for**

---

A description of FedAdaAdam is given as Algorithm 7. A few remarks are in order. Firstly, to allow for straggler mitigation, we allow the number of client $i$ epochs $\overline{K}_i^t$ at timestep $t$ to vary among the clients $i \in \mathcal{S}_i$. Although Algorithm 7 sets a schedule for client epochs and pseudogradient weights for clarity of exposition, dynamic allocation still allows the convergence proof to go through, as long as the schedule weights are bounded. By default, we set $\overline{K}^t = K$ and $\Xi^t = B = 1$ to avoid tuning a

large number of hyperparameters or having to sample from a client epoch count distribution for the client subsampling case.

Secondly, for the purposes of the proof we shall consider a local device to have been dropped and unsampled if any runs less than 1 epoch. We also enforce that pseudogradient weights are bounded positively from below, i.e. $\Xi_i^t > \varepsilon_w > 0$. We now provide a convergence bound for the general, non-convex case which holds for both full and partial client participation.

**Corollary F.1.** *For Algorithm 7, we have an identical bound to Theorem 4.1 with $\Psi_3, \Psi_4$ replaced by*

$$
\Psi_3 = \frac{(1 - \widetilde{\beta}_1^T)\eta\eta_\ell(1 - \beta_1^{2K})K\widetilde{L}B^2T\|\Phi_1^K\|^2}{2\widetilde{\alpha}_1\tau\varepsilon^2},
$$

$$
\Psi_4 = \frac{(1 - \widetilde{\beta}_1)\eta\eta_\ell(1 - \beta_1^{2K})KLTB^2c(\widetilde{\beta}_1)\|\Phi_2^K\|^2}{2\widetilde{\alpha}_1\tau\varepsilon^2}.
$$

*Here, the intermediary $\widetilde{\gamma}_1, \widetilde{\alpha}_1$ values are defined for $K^- := \min_{i,t} \overline{K}_i^t \geq 1$ as*

$$
\widetilde{\gamma}_1 := \eta_\ell\varepsilon_w \sum_{p=1}^{K^-} \frac{1 - \beta_1^p}{G\sqrt{1 - \beta_2^{\lceil \frac{p}{z} \rceil}} + \varepsilon}, \quad \widetilde{\alpha}_1 := \sum_{p=1}^{K^-} \frac{\varepsilon_w(1 - \beta_1^p)}{\left(G\sqrt{1 - \beta_2^{\lceil \frac{p}{z} \rceil}} + \varepsilon\right)(K+1)^2}.
$$

The proof is subsumed by or analogous to Theorems 4.1 and E.1, with changes summarized in the following lemma.

**Lemma F.2.** *Under Algorithm 7, $|\Delta_i^t|$ is bounded by*

$$
|\Delta_i^t| \leq \Phi_1^{\overline{K}_i^t} := |\Xi_i^t| \cdot \left( \eta_\ell \overline{K}_i^t \sqrt{\left( \sum_{r=1}^{\lceil \frac{\overline{K}_i^t}{z} \rceil} \frac{\beta_1^{2\lceil \frac{\overline{K}_i^t}{z} \rceil - 2r}}{\beta_2^{\lceil \frac{\overline{K}_i^t}{z} \rceil - r}} \right)} + \Phi_0^{\overline{K}_i^t} \right)
$$

*where*

$$
\Phi_0^{\overline{K}_i^t} := \frac{\overline{K}_i^t G\eta_\ell(1 - \beta_1^{\overline{K}_i^t})}{\varepsilon}.
$$

*Proof.* Recall that $\Delta_t = 1/|\mathcal{S}^t| \sum_{i \in \mathcal{S}^t} \Delta_i^t$ and $\Delta_i^t = \Xi_i^t \left( x_{i,\overline{K}_i^t}^t - x_{i,0}^t \right)$. By telescoping for $\overline{K}_i^t$ local steps and the definition of gradient updates in ADMU, we obtain

$$
\Delta_i^t = \sum_{p=1}^{\overline{K}_i^t} -\eta_\ell\Xi_i^t \frac{\hat{m}_p}{\sqrt{\hat{v}_p} + \varepsilon} = -\eta_\ell\Xi_i^t \sum_{p=1}^{\overline{K}_i^t} \frac{m_0\beta_1^p + (1 - \beta_1) \sum_{r=1}^p \beta_1^{p-r} \cdot g_{i,r}^t}{\sqrt{v_0\beta_2^{\lfloor \frac{p-1}{z} \rfloor + 1} + (1 - \beta_2) \sum_{r=1}^{\lceil \frac{p}{z} \rceil} \beta_2^{\lceil \frac{p}{z} \rceil - r}(g_{i,(r-1)z+1}^t)^2} + \varepsilon}
$$

We assume $m_0, v_0 \leftarrow 0$ for expository purposes, although $v_0 > 0$ also suffices for the analysis (ending in a slightly different $\Phi_1^{\overline{K}_i^t}$). This gives that

$$
\Delta_i^t = -\eta_\ell\Xi_i^t \sum_{p=1}^{\overline{K}_i^t} \frac{(1 - \beta_1) \sum_{r=1}^p \beta_1^{p-r} \cdot g_{i,r}^t}{\sqrt{(1 - \beta_2) \sum_{r=1}^{\lceil \frac{p}{z} \rceil} \beta_2^{\lceil \frac{p}{z} \rceil - r}(g_{i,(r-1)z+1}^t)^2} + \varepsilon}
$$

$$
= -\eta_\ell\Xi_i^t \sum_{p=1}^{\overline{K}_i^t} \frac{(1 - \beta_1) \sum_{r=1}^{\lceil \frac{p}{z} \rceil} \beta_1^{\lceil \frac{p}{z} \rceil - r} \cdot g_{i,(r-1)z+1}^t}{\sqrt{(1 - \beta_2) \sum_{r=1}^{\lceil \frac{p}{z} \rceil} \beta_2^{\lceil \frac{p}{z} \rceil - r}(g_{i,(r-1)z+1}^t)^2} + \varepsilon}
$$

$$
- \eta_\ell\Xi_i^t \sum_{p=1}^{\overline{K}_i^t} \frac{(1 - \beta_1) \sum_{r=1}^p \beta_1^{p-r} \cdot g_{i,r}^t \cdot \chi_{\left\{ \frac{p-1}{z} \notin \mathbb{Z} \right\}}}{\sqrt{(1 - \beta_2) \sum_{r=1}^{\lceil \frac{p}{z} \rceil} \beta_2^{\lceil \frac{p}{z} \rceil - r}(g_{i,(r-1)z+1}^t)^2} + \varepsilon}.
$$

To obtain a deterministic bound, we cannot ignore the worst-case stochastic realization that $g_{i,(r-1)z+1}^t = 0$ for $\forall r \in [\lceil \frac{p}{z} \rceil]$. Therefore, we form the intermediary upper bound

$$\left|\Delta_i^t\right| \le \eta_\ell |\Xi_i^t| \sum_{p=1}^{\overline{K}_i^t} \frac{(1-\beta_1) \sum_{r=1}^{\lceil \frac{p}{z} \rceil} \beta_1^{\lceil \frac{p}{z} \rceil - r} \cdot \left|g_{i,(r-1)z+1}^t\right|}{\sqrt{(1-\beta_2) \sum_{r=1}^{\lceil \frac{p}{z} \rceil} \beta_2^{\lceil \frac{p}{z} \rceil - r} (g_{i,(r-1)z+1}^t)^2 + \varepsilon}}$$
$$+ \frac{\eta_\ell |\Xi_i^t|(1-\beta_1)}{\varepsilon} \left( \sum_{p=1}^{\overline{K}_i^t} \sum_{r=1}^{p} \beta_1^{p-r} \cdot \left|g_{i,r}^t\right| \cdot \chi_{\left\{ \frac{p-1}{z} \notin \mathbb{Z} \right\}} \right). \quad (18)$$

Note that the first term is $0$ in the worst-case scenario above, which implies that any non-negative upper bound is trivially satisfied. Therefore, we may assume without loss of generality that at least one sampled gradient $g_{i,(r-1)z+1}^t$ is nontrivial and remove $\varepsilon$ from the denominator to obtain an upper bound. By Cauchy-Schwartz, we have

$$\left( \sum_{r=1}^{\lceil \frac{p}{z} \rceil} \beta_2^{\lceil \frac{p}{z} \rceil - r} (g_{i,(r-1)z+1}^t)^2 \right) \left( \sum_{r=1}^{\lceil \frac{p}{z} \rceil} \frac{\beta_1^{2\lceil \frac{p}{z} \rceil - 2r}}{\beta_2^{\lceil \frac{p}{z} \rceil - r}} \right) \ge \left( \sum_{r=1}^{\lceil \frac{p}{z} \rceil} \beta_1^{\lceil \frac{p}{z} \rceil - r} \cdot \left|g_{i,(r-1)z+1}^t\right| \right)^2$$

which implies

$$\left|\Delta_i^t\right| \le \eta_\ell |\Xi_i^t| \sum_{p=1}^{\overline{K}_i^t} \sqrt{\left( \sum_{r=1}^{\lceil \frac{p}{z} \rceil} \frac{\beta_1^{2\lceil \frac{p}{z} \rceil - 2r}}{\beta_2^{\lceil \frac{p}{z} \rceil - r}} \right)} + \frac{\eta_\ell |\Xi_i^t|(1-\beta_1)}{\varepsilon} \left( \sum_{p=1}^{\overline{K}_i^t} \sum_{r=1}^{p} \beta_1^{p-r} \cdot \left|g_{i,r}^t\right| \cdot \chi_{\left\{ \frac{p-1}{z} \notin \mathbb{Z} \right\}} \right)$$

$$\le \eta_\ell |\Xi_i^t| \sum_{p=1}^{\overline{K}_i^t} \sqrt{\left( \sum_{r=1}^{\lceil \frac{p}{z} \rceil} \frac{\beta_1^{2\lceil \frac{p}{z} \rceil - 2r}}{\beta_2^{\lceil \frac{p}{z} \rceil - r}} \right)} + \frac{\overline{K}_i^t G \eta_\ell |\Xi_i^t|(1-\beta_1)}{\varepsilon} \cdot \frac{(1-\beta_1^{\overline{K}_i^t})}{(1-\beta_1)}$$

$$\le \eta_\ell |\Xi_i^t| \overline{K}_i^t \sqrt{\left( \sum_{r=1}^{\lceil \frac{\overline{K}_i^t}{z} \rceil} \frac{\beta_1^{2\lceil \frac{\overline{K}_i^t}{z} \rceil - 2r}}{\beta_2^{\lceil \frac{\overline{K}_i^t}{z} \rceil - r}} \right)} + \frac{\overline{K}_i^t G \eta_\ell |\Xi_i^t|(1-\beta_1^{\overline{K}_i^t})}{\varepsilon}.$$

$\square$

It can be shown that case of no update delay $z = 1$ allows for $\Phi_0^{\overline{K}_i^t} = 0$, following a similar proof to the one given above. Note that $\Phi_0^{\overline{K}_i^t}$ handles the superfluous gradient terms cemented by delaying preconditioner updates for the second moment, while moving averaging is performed for the first moment estimate. It also follows that $\Delta_t$ is also upper bounded by the identical bound scaled by $\max_t \|\Xi^t\|_\infty \le B$, as the average of the $\Delta_i^t$.

## G   AdaGrad with Delayed Updates (AGDU)

We present AdaGrad with delayed preconditioner as Algorithm 8 for completeness.

Note that due to delayed updates, local gradient updates are not necessarily elementwise bounded in absolute value by $\eta_\ell$. We may expand the delayed updates for $v_t$ as

$$v_t = v_0 + \sum_{r=1}^{\lceil \frac{t}{z} \rceil} g_{(r-1)z+1} \odot g_{(r-1)z+1}.$$

We have the following convergence bound.

**Corollary G.1.** *Let $K^- := \min_{i,t} \overline{K}_i^t \ge 1$ and*

$$\widetilde{\gamma}_1 := \eta_\ell \varepsilon_w \sum_{p=1}^{K^-} \frac{1}{\sqrt{v_0 + \lceil \frac{K}{z} \rceil G^2} + \varepsilon}, \quad \widetilde{\alpha}_1 := \frac{\varepsilon_w K^-}{2K \left( \sqrt{v_0 + \lceil \frac{K}{z} \rceil G^2} + \varepsilon \right)}.$$

*Then Algorithm 9 has an identical convergence bound to Theorem 4.1.*

---

**Algorithm 8** AdaGrad with Delayed Updates (AGDU)

---

**Require:** $\eta_\ell$: Step size
**Require:** $z \in \mathbb{Z}_{\geq 1}$: Step delay for second moment estimate updates (where $z = 1$ gives no delay)
**Require:** $f(x)$: Stochastic objective function with parameters $x$
**Require:** $x_0$: Initial parameter vector
**Require:** $\varepsilon > 0$: Smoothing term
 1: Initialize $v_0 \leftarrow 0$ (2nd moment vector)
 2: Initialize $t \leftarrow 0$ (Timestep)
 3: **while** not converged **do**
 4:     $t \leftarrow t + 1$
 5:     $g_t \leftarrow \nabla_x f_t(x_{t-1})$
 6:     **if** $(t-1)/z \in \mathbb{Z}$ **then**
 7:         $v_t \leftarrow v_{t-1} + g_t^2$
 8:     **else**
 9:         $v_t \leftarrow v_{t-1}$
10:     **end if**
11:     $x_t \leftarrow x_{t-1} - \eta_\ell \cdot g_t/(\sqrt{v_t} + \varepsilon)$
12: **end while**
13: **return** $x_t$

---

---

**Algorithm 9** Adaptive server and client-side ADAGRAD (FedAdaAdagrad)

---

**Require:** Update delay step size $z \in \mathbb{Z}_{\geq 1}$, initializations $x_0, \widetilde{v}_0 \geq \tau^2$ and $\widetilde{m}_0 \leftarrow 0$
**Require:** Global decay parameter $\widetilde{\beta}_1 \in [0, 1)$
**Require:** Pseudogradient weighting schedule $\Xi^1 \times \cdots \times \Xi^T \in \mathbb{R}^{|\mathcal{S}^1|} \times \cdots \times \mathbb{R}^{|\mathcal{S}^T|}$ for $\|\Xi^t\|_\infty \leq B$
**Require:** Client epoch schedule $\overline{K}^1 \times \cdots \times \overline{K}^T \in \mathbb{Z}_{\geq 1}^{|\mathcal{S}^1|} \times \cdots \times \mathbb{Z}_{\geq 1}^{|\mathcal{S}^T|}$ for $\|\overline{K}^t\|_\infty \leq K, \forall t \in [T]$
**Require:** Local epsilon smoothing term $\varepsilon_s > 0$, global smoothing term $\tau > 0$
 1: **for** $t = 1, \ldots, T$ **do**
 2:     Sample subset $\mathcal{S}^t \subset [N]$ of clients
 3:     **for** each client $i \in \mathcal{S}^t$ (in parallel) **do**
 4:         $x_{i,0}^t \leftarrow x_{t-1}$
 5:         Initialize $v_0 \geq 0$ with default value $v_0 \leftarrow 0$
 6:         **for** $k = 1, \ldots, \overline{K}_i^t$ **do**
 7:             Draw stochastic gradient $g_{i,k}^t \sim \mathcal{D}(x_{i,k-1}^t)$ with mean $\nabla F_i(x_{i,k-1}^t) \in \mathbb{R}^d$
 8:             $m_k \leftarrow g_{i,k}^t$
 9:             **if** $(k-1)/z \in \mathbb{Z}$ **then**
10:                 $v_k \leftarrow v_{k-1} + g_{i,k}^t \odot g_{i,k}^t$
11:             **else**
12:                 $v_k \leftarrow v_{k-1}$
13:             **end if**
14:             **if** $0 < \|m_k/(\sqrt{v_k} + \epsilon)\| < \varepsilon_s$ **then**
15:                 $m_k \leftarrow 0$
16:             **end if**
17:             $x_{i,k}^t \leftarrow x_{i,k-1}^t - \eta_\ell \cdot m_k/(\sqrt{v_k} + \epsilon)$
18:         **end for**
19:         $\Delta_i^t = \Xi_i^t \left( x_{i,\overline{K}_i^t}^t - x_{t-1} \right)$
20:     **end for**
21:     $\Delta_t = \frac{1}{|\mathcal{S}^t|} \sum_{i \in \mathcal{S}^t} \Delta_i^t$
22:     $\widetilde{m}_t = \widetilde{\beta}_1 \widetilde{m}_{t-1} + (1 - \widetilde{\beta}_1)\Delta_t$
23:     $\widetilde{v}_t = \widetilde{v}_{t-1} + \Delta_t^2$
24:     $x_t = x_{t-1} + \eta \frac{\widetilde{m}_t}{\sqrt{\widetilde{v}_t} + \tau}$
25: **end for**

---

Similar to delayed Adam, the proof is analogous to Theorem 4.1 with changes summarized in the following lemma.

**Lemma G.2.** *Under Algorithm 9, $|\Delta_i^t|$ is bounded by*

$$|\Delta_i^t| \leq \Phi_1^K := \eta_\ell B \left( \left\lfloor \frac{K-1}{z} \right\rfloor + 1 + \frac{KG}{\sqrt{v_0} + \varepsilon} \right).$$

*Proof.* Recall that $\Delta_t = 1/|\mathcal{S}^t| \sum_{i \in \mathcal{S}^t} \Delta_i^t$ and $\Delta_i^t = \Xi_i^t \left( x_{i,\overline{K}_i^t}^t - x_{i,0}^t \right)$. By telescoping for $\overline{K}_i^t$ local steps and the definition of gradient updates in FedAdAdagrad, we obtain

$$\Delta_i^t = \sum_{p=1}^{\overline{K}_i^t} -\eta_\ell \Xi_i^t \frac{m_p}{\sqrt{v_p} + \varepsilon} = -\eta_\ell \Xi_i^t \sum_{p=1}^{\overline{K}_i^t} \frac{g_{i,p}^t}{\sqrt{v_0 + \sum_{r=1}^{\lceil \frac{p}{z} \rceil} (g_{i,(r-1)z+1}^t)^2} + \varepsilon}$$

For $\mathcal{F} = \{0, 1, \ldots, \lfloor (\overline{K}_i^t - 1)/z \rfloor\} z + 1$, we thus have that

$$\Delta_i^t = -\eta_\ell \Xi_i^t \sum_{p \in \mathcal{F}} \frac{g_{i,p}^t}{\sqrt{v_0 + \sum_{r=1}^{\lceil \frac{p}{z} \rceil} (g_{i,(r-1)z+1}^t)^2} + \varepsilon}$$

$$- \eta_\ell \Xi_i^t \sum_{p \in [\overline{K}_i^t] \setminus \mathcal{F}} \frac{g_{i,p}^t}{\sqrt{v_0 + \sum_{r=1}^{\lceil \frac{p}{z} \rceil} (g_{i,(r-1)z+1}^t)^2} + \varepsilon}.$$

To obtain a deterministic bound, we cannot ignore the worst-case stochastic realization that $g_{i,(r-1)z+1}^t = 0$ for $\forall r \in [\lceil \frac{p}{z} \rceil]$. Therefore, we form the upper bound

$$\left| \Delta_i^t \right| \leq \eta_\ell |\Xi_i^t| \sum_{p \in \mathcal{F}} \frac{|g_{i,p}^t|}{\sqrt{v_0 + |g_{i,p}^t|^2 + \sum_{r=1}^{\lceil \frac{p}{z} \rceil - 1} (g_{i,(r-1)z+1}^t)^2} + \varepsilon}$$

$$+ \frac{\eta_\ell |\Xi_i^t|}{\sqrt{v_0} + \varepsilon} \left( \sum_{p \in [\overline{K}_i^t] \setminus \mathcal{F}} |g_{i,p}^t| \right) \tag{19}$$

$$\leq \eta_\ell |\Xi_i^t| \left( \left\lfloor \frac{K-1}{z} \right\rfloor + 1 \right) + \frac{\eta_\ell |\Xi_i^t| KG}{\sqrt{v_0} + \varepsilon}$$

where the last line uses that the local epoch schedules are upper bounded by $K$. Noting that $\|\Xi_i^t\|_\infty \leq B$, we are done. □

## H   Datasets, Models, and Baselines

Below, we summarize the dataset statistics and provide a more in-depth description.

Table 2: Summary of datasets and models.

| Datasets | # Devices | Non-IID Partition | Model | Tasks |
|---|---|---|---|---|
| StackOverflow [27] | 400 | Natural | Logistic Regression | 500-Class Tag Classification |
| CIFAR-100 [28] | 1000 | LDA | ViT-S | 100-Class Image Classification |
| GLD-23K [30] | 233 | Natural | ViT-S | 203-Class Image Classification |
| FEMNIST [29] | 500 | Natural | ViT-S | 62-Class Image Classification |

### H.1   StackOverflow Dataset

The StackOverflow dataset [27] is a language dataset composed of questions and answers extracted from the StackOverflow online community. Each data entry includes associated metadata such as tags (e.g., "python"), the time the post was created, the title of the question, the score assigned to the question, and the type of post (question or answer). The dataset is partitioned by users, with each

client representing an individual user and their collection of posts. This dataset exhibits significant imbalance, with some users contributing only a few posts while others have a much larger number of entries. In this paper, we work with a randomly selected 400-client subset of the full StackOverflow Dataset, with a client participation fraction of $0.1$.

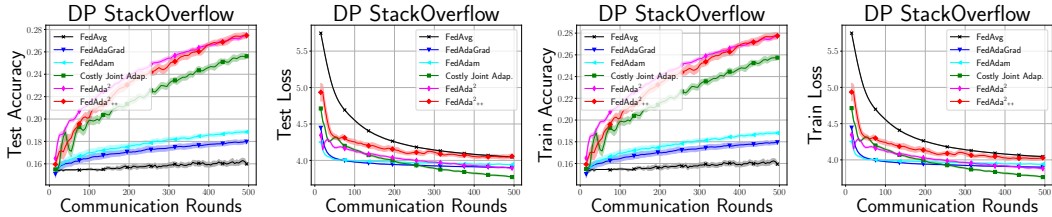

Figure 5: Identical setting as Figure 1, except that 1 local epoch is taken instead of 5. Additionally, the adaptive baseline algorithms employ the Adam optimizer instead of AdaGrad. We observe that jointly adaptive methods significantly outperform server-only adaptive methods, which in turn outperform non-adaptive FedAvg. The setting studies the noise multiplier $\sigma = 1$, with a privacy budget of $(\varepsilon, \delta) = (13.1, 0.0025)$ with optimal Rényi-Differential Privacy (RDP) [39] order 2.0.

## H.2    GLD-23K Dataset

The GLD-23K dataset is a subset of the GLD-160k dataset introduced in [30]. It contains 23,080 training images, 203 landmark labels, and 233 clients. Compared to CIFAR-10/100, the landmarks dataset consists of images of far higher quality and resolution, and therefore represents a more challenging learning task. The client participation fraction for all GLD-23K experiments are set to $0.01$.

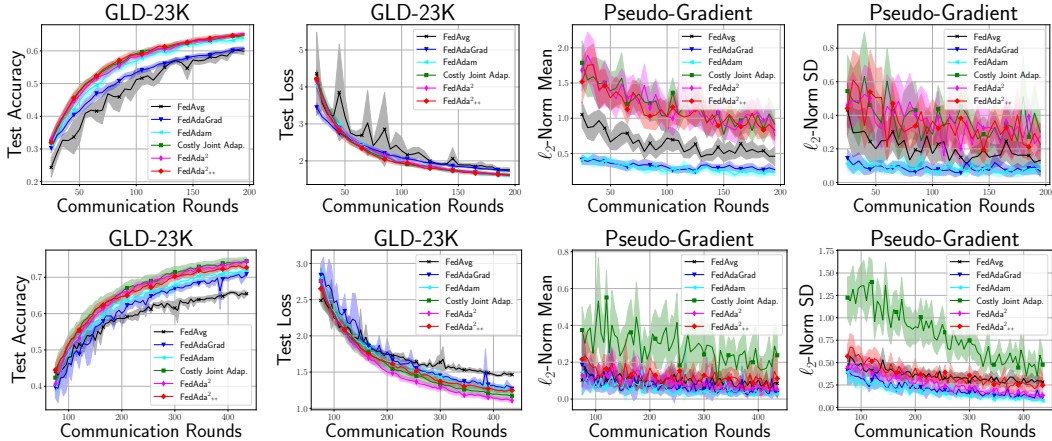

Figure 6: (Top) Additional results for the experiments in Figure 4 (b), where clients train over 5 epochs. (Bottom) Analogous experiments for full fine-tuning, where the entire net is unfrozen after replacing the classification layer. All adaptive optimizers are instantiated with Adam, with the exception of FedAdaGrad where the server-side adaptive optimizer is AdaGrad.

## H.3    CIFAR-100 Dataset

The CIFAR-10/100 datasets [28] consist of $32 \times 32 \times 3$ images. In the smaller variant CIFAR-10, there are 10 labels, with 50,000 training images and 10,000 test images. The 10 classes represent common objects: airplanes, automobiles, birds, cats, deer, dogs, frogs, horses, ships, and trucks. CIFAR-100 is meant to be an extension of CIFAR-10, consisting of 60,000 color images, but with 100 classes instead of 10. Each class in CIFAR-100 contains 600 images, and the dataset is similarly split into 50,000 training images and 10,000 test images. Unlike CIFAR-10, every class in CIFAR-100 is subsumed by one of 20 superclasses, and each image is provided a fine label and a coarse label that represents the former and latter (super-)class. In this paper, we train and evaluate all algorithms

against the fine label. In Figure 7, we show the convergence of FedAda$^2$ as compared to all other adaptive or non-adaptive benchmarks using CIFAR-100.

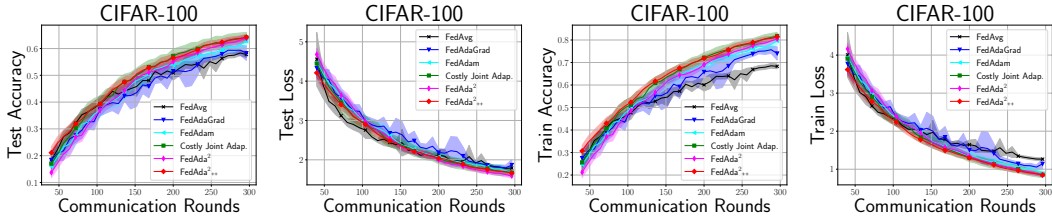

Figure 7: Training and testing accuracies of optimal hyperparameters for CIFAR-100. At each logging step, train/test accuracy and loss evaluation is done over *all* of training and testing data, disjointly, resulting in robust and similar-looking curves. Averaged over 20 random seeds for better convergence. Adaptive optimizer instantiation conventions are identical with Figure 6. Though minimal, jointly adaptive baselines (Costly Joint Adaptivity, FedAda$^2$, FedAda$^2$++) outperform server-only adaptive baselines (FedAdam, FedAdaGrad) and non-adaptive FedAvg.

## H.4    FEMNIST Dataset

The FEMNIST dataset [29] extends the MNIST dataset [50] to include both digits and letters, comprising 62 unbalanced classes and a total of 805,263 data points. It is specifically designed for federated learning research, featuring a natural, non-IID partitioning of data. Each user in the dataset corresponds to a distinct writer who contributed to the original EMNIST dataset, capturing the individuality of handwriting styles. This user-level segmentation provides a realistic federated learning setting, simulating scenarios where data is distributed heterogeneously across clients. FEMNIST serves as a benchmark for evaluating the performance of federated learning algorithms under non-IID conditions, emphasizing challenges such as personalization and robustness to client heterogeneity.

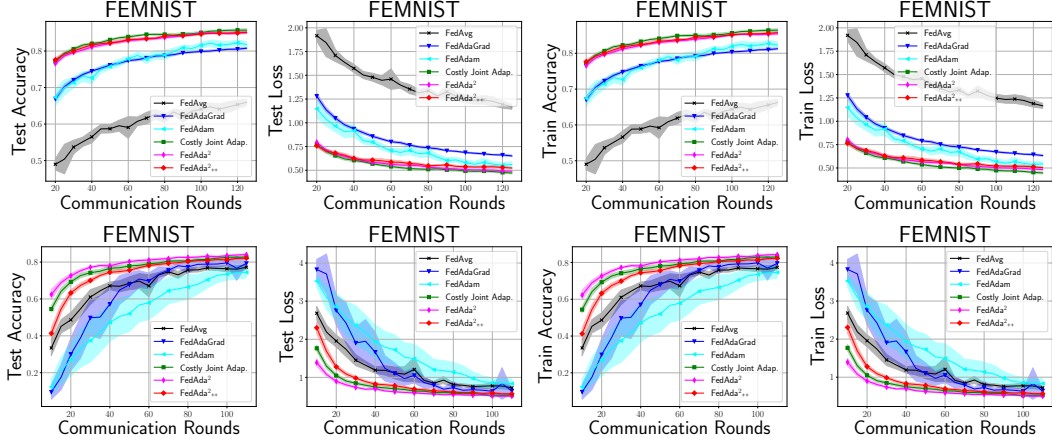

Figure 8: Training and testing accuracies of optimal hyperparameters for FEMNIST, with $0.5\%$ participation (2 clients per round). Averaged over 20 random seeds for clearer convergence. Adaptive optimizer instantiation conventions are identical with Figure 6, where jointly adaptive optimizing paradigms use Adam due to better performance. We see that FedAvg is the least robust, both in terms of stability (i.e., confidence interval region) and final performance. By contrast, adding server-side adaptivity greatly strengthens the performance, and introducing client-side adaptive optimization further enhances the speed of convergence as well as test-time accuracy. We see that removing preconditioner transmission, and compressing client-side gradient statistics to save on-device memory as in FedAda$^2$, does not detract from the performance of joint adaptivity. The top row displays the results when each client takes 5 local epochs prior to server synchronization, where the gap between the jointly adaptive and non-adaptive baselines is more pronounced. The bottom row gives the results for 1 local epoch.

## H.5 Descriptions of Baselines

In the original FedAvg algorithm introduced by [1], the server-side aggregation is performed without any additional momentum, relying solely on simple averaging. On the other hand, algorithms like FedAdaGrad and FedAdam represent examples of server-only adaptive approaches [4], where the server employs adaptive optimizers such as AdaGrad or Adam instead of vanilla averaging. We note that server-only adaptive frameworks such as FedAdam and FedAdaGrad are optimizer-specific instantiations of FedOpt [4], a competitive framework that has been utilized in recent works to develop leading applications (e.g., by Google Deepmind to develop DiLoCo [51, 52, 53]). The concept of 'Costly Joint Adaptivity' (Costly Joint Adap.) refers to a training paradigm where the server's adaptive preconditioners are shared with clients during each communication round. An example of this is the AdaGrad-AdaGrad setup used as a differential privacy baseline in the StackOverflow task, where the server-side AdaGrad preconditioners are applied to client-side AdaGrad optimizers, guiding client model updates.

Alternatively, by eliminating the transmission of server-side preconditioners and initializing client-side preconditioners to zero, we derive the 'Joint Adaptivity without Preconditioner Communication' (Joint Adap. w/o Precond. Commu.) baseline, which is more communication-efficient. Further, compressing local preconditioners to align with client memory constraints leads to the development of $\texttt{FedAda}^2$. Thus, $\texttt{FedAda}^2$ and the various baselines can be viewed as logically motivated extensions, incorporating adaptive updates and memory-efficient strategies. We provide comprehensive evaluations of all 15 algorithms (including 12 jointly adaptive methods tailored to each adaptive optimizer, 2 server-only adaptive methods, and 1 non-adaptive method) in Section 5 and in the Appendix H, J.

For the ViT model for instance, we require just 0.48% memory to store the second moment EMA compared to the full gradient statistic during preconditioning when using SM3.

# I Hyperparameter Selection

## I.1 Hyperparameters for DP StackOverflow

We use a subsampling rate of 0.1, for a total of 400 clients and 500 communication rounds. We investigate the setting of noise multiplier $\sigma = 1$, which provides a privacy budget of $(\varepsilon, \delta) = (13.1, 0.0025)$ with optimal Rényi-Differential Privacy (RDP) order 2.0. We sweep over the following hyperparameters:

$$c \in \{0.1, 0.5, 1\},$$
$$\eta_l \in \{0.001, 0.01, 0.1, 0.5, 1\},$$
$$\eta_s \in \{0.001, 0.01, 0.1, 0.5, 1\},$$
$$\tau_l \in \{10^{-7}, 10^{-5}, 10^{-3}\},$$
$$\tau_s \in \{10^{-7}, 10^{-5}, 10^{-3}\},$$

where $c$ is the gradient clip value. Here, $\eta_l, \eta_s$ indicates the client and server learning rates, while $\tau_l, \tau_s$ represents their respective adaptivity parameters. In the case of singular adaptivity, we ignore the irrelevant terms (i.e. client adaptivity parameter for FedAdaGrad). For FedAvg only, we select best hyperparameters using the expanded local learning rate grid

$$\eta_l \in \{0.001, 0.01, 0.1, 0.5, 1, 5, 20, 40, 80, 160\}.$$

The optimal hyperparameters are summarized in Table 3, which were chosen based on optimal test accuracy over a running average of the last 10 logged datapoints. In Figure 4 (bottom), we see that adaptive optimization on either the client or server induces varying model training dynamics. Notably, we see in our experiments that for this privacy budget, removing preconditioners from jointly adaptive systems supercedes the performance of costly joint adaptivity. Compressing client adaptive preconditioning ($\texttt{FedAda}^2$) reduces the performance slightly, but still performs the best among all other baselines.

Table 3: Best performing hyperparameters for DP StackOverflow with $\sigma = 1$

|        | FedAvg | FedAdaGrad | Costly Joint Adap. | Joint Adap. w/o Precond. Commu. | FedAda$^2$ |
|--------|--------|------------|--------------------|----------------------------------|------------|
| $c$    | 1.0    | 0.1        | 0.5                | 0.5                              | 0.1        |
| $\eta_s$ | N/A  | 1.0        | 1.0                | 1.0                              | 1.0        |
| $\eta_l$ | 20.0 | 1.0        | 1.0                | 0.1                              | 0.1        |
| $\tau_s$ | N/A  | 1e-3       | 1e-3               | 1e-5                             | 1e-5       |
| $\tau_l$ | N/A  | N/A        | 1e-3               | 1e-3                             | 1e-3       |

## I.2   Hyperparameters for Image Datasets

For all ViT experiments, images were resized to $224 \times 224$ pixels, and the client optimizer employed a linear learning rate warm-up, increasing from 0 to the final value over the first 10 local backpropagation steps. The local batch size was consistently set to 32 across all datasets used in this paper. Due to better empirical performance, Adam was selected as the main optimizer strategy for ViT fine-tuning against the image datasets. We utilized prior work [4] as well as small-scale experiments regarding server-only adaptivity to guide the selection of the momentum parameters $\beta_1 = 0.9$, $\beta_2 = 0.999$ for server Adam. The identical parameters were selected for client Adam, and better choices may exist for either the server or client. In order to determine suitable learning rates and adaptivity parameters, we conduct extensive hyperparameter sweeps using a two-step procedure.

**(Step 1)**   The first step involved a symmetric sweep over the values

$$\eta_l \in \{0.001, 0.01, 0.1, 0.5, 1, 5, 20\},$$
$$\eta_s \in \{0.001, 0.01, 0.1, 0.5, 1, 5, 20\},$$
$$\tau_l \in \{10^{-9}, 10^{-7}, 10^{-5}, 10^{-3}\},$$
$$\tau_s \in \{10^{-9}, 10^{-7}, 10^{-5}, 10^{-3}\}.$$

Similar to the StackOverflow case, $\eta_l, \eta_s$ indicates the client and server learning rates, while $\tau_l, \tau_s$ represents their respective adaptivity parameters. For FedAvg only, we probe over the expanded grid

$$\eta_l \in \{0.001, 0.01, 0.1, 0.5, 1, 5, 20, 40, 80, 160, 320\}.$$

**(Step 2)**   Based on the sweep results over all 10 algorithm and dataset combinations, a second asymmetric search was launched over the most promising hyperparameter regions, which probed over the following:

$$\eta_l \in \{10^{-6}, 10^{-5}, 10^{-4}, 10^{-3}, 10^{-2}, 10^{-1}\},$$
$$\eta_s \in \{10^{-7}, 10^{-6}, 10^{-5}, 10^{-4}, 10^{-3}, 10^{-2}\},$$
$$\tau_l \in \{10^{-7}, 10^{-5}, 10^{-3}, 10^{-1}, 1\},$$
$$\tau_s \in \{10^{-12}, 10^{-11}, 10^{-10}, 10^{-9}, 10^{-5}\}.$$

Afterwards, the best performing hyperparameters were selected. For FedAvg only, the final grid increased additively by $10^{-3}$ from $10^{-3}$ to $10^{-2}$, then by $10^{-2}$ onward until the largest value $10^{-1}$. That is, we sweep over the following:

$$\eta_l \in \{0.001, 0.002, 0.003, \ldots, 0.009, 0.01, 0.02, \ldots, 0.09, 0.1\}.$$

For server-only adaptivity or FedAvg, any irrelevant hyperparameters were ignored during the sweep. In Tables 4 and 5, we summarize the best performing learning rates and adaptivity parameters. In this subsection, any notion of adaptivity in jointly adaptive systems refers to the Adam optimizer, and 5 local epochs were taken prior to server synchronization. Full fine-tuning indicates that the entire net was unfrozen after replacement of the linear classification layer. For FedAdaGrad, full fine-tuning, Step 2 utilized an expanded hyperparameter grid search due to poor performance.

Table 4: Server/Client Learning Rates $\eta_s / \eta_l$

|                | FedAvg | FedAdaGrad | FedAdam | Costly Joint Adap. | Joint Adap. w/o Precond. Commu. | FedAda$^2$ |
|----------------|--------|------------|---------|--------------------|----------------------------------|------------|
| FEMNIST        | N/A / 8e-3 | 1e-4 / 1e-3 | 1e-4 / 1e-3 | 1e-3 / 1e-3 | 1e-3 / 1e-3 | 1e-3 / 1e-3 |
| CIFAR-100      | N/A / 1e-1 | 1e-2 / 1e-5 | 1e-3 / 1e-3 | 1e-3 / 1e-2 | 1e-3 / 1e-2 | 1e-3 / 1e-2 |
| GLD-23K        | N/A / 0.04 | 1e-2 / 1e-2 | 1e-3 / 1e-2 | 1e-3 / 1e-2 | 1e-3 / 1e-2 | 1e-3 / 1e-2 |
| GLD-23K (Full) | N/A / 0.02 | 1e-4 / 1e-2 | 1e-4 / 1e-2 | 1e-4 / 1e-4 | 1e-4 / 1e-2 | 1e-4 / 1e-4 |

Table 5: Server/Client Adaptivity Parameters $\tau_s/\tau_l$

|  | FedAvg | FedAdaGrad | FedAdam | Costly Joint Adap. | Joint Adap. w/o Precond. Commu. | FedAda$^2$ |
|---|---|---|---|---|---|---|
| FEMNIST | N/A / N/A | 1e-7 / N/A | 1e-7 / N/A | 1e-5 / 1e-7 | 1e-5 / 1e-7 | 1e-5 / 1e-7 |
| CIFAR-100 | N/A / N/A | 1e-10 / N/A | 1e-5 / N/A | 1e-5 / 1.0 | 1e-5 / 1.0 | 1e-5 / 1.0 |
| GLD-23K | N/A / N/A | 1e-5 / N/A | 1e-5 / N/A | 1e-5 / 0.1 | 1e-5 / 0.1 | 1e-5 / 0.1 |
| GLD-23K (Full) | N/A / N/A | 1e-2 / N/A | 1e-5 / N/A | 1e-5 / 1e-3 | 1e-5 / 1 | 1e-5 / 1e-3 |

**Hyperparameter Sweep for FEMNIST.** The setup was almost analogous to above. The only difference is that due to limited resources in **(Steps 1-2)**, we swept over the grid

$$\eta_l \in \left\{ 10^{-4}, 10^{-3}, 10^{-2}, 10^{-1} \right\},$$
$$\eta_s \in \left\{ 10^{-4}, 10^{-3}, 10^{-2}, 10^{-1} \right\},$$
$$\tau_l \in \left\{ 10^{-7}, 10^{-5}, 10^{-3} \right\},$$
$$\tau_s \in \left\{ 10^{-7}, 10^{-5}, 10^{-3} \right\}.$$

For FedAvg only, we utilized the expanded learning rate grid

$$\eta_l \in \{0.001, 0.002, 0.003, \ldots, 0.009, 0.01, 0.02, \ldots, 0.09, 0.1\}.$$

**Hyperparameters for varying client resources, GLD-23K.** Analogous sweeps as in **(Step 1)** above for the limited and sufficient client resource settings (locally training over 1, 20 local epochs prior to server synchronization) were taken. For the constrained setting, there were no changes to the **(Step 2)** grid. In the abundant setting, the modified final search space for adaptive methods was

$$\eta_l \in \left\{ 10^{-6}, 10^{-5}, 10^{-4}, 10^{-3}, 10^{-2}, 10^{-1} \right\},$$
$$\eta_s \in \left\{ 10^{-3}, 10^{-2}, 10^{-1}, 1, 4, 16, 32 \right\},$$
$$\tau_l \in \left\{ 10^{-7}, 10^{-5}, 10^{-3}, 10^{-1}, 1 \right\},$$
$$\tau_s \in \left\{ 10^{-12}, 10^{-11}, 10^{-10}, 10^{-9}, 10^{-5} \right\},$$

and the optimal hyperparameters are summarized in Table 6.

Table 6: Hyperparameters for GLD-23K under restricted/sufficient client resource settings

|  | FedAvg | FedAdaGrad | FedAdam | Costly Joint Adap. | Joint Adap. w/o Precond. Commu. | FedAda$^2$ |
|---|---|---|---|---|---|---|
| $\eta_s$ | N/A / N/A | 1e-2 / 1e-2 | 1e-3 / 1e-3 | 1e-3 / 1e-3 | 1e-3 / 1e-3 | 1e-3 / 1e-3 |
| $\eta_l$ | 7e-2 / 1e-2 | 1e-2 / 1e-2 | 1e-1 / 1e-2 | 1e-2 / 1e-3 | 1e-2 / 1e-3 | 1e-1 / 1e-3 |
| $\tau_s$ | N/A / N/A | 1e-9 / 1e-7 | 1e-5 / 1e-7 | 1e-5 / 1e-7 | 1e-5 / 1e-7 | 1e-5 / 1e-7 |
| $\tau_l$ | N/A / N/A | N/A / N/A | N/A / N/A | 1e-3 / 1e-1 | 1e-3 / 1e-1 | 1e-1 / 1e-1 |

## I.3 Compute Resources

Experiments were performed on a computing cluster managed by Slurm, consisting of nodes with various configurations. The cluster includes nodes with multiple GPU types, including NVIDIA RTX 2080 Ti, A40, and H100 GPUs.

## J  Additional Experiments

### J.1  Sensitivity to Hyperparameters

### J.2  Dynamics of Heterogeneous Client-Server Adaptivity

In Figure 10, we display the effects of heterogeneous client-server adaptivity in the setting of ViT fine-tuning over GLD-23K. All hyperparameter sweeps were done over the following grid:

$$\eta_l \in \left\{ 10^{-4}, 10^{-3}, 10^{-2}, 10^{-1} \right\},$$
$$\eta_s \in \left\{ 10^{-4}, 10^{-3}, 10^{-2}, 10^{-1} \right\},$$
$$\tau_l \in \left\{ 10^{-7}, 10^{-5}, 10^{-3}, 10^{-1}, 1 \right\},$$
$$\tau_s \in \left\{ 10^{-7}, 10^{-5}, 10^{-3}, 10^{-1}, 1 \right\}. \tag{20}$$

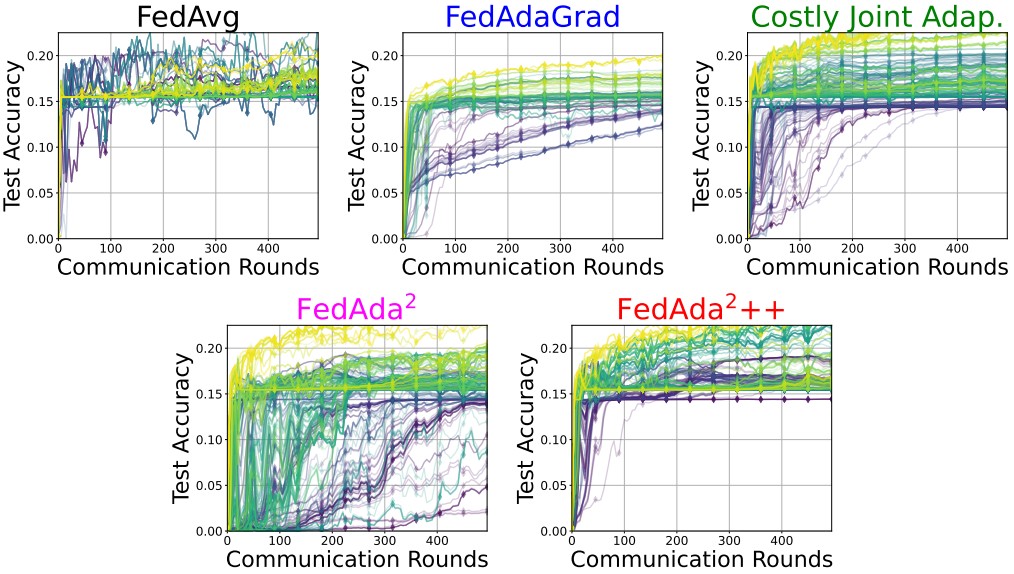

Figure 9: We plot all test accuracies obtained during the hyperparameter sweeps detailed in Appendix I.1, with fixed client subsampling random seed. The runs are ranked hierarchically from the lowest to the highest final test loss, with the colors transitioning from lighter to darker shades accordingly.

### J.3   Effect of Delayed Updates

Similar to Figure 10, we demonstrate the effects of delayed updates in Figure 11. Hyperparameter configuration for delayed updates is identical to Figure 4 (b), except that client-side preconditioner updates are delayed. Hyperparameter sweeps were done over the following grid:

$$\eta_l \in \left\{10^{-4}, 10^{-3}, 10^{-2}, 10^{-1}\right\},$$
$$\eta_s \in \left\{10^{-4}, 10^{-3}, 10^{-2}, 10^{-1}\right\},$$
$$\tau_l \in \left\{10^{-3}, 10^{-1}, 1\right\},$$
$$\tau_s \in \left\{10^{-5}, 10^{-3}, 10^{-1}\right\}.$$

We see that delaying the computation of the preconditioners does not significantly degrade the performance.

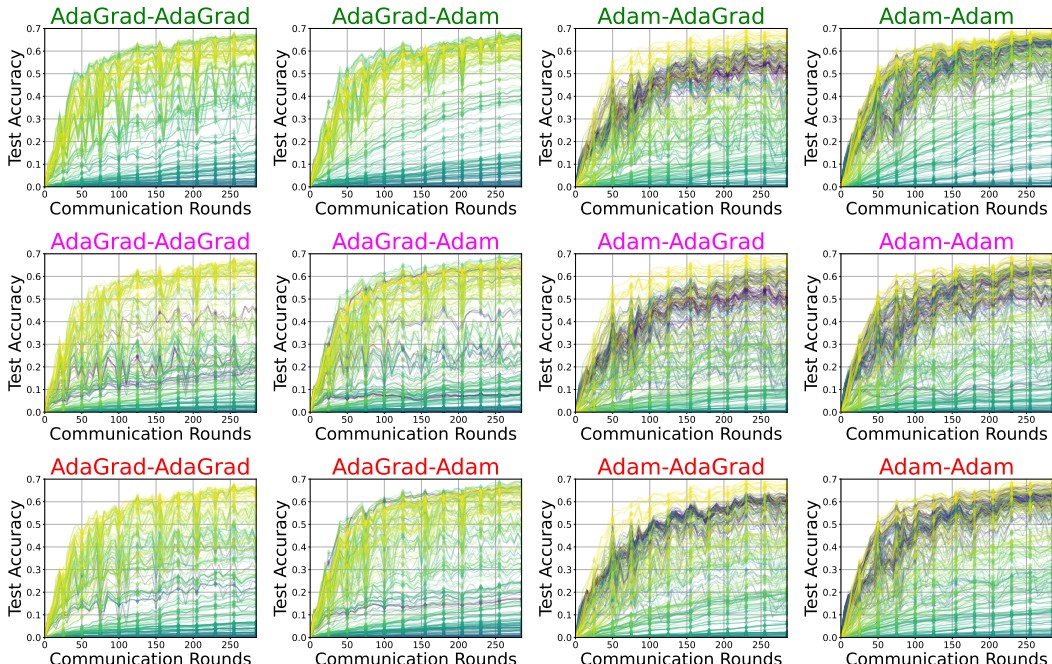

Figure 10: Each test accuracy is color-coded and ranked based on the final test loss, and lighter colors indicate lower loss. Algorithm title colors are also consistent with labels; green for Costly Joint Adaptivity (top), magenta for Joint Adaptivity without Preconditioner Transmission (middle), and red for FedAda$^2$ (bottom). Title ordering indicates server- and client-side optimizers, respectively; i.e. AdaGrad-Adam uses server AdaGrad and client Adam. In the case of Costly Joint Adaptivity with heterogeneous client-server optimizers, we transmit the *mismatched* server-side preconditioner to the client, which to our surprise demonstrates considerable performance. For FedAda$^2$, we add SM3 compression to the client-side optimizer after zero initialization of the local preconditioner.

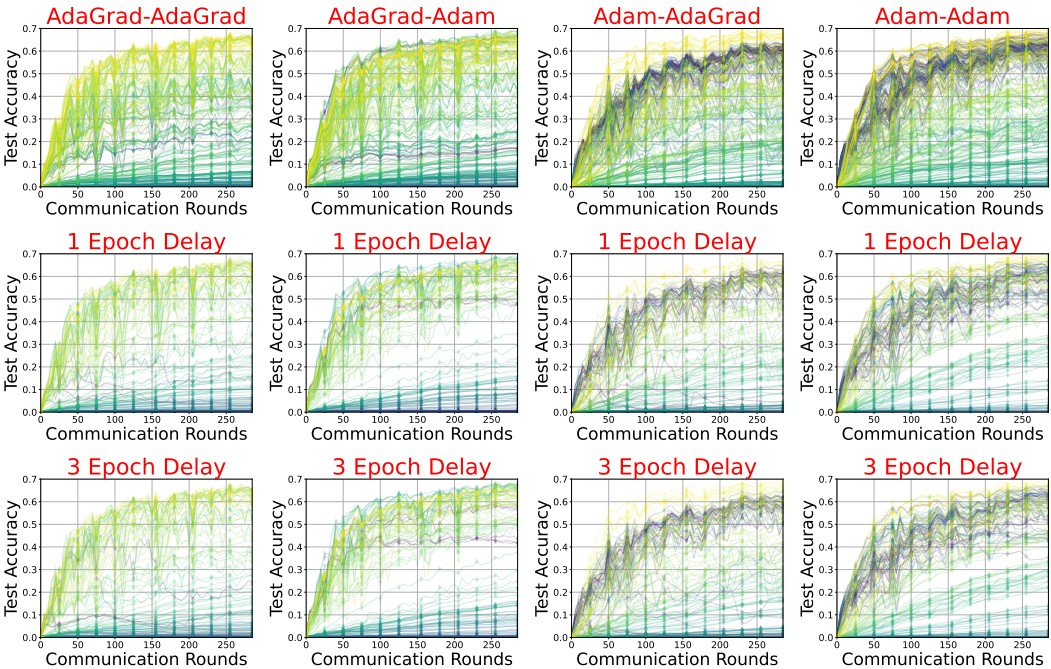

Figure 11: After updating preconditioners per every local backpropagation step for the first client epoch, preconditioners are periodically frozen for the next 1 (middle), 3 (bottom) epochs, respectively, for each communication round. Algorithms are consistent across columns, and the top row is identical to the FedAda$^2$ results in Figure 10 with hyperparameter sweep (20).

