# OpenReview forum: "Efficient Adaptive Federated Optimization"
_NeurIPS.cc/2025/Conference — NeurIPS 2025 poster_

### Official Review · Reviewer_T2Br · 2025-06-23

**Clarity:** 3
**Significance:** 3
**Originality:** 4
**Rating:** 5
**Confidence:** 3

**Summary:**

This paper proposes FedAda 2 and FedAda 2++, two communication- and memory-efficient adaptive algorithms for federated learning that retain the benefits of joint adaptivity while achieving the same convergence rates as more resource-intensive methods, with strong empirical performance across image and text tasks.

**Questions:**

Since the proposed methods eliminate the need for communicating preconditioners, one would expect a significant reduction in communication cost, as reflected in Figure 2. However, this improvement is not clearly observed—could the authors clarify the reason for this discrepancy?

**Ethical Concerns:**

["NO or VERY MINOR ethics concerns only"]

**Final Justification:**

I justify the current score based on my review.

**Limitations:**

No significant limitation.

**Paper Formatting Concerns:**

No concern.

**Quality:**

3

**Strengths And Weaknesses:**

### Strengths:
The paper demonstrates that communicating preconditioners in adaptive optimization is unnecessary. This is an interesting finding, as it enables both memory and communication efficiency. Moreover, the approach does not compromise convergence performance, making it a promising direction for future research.


### Weaknesses
1. Without an aggregated or communicated preconditioner, the server must compute the preconditioner independently. This can introduce additional computational and memory overhead, especially since some prior work often assumes that the server performs only minimal computations.

2. Although the authors note that the bounded gradients assumption has been used in prior work, it remains a strong assumption given the potential variability of local objectives. It would be more exciting to explore results under weaker or more realistic assumptions.

---

> ### Author Rebuttal · Authors · 2025-07-31
>
> We thank Reviewer T2Br for their thoughtful review. We are pleased that the reviewer found our results on communication- and memory-efficient joint adaptivity promising, and appreciated our convergence guarantees and empirical performance.
>
> ---
>
> **[On Communication Gains in Figure 2]**
>
> Thank you for the question regarding Figure 2. The communication savings may look less visually pronounced in the plot due to y-axis scale, but the improvement is significant when measured in actual *communicated bits* (i.e., number of rounds × bits per round). Additionally, $FedAda^2$/$FedAda^2$++ attains more benefits in resource usage:
>
>
> - $FedAda^2$ and $FedAda^2$++ achieve faster convergence with fewer bits transmitted than costly joint adaptivity.
> - When $FedAda^2$++ is instantiated with **SM3-AdaGrad** on ViT, the client-side memory overhead for storing preconditioners drops from **100% to just 0.48%**.
> - This results in a **99% reduction** in memory overhead, while still achieving strong performance.
>
> We will consider including a table or numerical summary in the revision to highlight these reductions more clearly alongside the visual plots.
>
> ---
>
> **[On Server-Side Preconditioner Computation]**
>
> We appreciate the reviewer’s observation regarding the server-side cost. While the server independently maintains a preconditioner, this cost is generally manageable, particularly when compared to the resource limitations on client devices. Prior work often assumes that the server is relatively powerful, and thus, in cross-device FL settings (e.g., mobile/IoT), it is often a favorable trade-off to reduce client memory and communication at the expense of modest server computation (e.g., [1-2]).
>
> ---
>
> **[On the Bounded Gradient Assumption]**
>
> We acknowledge that the coordinate-wise bounded gradient assumption is standard in prior analyses of adaptive FL (e.g., FedAdam, FedAdagrad, FedOPT). Our goal was to build upon and generalize this framework to jointly adaptive settings with memory-efficient client optimizers, which has not been done previously.
>
> That said, we agree it would be exciting to explore convergence under weaker assumptions (e.g., sub-Gaussian or heavy-tailed gradients). In fact, **Appendix A** presents a novel regret-based analysis in heavy-tailed regimes, which sheds light on how client-side adaptivity mitigates the propagation of noisy updates—particularly relevant in transformer-based training. We thank the reviewer for highlighting this direction and will aim to emphasize it more clearly in the revision.
>
> ---
>
> **[Final Remarks]**
>
> We are grateful for the reviewer’s feedback and insightful suggestions. In the revision, we will enhance the clarity of communication efficiency results (Figure 2), clarify server-side trade-offs, and more clearly highlight the theoretical contribution on heavy-tailed settings in Appendix A. We thank the reviewer again for thoughtful comments.
>
> [1] FS-Real: Towards Real-World Cross-Device Federated Learning. Daoyuan Chen, Dawei Gao, Yuexiang Xie, Xuchen Pan, Zitao Li, Yaliang Li, Bolin Ding, Jingren Zhou. Arxiv, 2023.
>
> [2]  A survey on federated learning: challenges and applications. Jie Wen, Zhixia Zhang, Yang Lan, Zhihua Cui, Jianghui Cai & Wensheng Zhang. International Journal of Machine Learning and Cybernetics, 2023.

---

> > ### Comment · Reviewer_T2Br · 2025-08-03
> >
> > Thanks for the reply. I will keep my score.

---

### Official Review · Reviewer_NuBp · 2025-07-02

**Clarity:** 2
**Significance:** 3
**Originality:** 2
**Rating:** 4
**Confidence:** 2

**Summary:**

The authors propose an adaptive federated optimizer featuring local preconditioner re‑initialization, global preconditioner updates, and memory‑efficient client optimizers, matching prior convergence rates and validated on image/text benchmarks. They further demonstrate that this design balances adaptivity with communication and memory efficiency, yielding performance on par with baselines under resource constraints.

**Questions:**

./.

**Ethical Concerns:**

["NO or VERY MINOR ethics concerns only"]

**Final Justification:**

After carefully considering my initial assessment in the light of the rebuttal, I am willing to raise the score.
The rebuttal was effective and sufficiently convincing explanation of the non-trivial nature of their convergence proofs.
While I believe that the contribution is correct, technically solid, and non-trivial, judging the magnitude of the contribution is not straightforward for me.
If one is willing to accept that the magnitude of this contribution is more than minor, I've to concur that the remaining points do not _clearly_ justify a rejection of this work. Therefore, the paper's strengths now outweigh its acknowledged weaknesses, making a score of borderline accept appropriate, while highlighting my uncertainty with lowering my weight.

**Limitations:**

- The authors list some technical limitations. Broader societal impacts (e.g. misapplication in high‑stakes domains) are not well discussed, but risks appear negligible.

**Paper Formatting Concerns:**

./.

**Quality:**

3

**Strengths And Weaknesses:**

Strengths

- Provides rigorous convergence proofs for non‑convex jointly adaptive federated training under minimal optimizer conditions.
- Reduces communication overhead by initializing local preconditioners to zero without degrading performance.
- Integrates memory‑efficient optimizers (e.g., SM3) to dramatically reduce client‑side memory.

Weaknesses

- Thm 4.1 is not introduced well, and is not very intuitive.
- The zero‑cost re‑initialization trick is conceptually obvious once you view the server preconditioner as an aggregator.
- Memory‑efficient optimizers (SM3) are basically plugged in.
- The theoretical insights beyond standard bounds are limited; proofs largely follow existing templates.
- Empirical gains are modest and confined to standard benchmarks, with no demonstration on truly large‑scale or heterogeneous deployments.
- Convergence rates match prior Ada‑style FL methods.
- Experiments lack ablation on key hyperparameters (e.g., smoothing ε, delay z) in main text.
- Scalability evaluation omits wall‑clock time or system‑level benchmarks in true cross‑device settings.
- Privacy study is limited to logistic regression on StackOverflow.
- Dataset diversity is modest (only four benchmarks), with no evaluation on non‑vision/text tasks or personalization objectives.
- Ablation of memory vs. accuracy trade‑offs for varying model sizes is not provided.
- Experiments with heterogeneous client data distributions do not control the degree of heterogeneity.

---

> ### Author Rebuttal · Authors · 2025-07-31
>
> We thank Reviewer NuBp for their review. We appreciate the recognition of our theoretical contributions and the practicality of our memory- and communication-efficient approach to joint adaptivity in federated learning. Below, we address each of the reviewer’s comments in turn.
>
> ---
>
> **[Theorem 4.1 and Theoretical Contributions]**
>
> We appreciate the reviewer’s comments on Theorem 4.1. The full version is given in **Appendix E**, and a key strength of this result is that it accommodates *different client-side optimizers*, enabled by the flexibility of $FedAda^2$/$FedAda^2$++.
>
>
> The theorem establishes a general non-convex convergence guarantee for $FedAda^2$/$FedAda^2$++, bounding the minimum gradient norm across $T$ rounds. Specifically, $\min_{t\in[T]} \|\nabla f(x_{t-1})\|^2$ is upper-bounded by $\sum_{i=1}^5 \Psi_i / \Psi_6$, where:
>
> - The numerator aggregates smoothness-based descent, effects of exponential moving averaging in adaptive preconditioning, and learning-rate based decay, and
> - The denominator reflects effective learning rate scaled by the adaptive server-side preconditioner, and is stabilized by the smoothing terms on the client or server.
>
> The proof requires non-trivial decompositions of momentum updates, exponential decay control using a new constant $c(\widetilde{\beta}_1)$, and log-sum bounds over adaptive gradients. Crucially, convergence is preserved even under fixed client step sizes, despite omitting preconditioner transmission.
>
> While the zero-cost reinitialization trick may appear conceptually simple once the server preconditioner is seen as an aggregator, proving convergence with reinitialization and joint adaptivity is non-trivial. Notably, the proof must incorporate $\varepsilon$-smoothing dependencies and account for clients retaining historical gradients—a challenge absent in typical FedOPT-style analyses.
>
> We also include a novel regret-based analysis in **Appendix A** for federated training in heavy-tailed regimes. To our knowledge, this is the first regret-based theoretical result showing that client-side adaptivity reduces gradient noise propagation in such settings—something not addressed by prior works.
>
> ---
>
> **[On Use of Memory-Efficient Optimizers]**
>
> We respectfully clarify that the statement “memory-efficient optimizers are basically plugged in” is incorrect for $FedAda^2$. Unlike $FedAda^2$++, $FedAda^2$ does not need to use memory-efficient optimizers by default, and its implementation and analysis do not rely on such plug-ins.
>
> ---
>
> **[Empirical Results and Performance Gains]**
>
> We respectfully disagree with the assessment that empirical gains are modest. Figure 1 shows consistent and meaningful improvements across all datasets:
>
> - Adaptive methods (e.g., FedAdam, FedAdaGrad) outperform FedAvg.
> - Jointly adaptive methods ($FedAda^2$, $FedAda^2$++) consistently outperform server-only adaptive baselines.
> - "Costly Joint Adaptivity" achieves similar accuracy to $FedAda^2$, but with **significantly** higher communication costs. Relaxing memory consumption by deploying $FedAda^2$++ does not meaningfully degrade performance, and in some cases, even enhances performance in noisy DP-settings (e.g., Appendix J, Figure 9).
>
> The benefits are especially pronounced on **FEMNIST** or **StackOverflow**, where the stratified gap between FedAvg, server-only adaptivity, and joint adaptivity is clearly visible.
>
> In **Figure 2**, when evaluating convergence in terms of *actual communicated bits*, $FedAda^2$ and $FedAda^2$++ significantly outperform costly joint adaptivity. Additionally, with ViT, $FedAda^2$++ instantiated via SM3-AdaGrad requires only **0.48%** additional memory to maintain client preconditioning, representing a **99% reduction** in extra client memory cost. This makes joint adaptivity significantly more practical for large-scale, resource-constrained deployments.
>
> ---
>
> **[Convergence Rate Comparison]**
>
> We agree that our convergence rate matches that of prior adaptive FL methods, as this is the best-known result for non-convex optimization: $\mathcal{O}(1/\sqrt{T})$. However, achieving this rate in the memory- and communication-efficient jointly adaptive setting is novel, especially when clients retain historical gradients and when no preconditioners are transferred.
>
> In FedOPT-style proofs, linearity of expectation allows clean reductions to server-only adaptivity. In contrast, joint adaptivity complicates the analysis due to time-dependent preconditioners on both client and server sides. Our work is the first, to our knowledge, to rigorously establish this rate under such general and practical conditions.
>
> ---
>
> **[Hyperparameter Ablations]**
>
> As mentioned in the main text, we in fact do include very extensive hyperparameter ablations, though these are deferred to the appendix for space reasons. To summarize:
>
> - **Appendix I** contains detailed sweeps over learning rates and $\varepsilon$.
> - **Appendix J** contains ablations on the delay parameter $z$, including what happens when there is a *mismatch* between the client and server adaptive optimizers (e.g., server-side Adam preconditioners communicated to client-side AdaGrad).
>
> ---
>
> **[Benchmark Diversity and Modalities]**
>
> We selected four well-established benchmarks across vision and text, representing the dominant modalities in FL research. These datasets and modalities are widely used in federated learning papers (e.g., FedOPT [1], FedNova [2], FedNAR [3]), and are considered standard baselines.
>
> Other modalities (e.g., audio) or personalization objectives are not the focus of our study, as our goal was to provide a general-purpose framework with convergence guarantees in the jointly adaptive case where preconditioners were not transmitted for memory and communication efficiency.
>
> ---
>
> **[Ablation on Memory–Accuracy Tradeoffs & Heterogeneity Control]**
>
> Our goal is to demonstrate the feasibility of joint adaptivity with strong theoretical guarantees under practical constraints. While trade-off studies (e.g., model size vs. accuracy scaling laws) and controlled heterogeneity sweeps are interesting, they fall outside the scope of this work. That said, our convergence result does allow different optimizers per client, making such studies possible in future work.
>
> ---
>
> **[Final Remarks]**
>
> We thank the reviewer for their feedback. In the revision, we will clarify the theoretical contributions and expand the empirical discussions. We hope this response has addressed the reviewer’s concerns, and please let us know if there are any further questions.
>
> [1] Adaptive Federated Optimization. Sashank Reddi, Zachary Charles, Manzil Zaheer, Zachary Garrett, Keith Rush, Jakub Konečný, Sanjiv Kumar, H. Brendan McMahan. International Conference on Learning Representations, 2021.
>
> [2] Tackling the Objective Inconsistency Problem in Heterogeneous Federated Optimization. Jianyu Wang, Qinghua Liu, Hao Liang, Gauri Joshi, H. Vincent Poor. Neural Information Processing Systems, 2020.
>
> [3] FedNAR: Federated Optimization with Normalized Annealing Regularization. Junbo Li, Ang Li, Chong Tian, Qirong Ho, Eric P. Xing, Hongyi Wang. Neural Information Processing Systems, 2023.

---

> > ### Comment · Reviewer_NuBp · 2025-08-08
> >
> > Thank you for your clear and thoughtful response. I particularly appreciate the discussion surrounding Theorem 4.1. Although I don’t agree with your assessment of the magnitude of your contribution and find some limitations in your evaluation, I accept your rebuttal and adjust my score accordingly.

---

### Official Review · Reviewer_egAL · 2025-07-03

**Clarity:** 3
**Significance:** 2
**Originality:** 3
**Rating:** 5
**Confidence:** 3

**Summary:**

This paper presents efficient adaptive federated optimization algorithms FedAda2 and FedAda2++. Specifically, these two algorithms avoid the transfer of preconditioners between the server and clients, and also incorporates memory-efficient adaptive optimizers on the client side. They shows the advantage both theoretically and empirically.

**Questions:**

1. What is the key new insight in the theoretical analysis, considering that the algorithm is fairly general? For instance, prior work such as [1] also provides theoretical guarantees for a general algorithmic framework. How does this work differ or advance the theory?


Ref:
1. Fednar: Federated optimization with normalized annealing regularization

**Ethical Concerns:**

["NO or VERY MINOR ethics concerns only"]

**Final Justification:**

Overall, I think this work solid and above the average quality of academic research in FL.

**Limitations:**

It's better to have experiments in larger scale.

**Quality:**

3

**Strengths And Weaknesses:**

Strengths:
1. The methods are theoretically well-founded. The paper first introduces a general algorithm, proves its convergence, and then develops a practical empirical version.
2. The writing is clear and easy to follow.

Weaknesses:
1. The empirical results, as shown in Figure 1, do not demonstrate a clear improvement over the baselines.
2. The scale of the experiments is limited. Given the current state of the field in 2025, more large-scale evaluations would be expected.
3. It would be better to have more insight on the advantage of certain joint adaptivity.

---

> ### Author Rebuttal · Authors · 2025-07-31
>
> We thank Reviewer egAL for their review and are glad that the reviewer found the theoretical foundations of $FedAda^2$ and $FedAda^2$++ to be rigorous, and the writing clear and easy to follow. Below, we incorporate the reviewer’s feedback regarding empirical results, experimental scale, and theoretical novelty.
>
> ---
>
>
> **[Empirical Results and Effectiveness of Joint Adaptivity]**
>
> The empirical results in Figure 1 do show clear improvements—that is, across all datasets, we observe consistent trends:
>
> - Adaptive methods (e.g., FedAdam, FedAdaGrad) outperform non-adaptive FedAvg.
> - Jointly adaptive methods (e.g., $FedAda^2$, $FedAda^2$++) converge faster and often achieve higher accuracy than server-only adaptive baselines.
> - While "Costly Joint Adaptivity" performs comparably to $FedAda^2$, our methods achieve this with drastically lower memory and communication costs.
>
> This performance gap is especially notable on **FEMNIST**, where we see clear separation between non-adaptive, server-only adaptive, and jointly adaptive methods. These trends are present across all datasets, demonstrating that joint adaptivity consistently brings empirical benefit, even when communication of preconditioners is avoided.
>
> In Figure 2, when convergence is measured by *actual communicated bits* (i.e., communication rounds × bits per round), $FedAda^2$ and $FedAda^2$++ significantly outperform costly joint adaptivity. For example, with ViT, when $FedAda^2$++ is instantiated via SM3-AdaGrad, client memory overhead for preconditioning drops from 100% to only **0.48%**—a **99% reduction**—making joint adaptivity far more practical for large-scale and resource-constrained deployments.
>
> ---
>
> **[Scope and Scale of Experiments]**
>
> We appreciate the suggestion to explore larger-scale experiments. Our goal in this paper is to target **cross-device** FL—settings such as mobile phones, wearables, or edge devices—where communication and memory constraints are significant.
>
> Our experiments are intentionally designed to reflect memory- and communication-constrained scenarios, and we demonstrate that our algorithms retain strong convergence behavior in such environments. While large-scale experiments (e.g., training LLaMA-like models) are a valuable future direction, they fall outside the scope of this work. Training those models on mobile phones is not the target use case of $FedAda^2$/$FedAda^2$++, and we will clarify this further in the revised manuscript.
>
> ---
>
> **[Theoretical Contributions and Comparison to Prior Work]**
>
> We thank the reviewer for prompting a deeper discussion of theoretical novelty. The key theoretical insight in our work is that **the benefits of joint adaptivity can be retained even without transmitting preconditioners**, as long as one controls $\varepsilon$-smoothing and learning rate schedules. Our analysis identifies important terms (e.g., $\varepsilon$) that ensure convergence under these constraints. This extends beyond prior work, such as FedNAR, which does not analyze joint adaptivity and focuses instead on regularization. Additionally, **Appendix A** provides theoretical insight into the *heavy-tailed gradient noise regime*, which is common in modern transformer training. We show that client-side adaptivity reduces regret propagation from heavy tails, which to our knowledge has not previously been reported in the FL literature.
>
> ---
>
> **[Comparison with FedNAR]**
>
>
> FedNAR aims to improve convergence via annealing-based weight decay and is largely orthogonal to our contributions. It primarily introduces regularization in an FL setting. In contrast:
>
>
> - $FedAda^2$/$FedAda^2$++ focus on **communication and memory efficiency**, especially in avoiding the transmission of local preconditioners.
> - Our algorithms implement **joint adaptivity**, combining client- and server-side adaptive updates, which requires a distinct convergence analysis.
> - By contrast, in the theoretical convergence analysis, FedNAR is analyzed only as a client-only adaptive method, as opposed to $FedAda^2$/$FedAda^2$++, which is analyzed as a jointly adaptive method.
>
> These frameworks are complementary, and FedNAR techniques can also be incorporated into our adaptive structure—e.g., $FedAda^2$++ with FedNAR-style decay.
>
> ---
>
> **[Final Remarks]**
>
> We thank the reviewer again for their feedback. In the revision, we will clarify the theoretical contributions, expand the discussion of empirical trends, and better motivate our focus on the cross-device federated learning setting. We would be happy to address any additional questions or suggestions.

---

> > ### Comment · Reviewer_egAL · 2025-08-07
> >
> > Thank you to the authors for the rebuttal. I am raising my score to accept. It would be better to clarify the theoretical contributions compared with existing frameworks and to expand the empirical observations in the revision as mentioned.

---

> > > ### Author Response · Authors · 2025-08-07
> > >
> > > Thank you for taking the time to carefully review our work and for raising your score to accept. We appreciate your constructive suggestions on clarifying the theoretical contributions relative to existing frameworks and on expanding the empirical discussion, and will ensure that the revised manuscript reflects these improvements.

---

### Official Review · Reviewer_CoJv · 2025-07-11

**Clarity:** 2
**Significance:** 3
**Originality:** 2
**Rating:** 3
**Confidence:** 4

**Summary:**

The paper introduces less resource-intensive adaptive algorithms, FedAda and FedAda++, for cross-device FL, and explores joint server and client-side adaptivity.  FedAda avoids the transfer of preconditioners between server and clients, and FedAda++ uses memory-efficient adaptive optimizers on clients. The authors establish similar convergence guarantees to those of the resource-intensive counterparts like FedAdam for general non-convex objectives with a coordinate-wise bounded gradient assumption.

PS: Not sure if you really need a superscript $.^2$ in the algorithm's name. What exactly will be difficulty if that is avoided?

**Questions:**

In addition to addressing the weaknesses mentioned above, see the following:

1. Table 1 does not contain the statistics for FedAdam. Can the authors explain why? It would have been insightful to compare the efficiency and performance of FedAda algorithms against FedOpt algorithms in [2].

2. The authors state that they aim to increase communication efficiency by minimizing the transfer of local preconditioners, as opposed to other works that use model compression like FedCAMS, FedAMS [1], etc. Can the authors explain or compare the efficiency metrics
between these two approaches? And which one to use in specific scenarios?

[1] Yujia Wang, Lu Lin, and Jinghui Chen. Communication-efficient adaptive federated learning. In International Conference on Machine Learning, pages 22802–22838. PMLR, 2022.

[2] Sashank Reddi, Zachary Charles, Manzil Zaheer, Zachary Garrett, Keith Rush, Jakub Konecný, Sanjiv Kumar, and Brendan McMahan. Adaptive federated optimization. International Conference on Learning Representations, 2021.

**Ethical Concerns:**

["NO or VERY MINOR ethics concerns only"]

**Final Justification:**

I acknowledged the rebuttal of the co-authors. I believe a restructure of the theoretical discussion in the paper to improve its readability is required. Will keep my score unchanged.

**Limitations:**

None.

**Paper Formatting Concerns:**

None.

**Quality:**

2

**Strengths And Weaknesses:**

**Strengths:**

1. The FedAda framework permits the use of different optimizers at the client and server sides.

**Weaknesses:**

2. The proofs presented are difficult to follow through and seem to introduce a lot of abstractions without sufficient explanation. A clearer explanation of the steps involved in high-level ideas at some points may aid the reader's understanding of the proofs. Eg., proof (ii) of Proposition A.3. is quite dense and should be clearly explained for better understanding.

3. Line 620 is unclear. It is challenging to assess the validity of the main theoretical results.

4. The experiments are conducted on an array of datasets and demonstrate that FedAda and FedAda++ performs better than other existing popular adaptive federated learning algorithms and are at par with costly joint adaptivity.

4. Overall, the readability of the paper can be substantially improved in the main body and proofs.

---

> ### Author Rebuttal · Authors · 2025-07-31
>
> We thank Reviewer CoJv for their thoughtful review. We appreciate the recognition of our proposed jointly adaptive framework—$FedAda^2$ and $FedAda^2$++—as being less resource-intensive while achieving strong empirical performance and convergence guarantees. Below, we incorporate the reviewer’s feedback.
>
> ---
>
> **[Clarity and Proof Abstractions]**
>
> We agree with the reviewer that some parts of the proofs, such as the second part of Proposition A.3, are dense and could benefit from higher-level explanation. In the revision, we will include more intuitive walkthroughs and clarify proof sketches in the appendix. Specifically, for Proposition A.3(ii), the result shows that under client-side AdaGrad with normalized gradients, the expected iterate norm converges to a noise-dependent constant at rate $\mathcal{O}(1/t)$, uniformly over client subsamples. The key idea is a decomposition of the update into a contraction term and a noise-dependent residual, showing the decay of the contraction under suitable $\varepsilon$ and learning rate schedules.
>
> Regarding Line 620, it refers back to the learning rate schedule defined in Line 618. We will make this reference clearer in the revised version. Generally, the abstractions introduced to generalize beyond FedAdam, or to apply a novel regret-based analysis in heavy-tailed settings as done in **Appendix A**, are non-trivial. Some of them were required to accommodate the jointly adaptive setting, where gradients are retained over time and related nonlinearly—preventing the use of standard linear expectation-based techniques. More broadly, we agree that some abstractions introduced in the proofs can be softened, and will include additional intuition and guiding commentary to make the results more accessible for dense derivations.
>
> ---
>
> **[About the Algorithm Naming Convention ($FedAda^2$, $FedAda^2$++)]**
>
> We use the superscript in $FedAda^2$ and $FedAda^2$++ to emphasize a core conceptual novelty: joint adaptivity on both the client and server sides. This setting substantially complicates the analysis—unlike server-only adaptive methods (e.g., FedAdam), the client adaptive updates retain gradient history locally and introduce nonlinear couplings across time, making standard FedOPT-style techniques inapplicable in the analysis. Our convergence proof thus departs from the FedAdam analysis and introduces new tools to handle these technical challenges. We appreciate the reviewer’s feedback on the proof complexity and will work to add clearer high-level overviews and intuitive scaffolding in the final version.
>
> ---
>
> **[Table 1 and FedAdam Statistics]**
>
> The reviewer asked about FedAdam in Table 1– its statistics are identical to those of FedAdaGrad on the client side, as both methods are server-only adaptive methods. For clarity, we will include a row explicitly labeled FedAdam in Table 1 of the revision. The updated table is:
>
> | Method              | Joint Adaptivity? | Communication | Computation (#grad. calls) | Memory (client) |
> |---------------------|-------------------|----------------|-----------------------------|------------------|
> | FedAvg              | N                 | 2d             | 1                           | d                |
> | FedAdaGrad          | N                 | 2d             | 1                           | d                |
> | FedAdam             | N                 | 2d             | 1                           | d                |
> | MIME                | N                 | 5d             | 3                           | 4d               |
> | MIMELite            | N                 | 4d             | 2                           | 3d               |
> | Costly Joint Adap.  | Y                 | 3d             | 1                           | 2d               |
> | $FedAda^2$          | Y                 | 2d             | 1                           | 2d               |
> | $FedAda^2$++        | Y                 | 2d             | 1                           | ~d               |
>
> We note that $FedAda^2$++ achieves performance comparable to costly joint adaptivity while using *99% less* additional client memory needed to maintain client preconditioners (i.e., only 0.48% overhead when using SM3-AdaGrad with ViT, compared to a full 1× increase in second-moment memory for standard joint methods). This makes $FedAda^2$++ particularly appealing in memory-constrained or differentially private (DP) settings in FL.
>
> ---
>
> **[Comparison with FedCAMS and FedAMS]**
>
> We appreciate the reviewer’s question regarding comparisons with FedCAMS and FedAMS. These methods focus on *compression* of local models for server-only adaptive optimizers, whereas our contribution lies in minimizing *communication* of preconditioners in *jointly adaptive* settings. Crucially, our framework is complementary to these techniques and can flexibly incorporate their compression strategies. We emphasize this core flexibility in Section 2 (Lines 63–82), and we will strengthen this point further in the revision.
>
> ---
>
> **[Performance Strength Clarification]**
>
> We believe the reviewer’s fourth weakness—that $FedAda^2$/$FedAda^2$++ outperforms existing adaptive FL algorithms and matches costly joint adaptivity—was intended as a strength. For example, in Figure 2, when evaluating convergence by *actual communicated bits*, $FedAda^2$ and $FedAda^2$++ outperform costly joint adaptivity substantially. This demonstrates that our methods are not only communication-efficient in theory, but also in practice.
>
> ---
>
> **[Final Remarks]**
>
> We thank the reviewer again for their constructive suggestions. We are committed to improving the clarity and accessibility of the proofs, as well as refining the empirical narrative around joint adaptivity and communication efficiency. We hope this response has addressed the reviewer’s concerns, and if so, respectfully request reconsideration of the overall score.
>
> Please let us know if any points remain unclear—we would be happy to elaborate further.

---

> > ### Comment · Reviewer_CoJv · 2025-08-04
> > **Appreciate the rebuttal**
> >
> > Thank you for your response.
> >
> > I appreciate the co-author's promise to improve the readability of the submission. I am of the opinion that the community will benefit from a revised submission of this work to the next top ML venue. With this motivation, I will keep my score unchanged and will leave it to other peer reviewers to decide on the acceptance of this paper.
> >
> >
> > Thank you.

---

### Comment · Area_Chair_WjJ8 · 2025-08-05

Dear Reviewers,

Thank you again for your time and efforts in reviewing papers for NeurIPS 2025.

I am writing to remind you that **active participation in the author-reviewer discussion phase is mandatory**. According to the guidelines from the NeurIPS program chairs, reviewers are **required to engage directly with the authors in the discussion thread**, especially in response to their rebuttals.

Please note the following important policy:

- Simply reading the rebuttal or internally considering it is **not sufficient** -- reviewers must **post at least one message to the authors**, even if it is only to confirm that their concerns were resolved. If they have not been addressed, please explain why.

- **Acknowledging the rebuttal without any engagement with the authors will be considered insufficient**. I am obligated to flag such cases using the *InsufficientReview* mechanism, which may **impact future reviewing invitations and result in desk rejection of your own submissions**.

If you have not yet responded to the authors in the discussion thread, I kindly ask you to do so **as soon as possible**, and **no later than August 8, 11:59pm AoE**.

Please don't hesitate to reach out to me if you have any questions or concerns.

Best regards,

AC

---

### Note · Authors · 2025-08-12

We thank all reviewers for their thoughtful evaluations and feedback during the discussion phase. We are encouraged that multiple reviewers recognized the practical relevance of $FedAda^2$ and $FedAda^2$++ for cross-device federated learning, our rigorous convergence analysis under joint adaptivity, and the significant communication and memory savings achieved without sacrificing accuracy.

We also appreciate the suggestions to further clarify the work. In summary, we will be providing the following modifications:

- Clarity of proofs: We will provide higher-level overviews of dense derivations (e.g., Proposition A.3(ii), Theorem 4.1) and improve cross-references in the main text, ensuring that the key technical ideas are intuitive and accessible.

- Positioning of theoretical contributions: We will clarify how our analysis extends or complements prior frameworks such as FedOPT and FedNAR, particularly in handling joint adaptivity in the absence of preconditioner transmission.

- Empirical presentation: We will expand discussion of observed trends and make the role and benefits of joint adaptivity more explicit.

- Experimental ablations: We will highlight additional hyperparameter ablations provided in the appendix in the main text based on some reviewer feedback.

We value the positive assessments that our approach is both practical and theoretically grounded, and that it opens a promising direction for efficient adaptive FL.

We thank the reviewers again for their time and constructive dialogue.

---

### Decision · Program_Chairs · 2025-09-17

**Decision:**

Accept (poster)

**Comment:**

**Summary.** This paper introduces a general framework for adaptive federated optimization, where clients may employ different adaptive optimization methods for local updates, and the server aggregates pseudogradients to perform a global adaptive step. Under assumptions of smoothness, bounded gradients, and mild restrictions on the form of the local optimizer - satisfied by standard methods such as SGD, AdaGrad, and Adam - the authors establish an $\mathcal{O}(1/\sqrt{T})$ convergence rate for the best squared norm of the gradient along the trajectory.

**Strengths and weaknesses.** The paper makes a meaningful contribution to the literature on adaptive federated optimization. Three reviewers (NuBp, T2Br, egAL) rated it positively (scores 4, 5, 5), while one reviewer (CoJv) leaned toward borderline rejection, primarily citing readability concerns. I agree that the readability needs improvement, but the rebuttal suggests the authors have adequately addressed this issue.

A notable weakness is the reliance on the bounded gradients assumption in addition to smoothness. While somewhat restrictive, this assumption is common in the analysis of Adam/AdaGrad-type methods and was also flagged by Reviewer T2Br. On balance, the mentioned strengths clearly outweigh the mentioned weaknesses.

**Requested clarifications.** One important point, overlooked by reviewers, is that Theorem 4.1 (Theorem E.1) analyzes a deterministic method: there are no assumptions on stochasticity, and the convergence guarantee holds with probability 1. It is crucial that the final version explicitly states that this convergence bound applies only in the deterministic setting. I also ask the authors to incorporate all promised changes from the rebuttal.

**Final recommendation.** I recommend acceptance of this paper, conditional on the authors making the requested clarifications and revisions.